# Instance-Dependent Near-Optimal Policy Identification in Linear MDPs via Online Experiment Design

**Andrew Wagenmaker**[*] **& Kevin Jamieson**[†]
Paul G. Allen School of Computer Science & Engineering
University of Washington
Seattle, WA 98195

## Abstract

While much progress has been made in understanding the minimax sample complexity of reinforcement learning (RL)—the complexity of learning on the "worst-case" instance—such measures of complexity often do not capture the true difficulty of learning. In practice, on an "easy" instance, we might hope to achieve a complexity far better than that achievable on the worst-case instance. In this work we seek to understand the "instance-dependent" complexity of learning near-optimal policies (PAC RL) in the setting of RL with linear function approximation. We propose an algorithm, PEDEL, which achieves a fine-grained instance-dependent measure of complexity, the first of its kind in the RL with function approximation setting, thereby capturing the difficulty of learning on each particular problem instance. Through an explicit example, we show that PEDEL yields provable gains over low-regret, minimax-optimal algorithms and that such algorithms are unable to hit the instance-optimal rate. Our approach relies on a novel online experiment design-based procedure which focuses the exploration budget on the "directions" most relevant to learning a near-optimal policy, and may be of independent interest.

## 1 Introduction

In the PAC (Probably Approximately Correct) reinforcement learning (RL) setting, an agent is tasked with exploring an unknown environment in order to learn a policy which maximizes the amount of reward collected. In general, we are interested in learning such a policy using as few interactions with the environment (as small sample complexity) as possible. We might hope that the number of samples needed would scale with the difficulty of identifying a near-optimal policy in our particular environment. For example, in a "hard" environment, we would expect that more samples might be required, while in an "easy" environment, fewer samples may be needed.

The RL community has tended to focus on developing algorithms which have near-optimal *worst-case* sample complexity—sample complexities that are only guaranteed to be optimal on "hard" instances. Such algorithms typically have complexities which scale, for example, as $\mathcal{O}(\mathrm{poly}(d, H)/\epsilon^2)$, for $d$ the dimensionality of the environment, $H$ the horizon, and $\epsilon$ the desired level of optimality. While we may be able to show this complexity is optimal on a hard instance, it is unable to distinguish between "hard" and "easy" problems. The scaling is identical for two environments as long as the dimensionality and horizon of each are the same—no consideration is given to the actual difficulty of the problem—and we therefore have no guarantee that our algorithm is solving the problem with complexity scaling as the actual difficulty. Indeed, as recent work has shown ([Wagenmaker et al.](#),

[*]ajwagen@cs.washington.edu
[†]jamieson@cs.washington.edu

36th Conference on Neural Information Processing Systems (NeurIPS 2022).

), this is not simply an analysis issue: worst-case optimal algorithms can be very suboptimal on "easy" instances.

Towards developing algorithms which overcome this, we might instead consider the *instance-dependent* difficulty—the hardness of solving a particular problem instance—and hope to obtain a sample complexity scaling with this instance-dependent difficulty, thereby guaranteeing that we solve "easy" problems using only a small number of samples, but still obtain the worst-case optimal rate on "hard" problems. While progress has been made in understanding the instance-dependent complexity of learning in RL, the results are largely limited to environments with a finite number of states and actions. In practice, real-world RL problems often involve large (even infinite) state-spaces and, in order to solve such problems, we must generalize across states. To handle such settings, the RL community has turned to function approximation-based methods, which allow for provable learning in large state-space environments. However, while worst-case optimal results have been shown, little is understood on the instance-dependent complexity of learning in these settings.

In this work we aim to bridge this gap. We consider, in particular, the linear MDP setting, and develop an algorithm which provably learns a near-optimal policy with sample complexity scaling as the difficulty of each individual instance. Furthermore, by comparing to our instance-dependent measure of complexity, we show that low-regret algorithms are provably suboptimal for PAC RL in function approximation settings. Our algorithm relies on a novel online experiment design-based procedure—adapting classical techniques from linear experiment design to settings where *navigation* is required to measure a particular covariate—which may be of independent interest.

## 1.1 Contributions

Our contributions are as follows:

- We propose an algorithm, PEDEL, which learns an $\epsilon$-optimal policy with instance-dependent sample complexity scaling as (up to $H$ factors):

$$\sum_{h=1}^{H} \inf_{\pi_{\exp}} \max_{\pi \in \Pi} \frac{\|\phi_{\pi,h}\|^2_{\Lambda^{-1}_{\pi_{\exp},h}}}{(V_0^\star - V_0^\pi)^2 \vee \epsilon^2} \cdot \left( d + \log \frac{1}{\delta} \right)$$

  for $\phi_{\pi,h}$ the "average feature vector" of policy $\pi$, $\Lambda_{\pi_{\exp},h}$ the expected covariance of the policy $\pi_{\exp}$, and $V_0^\star - V_0^\pi$ the "policy gap". We show that PEDEL also has worst-case optimal dimension-dependence—its sample complexity never exceeds $\widetilde{\mathcal{O}}(d^2 H^7/\epsilon^2)$—but that on "easy" instances it achieves complexity much smaller than the worst-case optimal rate.

- It is well-known that low-regret algorithms achieve the worst-case optimal rate for PAC RL. We construct an explicit example, however, where the instance-dependent complexity of PEDEL improves on the complexity of *any* low-regret algorithm by a factor of the dimensionality, providing the first evidence that low-regret algorithms are provably suboptimal on "easy" instances for PAC RL in function approximation settings.

- We develop a general experiment design-based approach to exploration in MDPs, which allows us to focus our exploration in the directions most relevant to learning near-optimal policies. Our approach is based on the key observation that, while solving an experiment design in an MDP would require knowledge of the MDP dynamics, we can approximately solve one without knowledge of the dynamics by running a regret minimization algorithm on a carefully chosen reward function, inducing the correct exploration.

## 2 Related Work

The sample complexity of RL has been studied for decades (Kearns & Singh, 1998; Brafman & Tennenholtz, 2002; Kakade, 2003). The two primary problems considered are the regret minimization problem (where the goal is to obtain large online reward) and the PAC policy identification problem (where the goal is to find a near-optimal policy using as few samples as possible), which is the focus of this work. In the tabular RL setting, the question of obtaining worst-case optimal algorithms is nearly closed (Dann & Brunskill, 2015; Dann et al., 2019; Ménard et al., 2020; Zhang et al., 2020). As such, in this section we focus primarily on results in the RL with function approximation literature, as well as results on instance-dependent RL.

**Sample-Efficient RL with Linear Function Approximation.** To generalize beyond MDPs with a finite number of states and acions, the RL community has considered *function approximation*, replacing the tabular model with more powerful settings that allow for generalization across states. Such settings have been considered in classical works (Baird, 1995; Bradtke & Barto, 1996; Sutton et al., 1999; Melo & Ribeiro, 2007), yet these works do not provide polynomial sample complexities. More recently, there has been intense interest in obtaining polynomial complexities for general function classes (Jiang et al., 2017; Du et al., 2021; Jin et al., 2021; Foster et al., 2021), and, in particular, linear function classes (Yang & Wang, 2019; Jin et al., 2020; Wang et al., 2019; Du et al., 2019; Zanette et al., 2020a,b; Ayoub et al., 2020; Jia et al., 2020; Weisz et al., 2021; Zhou et al., 2020, 2021; Zhang et al., 2021; Wang et al., 2021).

In the linear MDP setting, the state-of-the-art in PAC RL is the work of Wagenmaker et al. (2022), which proposes a computationally efficient algorithm achieving a complexity of $\mathcal{O}(d^2 H^5/\epsilon^2)$ for the more general reward-free RL problem, and shows a matching lower bound of $\Omega(d^2 H^2/\epsilon^2)$ for the PAC RL problem. While this result obtains tight dimension-dependence, it is still worst-case, and offers no insight on the instance-dependent complexity. Other works of note in this category are (Jin et al., 2020; Zanette et al., 2020b; Zhou et al., 2020), which establish regret guarantees in the setting of linear MDPs and the related setting of linear mixture MDPs. Jin et al. (2020) and Zanette et al. (2020b) obtain regret guarantees of $\mathcal{O}(\sqrt{d^3 H^4 K})$ and $\mathcal{O}(\sqrt{d^2 H^4 K})$, respectively, though the approach of Zanette et al. (2020b) is computationally inefficient. Via an online-to-batch conversion (Jin et al., 2018), these algorithms achieve PAC complexities of $\mathcal{O}(d^3 H^4/\epsilon^2)$ and $\mathcal{O}(d^2 H^4/\epsilon^2)$. In the setting of linear mixture MDPs, Zhou et al. (2020) show a regret bound of $\mathcal{O}(\sqrt{d^2 H^3 K})$ and a matching lower bound, yielding the first provably tight and computationally efficient algorithms for RL with function approximation.

**Instance-Dependent RL.** Much of the recent work on instance-dependent RL has focused on the tabular setting. Ok et al. (2018) provide an algorithm which achieves asymptotically optimal instance-dependent regret, yet it is computationally inefficient. Simchowitz & Jamieson (2019) show that standard optimistic algorithms achieve regret bounded as $\mathcal{O}(\sum_{s,a,h} \frac{\log K}{\Delta_h(s,a)})$, for $\Delta_h(s,a)$ the *value-function gap*, a result later refined by (Xu et al., 2021; Dann et al., 2021). Obtaining instance-dependent guarantees for policy identification has proved more difficult, yet a variety of results do exist (Zanette et al., 2019; Jonsson et al., 2020; Marjani & Proutiere, 2020; Marjani et al., 2021). In the tabular setting, the most comparable work to ours is that of Wagenmaker et al. (2021b), which propose a refined instance-dependent measure of complexity, the *gap-visitation complexity*, and show that it is possible to learn an $\epsilon$-optimal policy with complexity scaling as the gap-visitation complexity. While the gap-visitation complexity is shown to be tight in certain settings, no general lower-bounds exist. Towards obtaining sharp guarantees, Tirinzoni et al. (2022) show that in the simpler setting of deterministic MDPs, a quantity similar in spirit to the gap-visitation complexity is tight, providing matching upper and lower bounds.

In the setting of RL with function approximation, to our knowledge, only two existing works obtain guarantees that would be considered "instance-dependent". Wagenmaker et al. (2021a) show a "first-order" regret bound of $\mathcal{O}(\sqrt{d^3 H^3 V_0^\star K})$, where $V_0^\star$ is the value of the *optimal* policy on the particular MDP under consideration. He et al. (2020) show that standard optimistic algorithms achieve regret guarantees of $\mathcal{O}(\frac{d^3 H^5 \log K}{\Delta_{\min}})$ and $\mathcal{O}(\frac{d^2 H^5 \log^3 K}{\Delta_{\min}})$ in the settings of linear MDPs and linear mixture MDPs, respectively, for $\Delta_{\min}$ the minimum value-function gap. While both these works do obtain instance-dependent results, the instance-dependence is rather coarse, depending on only a single parameter ($V_0^\star$ or $\Delta_{\min}$)—our goal will instead be to obtain more refined instance-dependent guarantees.

**Experiment Design in Sequential Environments.** Experiment design is a well-developed subfield of statistics, and a full survey is beyond the scope of this work (see Pukelsheim (2006) for an overview). We highlight several works on experiment design in sequential environments that are particularly relevant. First, the work of Fiez et al. (2019) achieves the instance-optimal rate for best-arm identification in linear bandits and relies on an adaptive experiment design-based. The approach of Fiez et al. (2019), as well as the related work of Soare et al. (2014), provides inspiration for our algorithm—in some sense PEDEL can be seen as a generalization of the RAGE algorithm to problems with horizon greater than 1. Concurrent to our work, the RAGE algorithm was extended to the setting of contextual bandits in Li et al. (2022), achieving the instance-optimal rate. Second, the work of Wagenmaker et al. (2021c) provides an experiment design-based algorithm in the setting

of linear dynamical systems, and show that it hits the optimal instance-dependent rate for learning in such systems. While their results are somewhat more general, they specialize to the problem of identifying a near-optimal controller for the LQR problem—thereby solving the PAC RL problem optimally in the special case of quadratic losses and linear dynamical systems. It is not clear, however, if their approach generalizes beyond linear dynamical systems. Finally, while the current work was in preparation, Mutny et al. (2022) proposed an approach to solving experiment design problems in MDPs. To our knowledge, this is the only existing work that directly considers the problem of experiment design in MDPs. However, they make the simplifying assumption that the transition dynamics are *known*, which essentially reduces their problem to a computational one—in contrast, our approach handles the much more difficult setting of unknown dynamics, and shows that efficient experiment design is possible even in this more difficult setting. We remark as well that our online experiment design approach is somewhat related to several existing algorithms (Hazan et al., 2019; Zahavy et al., 2021).

## 3   Preliminaries

We let $\|\phi\|_{\mathbf{A}}^2 = \phi^\top \mathbf{A}\phi$, $\|\cdot\|_{\mathrm{op}}$ denote the matrix operator norm (matrix 2-norm), and $\|\cdot\|_{\mathrm{F}}$ denote the Frobenius norm. $\widetilde{\mathcal{O}}(\cdot)$ hides absolute constants and log factors of the arguments. $\lesssim$ denotes inequality up to constants. $\mathbb{E}_\pi$ and $\mathbb{P}_\pi$ denote the expectation and probability measure induced by playing some policy $\pi$ in our MDP. We let $\phi_{h,\tau} := \phi(s_{h,\tau}, a_{h,\tau})$ denote the feature vector encountered at step $h$ of episode $\tau$ (and similarly define $r_{h,\tau}$).

**Markov Decision Processes.**   In this work, we study episodic, finite-horizon, time inhomogeneous Markov Decision Processes (MDPs), denoted by a tuple, $\mathcal{M} = (\mathcal{S}, \mathcal{A}, H, \{P_h\}_{h=1}^H, \{\nu_h\}_{h=1}^H)$. We let $\mathcal{S}$ denote the state space, $\mathcal{A}$ the action space, $H$ the horizon, $\{P_h\}_{h=1}^H$ the transition kernel, and $\{\nu_h\}_{h=1}^H$ the reward distribution, where $P_h(\cdot|s,a) \in \triangle_{\mathcal{S}}$ denotes the distribution over the next state when playing action $a$ in state $s$ at step $h$, and $\nu_h(s,a) \in \triangle_{[0,1]}$ denotes the corresponding distribution over reward. We overload notation and let $\nu_h(s,a)$ also refer to the expected reward. We assume that every episode starts in state $s_1$, and that $\{P_h\}_{h=1}^H$ and $\{\nu_h\}_{h=1}^H$ are initially unknown.

Let $\pi = \{\pi_h\}_{h=1}^H$ denote a policy mapping states to distributions over actions, $\pi_h : \mathcal{S} \to \triangle_{\mathcal{A}}$. When $\pi$ is deterministic, we let $\pi_h(s)$ denote the action policy $\pi$ takes at $(s,h)$. An episode begins at state $s_1$. The agent takes action $a_1 \sim \pi_1(s_1)$, transitions to state $s_2 \sim P_1(\cdot|s_1,a_1)$, and receives reward $r_1(s_1,a_1) \sim \nu_1(s_1,a_1)$. In $s_2$, the agent chooses a new action $a_2 \sim \pi_2(s_2)$, and the process repeats. After $H$ steps, the episode terminates, and the agent restarts at $s_1$.

In general, we are interested in learning policies that collect a large amount of reward. We can quantify the performance of a policy in terms of the *value function*. In particular, the $Q$-value function, $Q_h^\pi(s,a)$, denotes the expected reward that will be obtained if we are in state $s$ at step $h$, play action $a$, and then play policy $\pi$ for the remainder of the episode. Formally, $Q_h^\pi(s,a) := \mathbb{E}_\pi[\sum_{h'=h}^H r_{h'}(s_{h'}, a_{h'})|s_h = s, a_h = a]$. The value function is similarly defined as $V_h^\pi(s) := \mathbb{E}_\pi[\sum_{h'=h}^H r_{h'}(s_{h'}, a_{h'})|s_h = s]$. For deterministic policies, $V_h^\pi(s) = Q_h^\pi(s, \pi_h(s))$. We denote the optimal $Q$-value function by $Q_h^\star(s,a) = \sup_\pi Q_h^\pi(s,a)$ and the optimal value function by $V_h^\star(s) = \sup_\pi V_h^\pi(s)$, where the suprema is taken over all policies, both deterministic and stochastic. We define the *value of a policy* as $V_0^\pi = V_1^\pi(s_1)$—the expected reward policy $\pi$ achieves over an entire episode—and say a policy $\pi$ is optimal if $V_0^\pi = V_0^\star$. For some set of policies $\Pi$ (which may not contain an optimal policy), we let $V_0^\star(\Pi) := \sup_{\pi \in \Pi} V_0^\pi$.

**PAC Reinforcement Learning.**   In PAC RL, the goal is to identify some policy $\widehat{\pi}$ using as few episodes as possible, such that, with probability at least $1 - \delta$,

$$V_0^\star - V_0^{\widehat{\pi}} \le \epsilon.$$

We say that such a policy is $\epsilon$-optimal, and an algorithm with such a guarantee on every environment and reward function is $(\epsilon, \delta)$-PAC. We will also refer to this problem as "policy identification".

### 3.1   Linear MDPs

In this work, we are interested in the setting where the state space could be infinite, and the learner must generalize across states. In particular, we consider the linear MDP model defined as follows.

**Definition 3.1** (Linear MDPs (Jin et al., 2020)). We say that an MDP is a *d-dimensional linear MDP*, if there exists some (known) feature map $\phi(s,a) : \mathcal{S} \times \mathcal{A} \to \mathbb{R}^d$, $H$ (unknown) signed vector-valued measures $\boldsymbol{\mu}_h \in \mathbb{R}^d$ over $\mathcal{S}$, and $H$ (unknown) reward vectors $\boldsymbol{\theta}_h \in \mathbb{R}^d$, such that:

$$P_h(\cdot|s,a) = \langle \phi(s,a), \boldsymbol{\mu}_h(\cdot) \rangle, \quad \mathbb{E}[\nu_h(s,a)] = \langle \phi(s,a), \boldsymbol{\theta}_h \rangle.$$

We will assume $\|\phi(s,a)\|_2 \le 1$ for all $s,a$; and for all $h$, $\||\boldsymbol{\mu}_h|(\mathcal{S})\|_2 = \|\int_{s \in \mathcal{S}} |\mathrm{d}\boldsymbol{\mu}_h(s)|\|_2 \le \sqrt{d}$ and $\|\boldsymbol{\theta}_h\|_2 \le \sqrt{d}$.

Linear MDPs encompass, for example, tabular MDPs, but can also model more complex settings, such as feature spaces corresponding to the simplex (Jin et al., 2020), or the linear bandit problem. Critically, linear MDPs allow for infinite state-spaces, as well as generalization across states—rather than learning the behavior in particular states, we can learn in the $d$-dimensional ambient space. Note that the standard definition of linear MDPs, for example as given in Jin et al. (2020), assumes the rewards are deterministic, while we assume the rewards are random but that their means are linear. We still assume, however, that the random rewards, $r_h(s,a)$, are contained in $[0,1]$ almost surely.

For a given policy $\pi$, we define the *feature-visitation* at step $h$, the expected feature vector policy $\pi$ encounters at step $h$, as $\phi_{\pi,h} := \mathbb{E}_\pi[\phi(s_h, a_h)]$. Note that this is a direct generalization of state-visitations in tabular RL—if our MDP is in fact tabular, $[\phi_{\pi,h}]_{s,a} = \mathbb{P}_\pi[s_h = s, a_h = a]$, so the feature visitation vector corresponds directly to the state visitations. Note also that we can write the value of a policy as $V_0^\pi = \sum_{h=1}^H \langle \phi_{\pi,h}, \boldsymbol{\theta}_h \rangle$. Denote the average feature vector induced by $\pi$ in a particular state $s$ as $\phi_{\pi,h}(s) = \mathbb{E}_{a \sim \pi_h(\cdot|s)}[\phi(s,a)]$. We also define $\boldsymbol{\Lambda}_{\pi,h} := \mathbb{E}_\pi[\phi(s_h, a_h)\phi(s_h, a_h)^\top]$, the *expected covariance* of policy $\pi$ at step $h$, $\lambda_{\min,h}^\star = \sup_\pi \lambda_{\min}(\boldsymbol{\Lambda}_{\pi,h})$ the largest achievable minimum eigenvalue at step $h$, and $\lambda_{\min}^\star = \min_h \lambda_{\min,h}^\star$. We will make the following assumption.

**Assumption 1** (Full Rank Covariates). *In our MDP, $\lambda_{\min}^\star > 0$.*

We remark that Assumption 1 is analogous to other explorability assumptions found in the RL with function approximation literature (Zanette et al., 2020c; Hao et al., 2021; Agarwal et al., 2021).

To reduce uncertainty in directions of interest, we will be interested in optimizing over the set of all realizable covariance matrices on our particular MDP. To this end, define $\boldsymbol{\Omega}_h := \{\mathbb{E}_{\pi \sim \omega}[\boldsymbol{\Lambda}_{\pi,h}] : \omega \in \boldsymbol{\Omega}_\pi\}$ for $\boldsymbol{\Omega}_\pi$ the set of all valid distributions over Markovian policies (both deterministic and stochastic). We can think of $\boldsymbol{\Omega}_h$, then, as the set of covariance matrices realizable by distributions over policies at step $h$.

## 4 Near-Optimal Policy Identification in Linear MDPs

We are now ready to state our algorithm, PEDEL.

**PEDEL Description.** PEDEL is a *policy-elimination*-style algorithm. It takes as input some set of policies, $\Pi$, and proceeds in epochs, maintaining a set of *active* policies, $\Pi_\ell$, such that all $\pi \in \Pi_\ell$ are guaranteed to satisfy $V_0^\pi \ge V_0^\star(\Pi) - 4\epsilon_\ell$, for $\epsilon_\ell = 2^{-\ell}$. After running for $\lceil \log \frac{4}{\epsilon} \rceil$ epochs, it returns any of the remaining active policies, which will be guaranteed to have value at least $V_0^\star(\Pi) - \epsilon$.

In order to ensure $\Pi_\ell$ only contains $4\epsilon_\ell$-optimal policies, sufficient exploration must be performed at every epoch to refine the estimate of each policy's value. While works such as Wagenmaker et al. (2022) have demonstrated how to efficiently traverse a linear MDP and collect the necessary observations, existing exploration procedures are unable to obtain the instance-dependent complexity we desire. To overcome this, PEDEL relies on a novel online experiment design procedure to ensure exploration is focused only on the directions necessary to evaluate the current set of active policies.

In particular, one can show that, if we have collected some covariates $\boldsymbol{\Lambda}_{h,\ell}$, the uncertainty in our estimate of the value of policy $\pi$ at step $h$ scales as $\|\widehat{\phi}_{\pi,h}^\ell\|_{\boldsymbol{\Lambda}_{h,\ell}^{-1}}$, for $\widehat{\phi}_{\pi,h}^\ell$ the estimated feature-visitation for policy $\pi$ at epoch $\ell$. To reduce our uncertainty at each round, we would therefore like to collect covariates such that $\|\widehat{\phi}_{\pi,h}^\ell\|_{\boldsymbol{\Lambda}_{h,\ell}^{-1}} \lesssim \epsilon_\ell$. Collecting covariates which satisfy this using the minimum number of episodes of exploration possible involves solving the experiment design:

$$\inf_{\boldsymbol{\Lambda}_{\exp} \in \boldsymbol{\Omega}_h} \max_{\pi \in \Pi_\ell} \|\widehat{\phi}_{\pi,h}^\ell\|_{\boldsymbol{\Lambda}_{\exp}^{-1}}^2. \tag{4.1}$$

---

[1]Note that $K_{h,\ell}$ is random and is the number of episodes Algorithm 2 runs for.

---

**Algorithm 1** **P**olicy **L**earning via **E**xperiment **De**sign in **L**inear MDPs (PEDEL)

1: **input:** tolerance $\epsilon$, confidence $\delta$, policy set $\Pi$
2: $\ell_0 \leftarrow \lceil \log_2 \frac{d^{3/2}}{H} \rceil$, $\Pi_{\ell_0} \leftarrow \Pi$, $\widehat{\phi}_{\pi,1}^1 \leftarrow \mathbb{E}_{a \sim \pi_1(\cdot|s_1)}[\phi(s_1,a)], \forall \pi \in \Pi$
3: **for** $\ell = \ell_0, \ell_0 + 1, \ldots, \lceil \log \frac{4}{\epsilon} \rceil$ **do**
4:     $\epsilon_\ell \leftarrow 2^{-\ell}$, $\beta_\ell \leftarrow 64H^4 \log \frac{4H^2|\Pi_\ell|\ell^2}{\delta}$
5:     **for** $h = 1, 2, \ldots, H$ **do**
6:         Solve (4.1) by running Algorithm 2, collect data[1] $\{(\phi_{h,\tau}, r_{h,\tau}, s_{h+1,\tau})\}_{\tau=1}^{K_{h,\ell}}$ such that:

$$\max_{\pi \in \Pi_\ell} \|\widehat{\phi}_{\pi,h}^\ell\|_{\Lambda_{h,\ell}^{-1}}^2 \le \epsilon_\ell^2 / \beta_\ell \quad \text{for} \quad \Lambda_{h,\ell} \leftarrow \sum_{\tau=1}^{K_{h,\ell}} \phi_{h,\tau} \phi_{h,\tau}^\top + 1/d \cdot I$$

7:         **for** $\pi \in \Pi_\ell$ **do**       // Estimate feature-visitations for active policies
8:         $\widehat{\phi}_{\pi,h+1}^\ell \leftarrow \left( \sum_{\tau=1}^{K_{h,\ell}} \phi_{\pi,h+1}(s_{h+1,\tau}) \phi_{h,\tau}^\top \Lambda_{h,\ell}^{-1} \right) \widehat{\phi}_{\pi,h}^\ell$
9:     $\widehat{\theta}_h^\ell \leftarrow \Lambda_{h,\ell}^{-1} \sum_{\tau=1}^{K_{h,\ell}} \phi_{h,\tau} r_{h,\tau}$            // Estimate reward vectors
10:     // Remove provably suboptimal policies from active policy set

$$\Pi_{\ell+1} \leftarrow \Pi_\ell \backslash \left\{ \pi \in \Pi_\ell : \widehat{V}_0^\pi < \sup_{\pi' \in \Pi_\ell} \widehat{V}_0^{\pi'} - 2\epsilon_\ell \right\} \quad \text{for} \quad \widehat{V}_0^\pi := \sum_{h=1}^H \langle \widehat{\phi}_{\pi,h}^\ell, \widehat{\theta}_h^\ell \rangle$$

11:     **if** $|\Pi_{\ell+1}| = 1$ **then return** $\pi \in \Pi_{\ell+1}$
12: **return** any $\pi \in \Pi_{\ell+1}$

---

Note that this design has the form of an XY-experiment design (Soare et al., 2014). Solving (4.1) will produce covariance $\Lambda_{\exp}$ which reduces uncertainty in relevant feature directions. However, to solve this design we require knowledge of which covariance matrices are realizable on our particular MDP. In general we do not know the MDP's dynamics, and therefore do not have access to this knowledge. To overcome this and solve (4.1), in Section 5 we provide an algorithm, Algorithm 2, that is able to solve (4.1) in an online manner without knowledge of the MDP dynamics by running a low-regret algorithm on a carefully chosen reward function.

**Estimating Feature-Visitations.** We remark briefly on the estimation of the feature-visitations on Line 8. If we assume that $\{\phi_{h,\tau}\}_{\tau=1}^{K_{h,\ell}}$ is fixed and that all randomness is due to $s_{h+1,\tau}$, then it is easy to see that, using the structure present in linear MDPs as given in Definition 3.1,

$$\mathbb{E}[\sum_{\tau=1}^{K_{h,\ell}} \phi_{\pi,h+1}(s_{h+1,\tau}) \phi_{h,\tau}^\top \Lambda_{h,\ell}^{-1}] = \sum_{\tau=1}^{K_{h,\ell}} \left( \int \phi_{\pi,h+1}(s) d\mu_h(s)^\top \phi_{h,\tau} \right) \phi_{h,\tau}^\top \Lambda_{h,\ell}^{-1}$$
$$= \int \phi_{\pi,h+1}(s) d\mu_h(s)^\top.$$

By Definition 3.1, we have $\phi_{\pi,h+1} = \left( \int \phi_{\pi,h+1}(s) d\mu_h(s)^\top \right) \phi_{\pi,h}$. Comparing these, we see that our estimator of $\phi_{\pi,h+1}$ on Line 8 is (conditioned on $\{\phi_{h,\tau}\}_{\tau=1}^{K_{h,\ell}}$) unbiased, assuming $\widehat{\phi}_{\pi,h}^\ell \approx \phi_{\pi,h}$.

## 4.1 Main Results

We have the following result on the performance of PEDEL.

**Theorem 1.** *Consider running* PEDEL *with some set of Markovian policies* $\Pi$ *on any linear MDP satisfying Definition 3.1 and Assumption 1. Then with probability at least* $1 - \delta$, PEDEL *outputs a policy* $\widehat{\pi} \in \Pi$ *such that* $V_0^{\widehat{\pi}} \ge V_0^\star(\Pi) - \epsilon$, *and runs for at most*

$$C_0 H^4 \cdot \sum_{h=1}^H \inf_{\Lambda_{\exp} \in \Omega_h} \max_{\pi \in \Pi} \frac{\|\phi_{\pi,h}\|_{\Lambda_{\exp}^{-1}}^2}{\max\{V_0^\star(\Pi) - V_0^\pi, \Delta_{\min}^\Pi, \epsilon\}^2} \cdot \left( \log |\Pi| + \log \frac{1}{\delta} \right) + C_1$$

*episodes, with* $C_0 = \log \frac{1}{\epsilon} \cdot \text{poly} \log(H, \log \frac{1}{\epsilon})$, $C_1 = \text{poly}\left(d, H, \frac{1}{\lambda_{\min}^\star}, \log \frac{1}{\delta}, \log \frac{1}{\epsilon}, \log |\Pi|\right)$, $\Delta_{\min}^\Pi := V_0^\star(\Pi) - \max_{\pi \in \Pi : V_0^\pi < V_0^\star(\Pi)} V_0^\pi$, *and* $\Omega_h$ *the set of covariance matrices realizable on our MDP.*

The proof of Theorem 1 is given in Appendix B. Theorem 1 quantifies, in a precise instance-dependent way, the complexity of identifying a policy $\widehat{\pi}$ with value at most a factor of $\epsilon$ from the

value of the optimal policy in $\Pi$. At a high level, the complexity measure can be thought of as quantifying the difficulty of exploring the MDP so as to eliminate suboptimal policies. $\|\phi_{\pi,h}\|_{\Lambda_{\exp}^{-1}}$ quantifies the difficulty of collecting data necessary to eliminate policy $\pi$ (in particular, $\phi_{\pi,h}$ is the direction we need to explore to learn about the performance of policy $\pi$, and $\Lambda_{\exp}^{-1}$ quantifies how easily we can reach directions in the MDP to do this), while $V_0^\star(\Pi) - V_0^\pi$ quantifies how suboptimal policy $\pi$ is, and therefore how many samples are needed to distinguish it from the optimal policy in the class. Thus, rather than scaling only with factors such as $d$ and $\epsilon$, our complexity scales with instance-dependent quantities—the covariance matrices we can obtain and the feature vectors we expect to observe on our particular MDP, and the policy gaps on our MDP. Our complexity measure has a close resemblance to the complexity measure for best-arm identification in linear bandits found in Fiez et al. (2019), but generalizes it to problems with long horizon where navigation is required.

Theorem 1 holds for an arbitrary set of policies, yet, in general, we are interested in learning a policy which has value within a factor of $\epsilon$ of the value of the *optimal* policy on the MDP, $V_0^\star$. Such a guarantee is immediately attainable by applying Theorem 1 with a policy set $\Pi$ such that $\sup_{\pi \in \Pi} V_0^\pi \geq V_0^\star - \epsilon$. The following result shows that it is possible to construct such a set of policies, and therefore learn a *globally* near-optimal policy.

**Corollary 1.** *There exists a set of policies $\Pi_\epsilon$ such that $\log |\Pi_\epsilon| \leq \widetilde{\mathcal{O}}(dH^2 \cdot \log 1/\epsilon)$ and, for any linear MDP satisfying Definition 3.1, $\sup_{\pi \in \Pi_\epsilon} V_0^\pi \geq V_0^\star - \epsilon$. If we run PEDEL with $\Pi \leftarrow \Pi_\epsilon$, then with probability at least $1 - \delta$, it returns a policy $\widehat{\pi}$ such that $V_0^{\widehat{\pi}} \geq V_0^\star - 2\epsilon$, and runs for at most*

$$C_0 H^4 \cdot \sum_{h=1}^{H} \inf_{\Lambda_{\exp} \in \Omega_h} \max_{\pi \in \Pi_\epsilon} \frac{\|\phi_{\pi,h}\|_{\Lambda_{\exp}^{-1}}^2}{\max\{V_0^\star - V_0^\pi, \epsilon\}^2} \cdot \left(dH^2 + \log \frac{1}{\delta}\right) + C_1$$

*episodes, for $C_0 = \text{poly} \log(d, H, \frac{1}{\epsilon})$.*

Note that the policy set constructed in Corollary 1, $\Pi_\epsilon$, is guaranteed to contain an $\epsilon$-optimal policy for *any* linear MDP. Thus, this result states that without any prior knowledge of our MDP, PEDEL can be applied to find an $\epsilon$-optimal policy. While Theorem 1 and Corollary 1 quantify the instance-dependent complexity of learning, it is natural to ask what the *worst-case* complexity of PEDEL is. The following result provides such a bound.

**Corollary 2.** *For any linear MDP satisfying Definition 3.1, $\inf_{\Lambda_{\exp} \in \Omega_h} \max_{\pi \in \Pi_\epsilon} \|\phi_{\pi,h}\|_{\Lambda_{\exp}^{-1}}^2 \leq d$, so the sample complexity of Algorithm 1 when run with $\Pi \leftarrow \Pi_\epsilon$ is no larger than*

$$\widetilde{\mathcal{O}} \left( \frac{dH^5(dH^2 + \log 1/\delta)}{\epsilon^2} + C_1 \right).$$

Corollary 2 shows that PEDEL has worst-case optimal dimension dependence, matching the lower bound of $\Omega(d^2 H^2/\epsilon^2)$ given in Wagenmaker et al. (2022), up to $H$ and log factors[2].

**Remark 4.1** (Performance on Linear Contextual Bandits). Corollary 1 applies directly to linear contextual bandits by setting $H = 1$[3]. To our knowledge, this is the first instance-dependent result on PAC policy identification in linear contextual bandits. Furthermore, Corollary 2 shows that we also obtain a worst-case complexity of $\widetilde{\mathcal{O}}(d^2/\epsilon^2)$ on linear contextual bandits, which is the optimal rate (Wagenmaker et al., 2022).

## 4.2 Low-Regret Algorithms are Suboptimal for PAC RL in Large State-Spaces

We next show that there are problems on which the instance-dependent complexity of PEDEL improves on the worst-case lower bound shown in Wagenmaker et al. (2022), thereby demonstrating that we do indeed obtain favorable complexities on "easy" instances.

**Proposition 2.** *For any $d > 2$, there exists a $d$-dimensional linear MDP with $H = 2$ such that with probability $1 - \delta$, PEDEL identifies an $\epsilon$-optimal policy on this MDP after running for only $\widetilde{\mathcal{O}}\big(\frac{\log d/\delta}{\epsilon^2} + \text{poly}(d, \log \frac{1}{\delta}, \log \frac{1}{\epsilon})\big)$ episodes.*

The complexity given in Proposition 2 is a factor of $d^2$ better than the worst-case lower bound of $\Omega(d^2/\epsilon^2)$. While this shows that PEDEL yields a significant improvement over existing worst-case

---

[2]We remark that the focus of this work is on instance-dependence and dimension-dependence, not in optimizing $H$ factors, and we leave improving our $H$ dependence for future work.

[3]We describe the exact mapping to linear contextual bandits in Appendix B.3.

lower bounds on favorable instances, it is natural to ask whether the same complexity is attainable with existing algorithms, perhaps by applying a tighter analysis. Towards answering this, we will consider a class of low-regret algorithms and an online-to-batch learning protocol.

**Definition 4.1** (Low-Regret Algorithm). We say that an algorithm is a *low-regret algorithm* if its expected regret is bounded as, for all $K$:

$$\mathbb{E}[\mathcal{R}_K] = \sum_{k=1}^{K}\mathbb{E}[V_0^{\star} - V_0^{\pi_k}] \leq \mathcal{C}_1 K^{\alpha} + \mathcal{C}_2$$

for some constants $\mathcal{C}_1, \mathcal{C}_2$, and $\alpha \in (0,1)$.

**Protocol 4.1** (Online-to-Batch Learning). The online-to-batch protocol proceeds as follows:

1. The learner plays a low-regret algorithm satisfying Definition 4.1 for $K$ episodes.
2. The learner stops at a (possibly random) time $K$, and, using the observations it has collected in any way it wishes, outputs a policy $\widehat{\pi}$ it believes is $\epsilon$-optimal.

In general, by applying online-to-batch learning, one can convert a regret guarantee of $\mathcal{C}_1 K^{\alpha} + \mathcal{C}_2$ to a PAC complexity of $\mathcal{O}((\frac{\mathcal{C}_1}{\epsilon})^{\frac{1}{1-\alpha}} + \frac{\mathcal{C}_2}{\epsilon})$ (Jin et al., 2018), allowing low-regret algorithms such as that of Zanette et al. (2020b) to obtain the minimax-optimal PAC complexity of $\mathcal{O}(d^2 H^4/\epsilon^2)$. The following result shows, however, that this protocol is unable to obtain the instance-optimal rate.

**Proposition 3.** *On the instance of Proposition 2, for small enough $\epsilon$, any learner that is $(\epsilon, \delta)$-PAC and follows Protocol 4.1 with stopping time $K$ must have $\mathbb{E}[K] \geq \Omega\big(\frac{d \cdot \log 1/\delta}{\epsilon^2}\big)$.*

Together, Proposition 2 and Proposition 3 show that running a low-regret algorithm to learn a near-optimal policy in a linear MDP is provably suboptimal—at least a factor of $d$ worse than the instance-dependent rate obtained by PEDEL. While a similar observation was recently made in the setting of tabular MDPs (Wagenmaker et al., 2021b), to our knowledge, this is the first such result in the RL with function approximation setting, implying that, in this setting, low-regret algorithms are insufficient for obtaining optimal PAC sample complexity. As standard optimistic algorithms are also low-regret, this result implies that all such optimistic algorithms are also suboptimal.

### 4.3 Tabular and Deterministic MDPs

To relate our results to existing results on instance-dependent RL, we next turn to the setting of tabular MDPs, where it is assumed that $S := |\mathcal{S}| < \infty, A := |\mathcal{A}| < \infty$. Define:

$$\Delta_h(s,a) = V_h^{\star}(s) - Q_h^{\star}(s,a), \quad w_h^{\pi}(s,a) = \mathbb{P}_{\pi}[s_h = s, a_h = a].$$

$\Delta_h(s,a)$ denotes the *value-function gap*, and quantifies the suboptimality of playing action $a$ in state $s$ at step $h$ and then playing the optimal policy, as compared to taking the optimal action in $(s,h)$. $w_h^{\pi}(s,a)$ denotes the state-action visitation distribution for policy $\pi$, and quantifies how likely policy $\pi$ is to reach $(s,a)$ at step $h$. Note that $[\phi_{\pi,h}]_{(s,a)} = w_h^{\pi}(s,a)$. We obtain the following corollary.

**Corollary 3.** *In the setting of tabular MDPs, PEDEL outputs an $\epsilon$-optimal policy with probability at least $1 - \delta$, and has sample complexity bounded as*

$$\widetilde{\mathcal{O}}\Big(\sum_{h=1}^{H} \inf_{\pi_{\exp}} \max_{\pi \in \Pi} \max_{s,a} \frac{H^4}{w_h^{\pi_{\exp}}(s,a)} \min\Big\{\frac{1}{w_h^{\pi}(s,a)\Delta_h(s,a)^2}, \frac{w_h^{\pi}(s,a)}{\Delta_{\min}(\Pi)^2}, \frac{w_h^{\pi}(s,a)}{\epsilon^2}\Big\}(SH + \log\frac{1}{\delta}) + C_1\Big),$$

*for $C_1 = \mathrm{poly}\big(S, A, H, \frac{1}{\min_h \min_s \sup_{\pi} w_h^{\pi}(s)}, \log\frac{1}{\delta}, \log\frac{1}{\epsilon}\big)$ and $\Pi$ the set of all deterministic policies.*

For tabular MDPs, the primary comparable result on instance-dependent policy identification is that obtained by Wagenmaker et al. (2021b), which introduces a different measure of complexity, the *gap-visitation complexity*, and an algorithm, MOCA, with sample complexity scaling as the gap-visitation complexity. The following result shows that the complexity PEDEL obtains on tabular MDPs and the gap-visitation complexity do not have a clear ordering.

**Proposition 4.** *Fix any $\epsilon \in (0, 1/2)$ and $S \geq \log_2(1/\epsilon)$. Then there exist tabular MDPs $\mathcal{M}_1$ and $\mathcal{M}_2$, each with $H = 2$, $S$ states, and $\mathcal{O}(S)$ actions, such that:*

- *On $\mathcal{M}_1$, the complexity bound of PEDEL given in Corollary 3 scales as $\mathrm{poly}(S, \log 1/\delta)$, while the gap-visitation complexity scales as $\Omega(1/\epsilon^2)$.*
- *On $\mathcal{M}_2$, the complexity bound of PEDEL given in Corollary 3 scales as $\Omega(1/\epsilon^2)$, while the gap-visitation complexity scales as $\mathrm{poly}(S, \log 1/\delta)$.*

The lack of ordering between the two complexity measures arises because, on some problem instances, it is easier to learn in policy-space (as PEDEL does), while on other instances, it is easier to learn near-optimal actions on individual states directly, and then synthesize these actions into a near-optimal policy (the approach MOCA takes). This difference arises because, in the former instance, the minimum *policy gap* is large ($V_0^\star - V_0^\pi = \Omega(1)$ for every deterministic policy $\pi \neq \pi^\star$), while in the latter instance, the minimum policy gap is small, but all value-function gaps are large, satisfying $\Delta_h(s,a) = \Omega(1)$ for all $a \neq \arg\max_{a\in\mathcal{A}} Q_h^\star(s,a)$ and all $s$ and $h$. Thus, on the former instance, it is much easier to learn over the space of policies, while on the latter it is much easier to learn optimal actions in individual states. Resolving this discrepency with an algorithm able to achieve the "best-of-both-worlds" is an interesting direction for future work.

**Deterministic MDPs.** Finally, we turn to the simplified setting of tabular, deterministic MDPs. Here, for each $(s,a,h)$, there exists some $s'$ such that $P_h(s'|s,a) = 1$. We still allow the rewards to be random, however, so the agent must still learn in order to find a near-optimal policy. Following the same notation as the recent work of Tirinzoni et al. (2022), let $\Pi_{sah} = \{\pi \text{ deterministic} : s_h^\pi = s, a_h^\pi = a\}$, where $s_h^\pi$ and $a_h^\pi$ are the state and action policy $\pi$ will be in at step $h$ (note that these quantities are well-defined quantities for deterministic policies). Also define the *deterministic return gap* as $\bar{\Delta}_h(s,a) := V_0^\star - \max_{\pi\in\Pi_{sah}} V_0^\pi$, and let $\bar{\Delta}_{\min} := \min_{s,a,h:\bar{\Delta}_h(s,a)>0} \bar{\Delta}_h(s,a)$ in the case when there exists a unique optimal deterministic policy, and $\bar{\Delta}_{\min} := 0$ otherwise. We obtain the following.

**Corollary 4.** *In the setting of tabular, deterministic MDPs,* PEDEL *outputs an $\epsilon$-optimal policy with probability at least $1 - \delta$, and has sample complexity bounded as*

$$\widetilde{\mathcal{O}}\left( H^4 \cdot \sum_{h=1}^{H} \sum_{s,a} \frac{1}{\max\{\bar{\Delta}_h(s,a), \bar{\Delta}_{\min}, \epsilon\}^2} \cdot (H + \log\tfrac{1}{\delta}) + \text{poly}\left(S, A, H, \log\tfrac{1}{\delta}, \log\tfrac{1}{\epsilon}\right) \right).$$

Up to $H$ and log factors and lower-order terms, the rate given in Corollary 4 matches the instance-dependent lower bound given in Tirinzoni et al. (2022)[4]. Thus, we conclude that, in the setting of tabular, deterministic MDPs, PEDEL is (nearly) instance-optimal. While Tirinzoni et al. (2022) also obtain instance-optimality in this setting, their algorithm and analysis are specialized to tabular, deterministic MDPs—in contrast, PEDEL requires no modification from its standard operation.

## 5   Online Experiment Design in Linear MDPs

As described in Section 4, to reduce our uncertainty and explore in a way that only targets the relevant feature directions, we must solve an XY-experiment design problem of the form:

$$\inf_{\mathbf{\Lambda}_{\exp}\in\mathbf{\Omega}_h} \max_{\phi\in\Phi} \|\phi\|_{\mathbf{\Lambda}_{\exp}^{-1}}^2. \tag{5.1}$$

This is not, in general, possible to solve without knowledge of the MDP dynamics. In this section we describe our approach to solving (5.1) without knowledge of the MDP dynamics by relying on a low-regret algorithm as an optimization primitive.

**Approximating Frank-Wolfe via Regret Minimization.** Given knowledge of the MDP dynamics, we could compute $\mathbf{\Omega}_h$ directly, and apply the celebrated Frank-Wolfe coordinate-descent algorithm (Frank & Wolfe, 1956) to solve (5.1). In this setting the Frank-Wolfe update for (5.1) is:

$$\mathbf{\Gamma}_t = \arg\min_{\mathbf{\Gamma}\in\mathbf{\Omega}_h} \langle \nabla_{\mathbf{\Lambda}}(\max_{\phi\in\Phi}\|\phi\|_{\mathbf{\Lambda}^{-1}}^2)|_{\mathbf{\Lambda}=\mathbf{\Lambda}_t}, \mathbf{\Gamma}\rangle, \quad \mathbf{\Lambda}_{t+1} = (1-\gamma_t)\mathbf{\Lambda}_t + \gamma_t\mathbf{\Gamma}_t \tag{5.2}$$

for step size $\gamma_t$. Standard Frank-Wolfe analysis shows that this update converges to a near-optimal solution to (5.1) at a polynomial rate. However, without knowledge of $\mathbf{\Omega}_h$, we are unable to solve for $\mathbf{\Gamma}_t$ and run the Frank-Wolfe update.

Our critical observation is that the minimization over $\mathbf{\Omega}_h$ in (5.2) can be approximated without knowledge of $\mathbf{\Omega}_h$ by running a low-regret algorithm on a particular objective. Some calculation shows that (except on a measure-zero set, assuming $\Phi$ is finite) $\nabla_{\mathbf{\Lambda}}(\max_{\phi\in\Phi}\|\phi\|_{\mathbf{\Lambda}^{-1}}^2)|_{\mathbf{\Lambda}=\mathbf{\Lambda}_t} = -\mathbf{\Lambda}_t^{-1}\widetilde{\phi}_t\widetilde{\phi}_t^\top\mathbf{\Lambda}_t^{-1}$ for $\widetilde{\phi}_t = \arg\max_{\phi\in\Phi}\|\phi\|_{\mathbf{\Lambda}_t^{-1}}^2$. If $\mathbf{\Gamma} = \mathbf{\Lambda}_{\pi,h} = \mathbb{E}_\pi[\phi_h\phi_h^\top]$ for some $\pi$, we have

$$\langle\nabla_{\mathbf{\Lambda}}(\max_{\phi\in\Phi}\|\phi\|_{\mathbf{\Lambda}^{-1}}^2)|_{\mathbf{\Lambda}=\mathbf{\Lambda}_t}, \mathbf{\Gamma}\rangle = -\text{tr}(\mathbf{\Lambda}_t^{-1}\widetilde{\phi}_t\widetilde{\phi}_t^\top\mathbf{\Lambda}_t^{-1}\mathbf{\Lambda}_{\pi,h}) = -\mathbb{E}_\pi[(\phi_h^\top\mathbf{\Lambda}_t^{-1}\widetilde{\phi}_t)^2].$$

---

[4]The lower bound of Tirinzoni et al. (2022) depends on a slightly different (but nearly equivalent) minimum gap term, $\bar{\Delta}_{\min}^h$. Similar to our upper bound, the upper bound of Tirinzoni et al. (2022) scales with $\bar{\Delta}_{\min}$ instead of $\bar{\Delta}_{\min}^h$. We offer a more in-depth discussion of this point in Appendix B.3.

Now, if we run a low-regret algorithm on the (deterministic) reward $\nu_h^t(s,a) = (\phi(s,a)^\top \Lambda_t^{-1} \widetilde{\phi}_t)^2$ for a sufficiently large number of episodes $K$, we will be guaranteed to collect reward at a rate close to that of the optimal policy, which implies we will collect some data $\{\phi_{h,\tau}\}_{\tau=1}^K$ such that

$$K^{-1} \cdot \widetilde{\phi}_t^\top \widehat{\Gamma}_K \widetilde{\phi}_t := K^{-1} \sum_{\tau=1}^K (\phi_{h,\tau}^\top \Lambda_t^{-1} \widetilde{\phi}_t)^2 \approx \sup_\pi \mathbb{E}_\pi[(\phi_h^\top \Lambda_t^{-1} \widetilde{\phi}_t)^2]. \tag{5.3}$$

However, this implies the covariates we have collected, $\widehat{\Gamma}_K$, approximately minimize (5.2). In other words, running a low-regret algorithm on $\nu_h^t$ allows us to obtain covariates which approximate the Frank-Wolfe update—without knowledge of $\Omega_h$, we can solve the Frank-Wolfe update by running a low-regret algorithm, and therefore solve (5.1). This motivates Algorithm 2.

---

**Algorithm 2** Online Frank-Wolfe via Regret Minimization (informal)

---

1: **input:** uncertain feature directions $\Phi$, step $h$, regularization $\Lambda_0 \succ 0$
2: $K_0 \leftarrow$ sufficiently large number of episodes to guarantee (5.3) holds
3: Run any policy for $K_0$ episodes, collect data $\{\phi_{h,\tau}\}_{\tau=1}^{K_0}$, set $\Lambda_1 \leftarrow K_0^{-1} \sum_{\tau=1}^{K_0} \phi_{h,\tau} \phi_{h,\tau}^\top$
4: **for** $t = 1, \ldots, T-1$ **do**
5:    $\widetilde{\phi}_t \leftarrow \arg\max_{\phi \in \Phi} \|\phi\|_{(\Lambda_t + \Lambda_0)^{-1}}^2$, $\nu_h^t(s,a) \leftarrow (\phi(s,a)^\top (\Lambda_t + \Lambda_0)^{-1} \widetilde{\phi}_t)^2$
6:    Run low-regret algorithm on $\nu_h^t$ for $K_0$ episodes, collect covariates $\widehat{\Gamma}_{K_0}^t$
7:    Set $\Lambda_{t+1} \leftarrow (1 - \gamma_t) \Lambda_t + \gamma_t K_0^{-1} \widehat{\Gamma}_{K_0}^t$ for $\gamma_t = \frac{1}{t+1}$
8: **return:** covariates $T K_0 \Lambda_T = \sum_{t=1}^{T-1} \widehat{\Gamma}_{K_0}^t + \Lambda_1$

---

**Theorem 5** (informal). *Consider running Algorithm 2 with some $\Lambda_0 \succ 0$. Then with properly chosen settings of $K_0$ and $T$, we can guarantee that, with probability at least $1 - \delta$, we will run for at most*

$$N \le 20 \cdot \frac{\inf_{\Lambda_{\exp} \in \Omega_h} \max_{\phi \in \Phi} \|\phi\|_{(\Lambda_{\exp} + \Lambda_0)^{-1}}^2}{\epsilon_{\exp}} + \mathrm{poly}\left(d, H, \|\Lambda_0^{-1}\|_{\mathrm{op}}, \log |\Phi|, \log 1/\delta\right)$$

*episodes, and return covariance $\widehat{\Lambda}_N$ satisfying $\max_{\phi \in \Phi} \|\phi\|_{(\widehat{\Lambda}_N + N\Lambda_0)^{-1}}^2 \le \epsilon_{\exp}$.*

Note that this rate is essentially optimal, up to constants and lower-order terms. If we let $\omega_{\exp}^\star$ denote the distribution over policies which minimize (5.1), then to collect covariance $\widehat{\Lambda}_N$ such that $\max_{\phi \in \Phi} \|\phi\|_{(\widehat{\Lambda}_N + N\Lambda_0)^{-1}}^2 \le \epsilon_{\exp}$, in expectation, we would need to play $\pi \sim \omega_{\exp}^\star$ for at least

$$\inf_{\Lambda_{\exp} \in \Omega_h} \max_{\phi \in \Phi} \|\phi\|_{(\Lambda_{\exp} + \Lambda_0)^{-1}}^2 \cdot \epsilon_{\exp}^{-1}$$

episodes, which is the same scaling as obtained in Theorem 5.

In practice, we instead run Algorithm 2 on a smoothed version of the objective in (5.1). We provide a full definition of Algorithm 2 with exact setting of $T$ and $K_0$ in Appendix C. Theorem 5 is itself a corollary of a more general result, Theorem 8, given in Appendix C, which shows our Frank-Wolfe procedure can be applied to minimize any smooth experiment-design objective—for example, collecting covariates which optimally maximize the minimum eigenvalue, E-optimal design.

## 6 Conclusion

In this work, we have shown that it is possible to obtain instance-dependent guarantees in RL with function approximation, and that our algorithm, PEDEL, yields provable gains over low-regret algorithms. As the first result of its kind in this setting, it opens several directions for future work.

The computational complexity of PEDEL scales as $\mathrm{poly}(d, H, \frac{1}{\epsilon}, |\Pi|, |\mathcal{A}|, \log \frac{1}{\delta})$. In general, to ensure $\Pi$ contains an $\epsilon$-optimal policy, $|\Pi|$ must be exponential in problem parameters, rendering PEDEL computationally inefficient. Furthermore, the sample complexity of PEDEL scales with $\lambda_{\min}^\star$, the "hardest-to-reach" direction. While this is not uncommon in the literature, we might hope that if a direction is very difficult to reach, learning in that direction should not be necessary, as we are unlikely to ever encounter it. Obtaining an algorithm with a similar instance-dependence but that is computationally efficient and does not depend on $\lambda_{\min}^\star$ is an interesting direction for future work.

Extending our results to the setting of general function approximation is also an exciting direction. While our results do rely on the linear structure of the MDP, we believe the online experiment-design approach we propose could be generally applicable in more complex settings. As a first step, it could be interesting to extend our approach to the setting of Bilinear classes (Du et al., 2021), which also exhibits a certain linear structure.

**Acknowledgements**

The work of AW was supported by an NSF GFRP Fellowship DGE-1762114. The work of KJ was funded in part by the AFRL and NSF TRIPODS 2023166.

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
