# A   Technical Results

**Lemma A.1** (Vershynin (2010))**.** *For any $\epsilon > 0$, the $\epsilon$-covering number of the Euclidean ball $\mathcal{B}^d(R) := \{x \in \mathbb{R}^d : \|x\|_2 \leq R\}$ with radius $R > 0$ in the Euclidean metric is upper bounded by $(1 + 2R/\epsilon)^d$.*

**Lemma A.2** (Lemma A.4 of Wagenmaker et al. (2022))**.** *If $x \geq C(2n)^n \log^n(2nCB)$ for $n, C, B \geq 1$, then $x \geq C \log^n(Bx)$.*

**Lemma A.3** (McSherry & Talwar (2007); Epasto et al. (2020))**.** *Consider some $(x_i)_{i=1}^n$. Then if $\eta \geq \log(n)/\delta$, we have*

$$\frac{\sum_{i=1}^n e^{\eta x_i} x_i}{\sum_{i=1}^n e^{\eta x_i}} \geq \max_{i \in [n]} x_i - \delta.$$

**Lemma A.4** (Azuma-Hoeffding)**.** *et $\mathcal{F}_0 \subset \mathcal{F}_1 \subset \ldots \subset \mathcal{F}_T$ be a filtration and let $X_1, X_2, \ldots, X_T$ be real random variables such that $X_t$ is $\mathcal{F}_t$-measurable, $\mathbb{E}[X_t|\mathcal{F}_{t-1}] = 0$, and $|X_t| \leq b$ almost surely. Then for any $\delta \in (0, 1)$, we have with probability at least $1 - \delta$,*

$$\left| \sum_{t=1}^T X_t \right| \leq \sqrt{8b^2 \log 2/\delta}.$$

**Lemma A.5** (Freedman's Inequality (Freedman, 1975))**.** *Let $\mathcal{F}_0 \subset \mathcal{F}_1 \subset \ldots \subset \mathcal{F}_T$ be a filtration and let $X_1, X_2, \ldots, X_T$ be real random variables such that $X_t$ is $\mathcal{F}_t$-measurable, $\mathbb{E}[X_t|\mathcal{F}_{t-1}] = 0$, $|X_t| \leq b$ almost surely, and $\sum_{t=1}^T \mathbb{E}[X_t^2|\mathcal{F}_{t-1}] \leq V$ for some fixed $V > 0$ and $b > 0$. Then for any $\delta \in (0, 1)$, we have with probability at least $1 - \delta$,*

$$\sum_{t=1}^T X_t \leq 2\sqrt{V \log 1/\delta} + b \log 1/\delta.$$

## A.1   Properties of Linear MDPs

**Lemma A.6.** *For any linear MDP satisfying Definition 3.1, we must have that $\|\phi(s, a)\|_2 \geq 1/\sqrt{d}$ for all $s$ and $a$, and $\|\phi_{\pi,h}\|_2 \geq 1/\sqrt{d}$ for all $\pi$ and $h$.*

*Proof.* By Definition 3.1, we know that $P_h(\cdot|s, a) = \langle \phi(s, a), \mu_h(\cdot) \rangle$ forms a valid probability distribution, and that $\| \int_{\mathcal{S}} |\mathrm{d}\mu_h(s)| \|_2 \leq \sqrt{d}$. It follows that

$$1 = \int_{\mathcal{S}} \langle \phi(s, a), \mathrm{d}\mu_h(s) \rangle \leq \|\phi(s, a)\|_2 \| \int_{\mathcal{S}} |\mathrm{d}\mu_h(s)| \|_2 \leq \sqrt{d} \|\phi(s, a)\|_2$$

from which the first result follows.

For the second result, using that $1 = \int_{\mathcal{S}} \langle \phi(s, a), \mathrm{d}\mu_h(s) \rangle$, we get

$$\int_{\mathcal{S}} \langle \phi_{\pi,h}, \mathrm{d}\mu_h(s) \rangle = \int_{\mathcal{S}} \langle \mathbb{E}_\pi[\phi_h], \mathrm{d}\mu_h(s) \rangle$$
$$= \mathbb{E}_\pi \left[ \int_{\mathcal{S}} \langle \phi_h, \mathrm{d}\mu_h(s) \rangle \right]$$
$$= \mathbb{E}_\pi[1]$$
$$= 1$$

where we can exchange the order of integration by Fubini's Theorem since the integrand is absolutely integrable, by Definition 3.1. As above, we then have

$$1 = \int_{\mathcal{S}} \langle \phi_{\pi,h}, \mathrm{d}\mu_h(s) \rangle \leq \sqrt{d} \|\phi_{\pi,h}\|_2$$

so the second result follows. $\qquad\square$

## A.2 Feature-Visitations in Linear MDPs

Define

$$\phi_{\pi,h} = \mathbb{E}_\pi[\phi(s_h, a_h)], \quad \phi_{\pi,h}(s) = \sum_{a \in \mathcal{A}} \phi(s,a)\pi_h(a|s)$$

and

$$\mathcal{T}_{\pi,h} := \int \phi_{\pi,h}(s)\mathrm{d}\mu_{h-1}(s)^\top.$$

**Lemma A.7.** $\phi_{\pi,h} = \mathcal{T}_{\pi,h}\phi_{\pi,h-1} = \ldots = \mathcal{T}_{\pi,h}\ldots\mathcal{T}_{\pi,1}\phi_{\pi,0}.$

*Proof.* By the linear MDP assumption, we have:

$$
\begin{aligned}
\phi_{\pi,h} &= \mathbb{E}_\pi[\phi(s_h, a_h)] \\
&= \mathbb{E}_\pi[\mathbb{E}[\phi(s_h, a_h)|\mathcal{F}_{h-1}]] \\
&= \mathbb{E}_\pi[\int\int \phi(s,a)\mathrm{d}\pi_h(a|s)\mathrm{d}\mu_{h-1}(s)^\top\phi(s_{h-1}, a_{h-1})] \\
&= \mathbb{E}_\pi[\int \phi_{\pi,h}(s)\mathrm{d}\mu_{h-1}(s)^\top\phi(s_{h-1}, a_{h-1})] \\
&= \int \phi_{\pi,h}(s)\mathrm{d}\mu_{h-1}(s)^\top\mathbb{E}_\pi[\phi(s_{h-1}, a_{h-1})] \\
&= \mathcal{T}_{\pi,h}\phi_{\pi,h-1}.
\end{aligned}
$$

This yields the first equality. Repeating this calculation $h-1$ more times yields the final equality. $\square$

**Lemma A.8.** *Fix some $h$ and $i < h$, and consider the vector*

$$\boldsymbol{v} := \mathcal{T}_{\pi,i+1}^\top \mathcal{T}_{\pi,i+2}^\top \ldots \mathcal{T}_{\pi,h-1}^\top \mathcal{T}_{\pi,h}^\top \boldsymbol{u}.$$

*Assume that either $\boldsymbol{u} = \boldsymbol{\theta}_h$ for some $\boldsymbol{\theta}_h$ which is a valid reward vector as defined in Definition 3.1, or $\boldsymbol{u} \in \mathcal{S}^{d-1}$. In either case, we have that, for any $s, a$, $|\boldsymbol{v}^\top\phi(s,a)| \leq 1$, and $\|\boldsymbol{v}\|_2 \leq \sqrt{d}$.*

*Proof.* By the linear MDP structure (see Proposition 2.3 of Jin et al. (2020)), for any $j$,

$$
\begin{aligned}
Q_j^\pi(s,a) &= \langle \phi(s,a), \boldsymbol{w}_j^\pi \rangle \\
&= \langle \phi(s,a), \boldsymbol{\theta}_j \rangle + \int V_{j+1}^\pi(s')\mathrm{d}\mu_j(s')^\top\phi(s,a) \\
&= \langle \phi(s,a), \boldsymbol{\theta}_j \rangle + \int \langle \boldsymbol{w}_{j+1}^\pi, \phi_{j+1,\pi}(s')\rangle\mathrm{d}\mu_j(s')^\top\phi(s,a) \\
&= \langle \phi(s,a), \boldsymbol{\theta}_j + \mathcal{T}_{\pi,j+1}^\top\boldsymbol{w}_{j+1}^\pi \rangle
\end{aligned}
$$

so in general,

$$\boldsymbol{w}_i^\pi = \sum_{h'=i}^{H}\left(\prod_{j=i+1}^{h'}\mathcal{T}_{\pi,j}^\top\right)\boldsymbol{\theta}_{h'}$$

where we order the product $\prod_{j=i+1}^{h'}\mathcal{T}_{\pi,j}^\top = \mathcal{T}_{\pi,i+1}^\top\mathcal{T}_{\pi,i+1}^\top\ldots\mathcal{T}_{\pi,h'}^\top$.

**Case 1: $\boldsymbol{u} = \boldsymbol{\theta}_h$.** We first consider the case where $\boldsymbol{u} = \boldsymbol{\theta}_h$ for some $\boldsymbol{\theta}_h$ which is a valid reward satisfying Definition 3.1. Assume that the reward in our MDP is set such that for $h' \neq h$, $\boldsymbol{\theta}_{h'} = 0$. In this case, we then have that

$$\boldsymbol{w}_i^\pi = \mathcal{T}_{\pi,i+1}^\top\mathcal{T}_{\pi,i+2}^\top\ldots\mathcal{T}_{\pi,h}^\top\boldsymbol{\theta}_h = \boldsymbol{v}.$$

In this case, we know that the trajectory rewards are always bounded by 1, so it follows that $Q_i^\pi(s,a) \leq 1$. Thus,

$$1 \geq Q_i^\pi(s,a) = \langle \phi(s,a), \boldsymbol{w}_i^\pi \rangle = \langle \phi(s,a), \boldsymbol{v} \rangle$$

and this holds for any $s, a$. Since $Q$-values are always positive, it also holds that $\langle \phi(s,a), \boldsymbol{v} \rangle \geq 0$.

To bound the norm of $\boldsymbol{v}$, we note that by the Bellman equation and the calculation above,

$$\|\boldsymbol{v}\|_2 = \|\boldsymbol{w}_i^\pi\|_2 = \|\boldsymbol{\theta}_i + \int V_{i+1}^\pi(s')\mathrm{d}\boldsymbol{\mu}_i(s')\|_2$$

$$\leq \|\boldsymbol{\theta}_i\|_2 + \|\int |V_{i+1}^\pi(s')|\mathrm{d}\boldsymbol{\mu}_i(s')\|_2$$

$$\leq \|\int |\mathrm{d}\boldsymbol{\mu}_i(s')|\|_2$$

$$\leq \sqrt{d}$$

where we have used that $|V_{i+1}^\pi(s')| \leq 1$ since the total episode return is at most 1 on our augmented reward function, and the linear MDP assumption.

**Case 2: $\boldsymbol{u} \in \mathcal{S}^{d-1}$.** We can repeat the argument above in the case where we only assume $\boldsymbol{u} \in \mathcal{S}^{d-1}$. Since $\|\boldsymbol{\phi}(s,a)\|_2 \leq 1$, it follows that with the reward vector at level $h$ set to $\boldsymbol{u}$, the reward will still be bounded in $[-1,1]$. Thus, essentially the same argument can be used, with the slight modification to handle $Q$-values that are negative. $\qquad\square$

**Lemma A.9.** *The set $\boldsymbol{\Omega}_h$ is convex and compact.*

*Proof.* Take $\boldsymbol{\Lambda}_1, \boldsymbol{\Lambda}_2 \in \boldsymbol{\Omega}_h$. By definition, $\boldsymbol{\Lambda}_1 = \mathbb{E}_{\pi \sim \omega_1}[\boldsymbol{\Lambda}_{\pi,h}], \boldsymbol{\Lambda}_2 = \mathbb{E}_{\pi \sim \omega_2}[\boldsymbol{\Lambda}_{\pi,h}]$. It follows that, for any $t \in [0,1]$, $t\boldsymbol{\Lambda}_1 + (1-t)\boldsymbol{\Lambda}_2 = \mathbb{E}_{\pi \sim t\omega_1 + (1-t)\omega_2}[\boldsymbol{\Lambda}_{\pi,h}]$. For $t\omega_1 + (1-t)\omega_2$ the mixture of $\omega_1$ and $\omega_2$. As $t\omega_1 + (1-t)\omega_2$ is a valid mixture over policies, it follows that $t\boldsymbol{\Lambda}_1 + (1-t)\boldsymbol{\Lambda}_2 \in \boldsymbol{\Omega}_h$, which proves convexity.

Compactness follows since $\|\boldsymbol{\phi}(s,a)\|_2 \leq 1$ for all $s, a$, so $\|\boldsymbol{\Lambda}_{\pi,h}\|_{\mathrm{op}} \leq 1$, which implies $\|\boldsymbol{\Lambda}\|_{\mathrm{op}} \leq 1$ for any $\boldsymbol{\Lambda} \in \boldsymbol{\Omega}_h$. Furthermore, the set $\boldsymbol{\Omega}_h$ is clearly closed, which proves compactness. $\qquad\square$

## A.3 Constructing the Policy Class

**Lemma A.10** (Lemma B.1 of Jin et al. (2020)). *Let $\boldsymbol{w}_h^\pi$ denote the set of weights such that $Q_h^\pi(s,a) = \langle \boldsymbol{\phi}(s,a), \boldsymbol{w}_h^\pi \rangle$. Then $\|\boldsymbol{w}_h^\pi\|_2 \leq 2H\sqrt{d}$.*

**Lemma A.11.** *For any $\delta > 0$ there exists sets of actions $(\widetilde{\mathcal{A}}_s)_{s \in \mathcal{S}}$, $\widetilde{\mathcal{A}}_s \subseteq \mathcal{A}$, such that $|\widetilde{\mathcal{A}}_s| \leq (1 + 8H\sqrt{d}/\delta)^d$ for all $s$ and, for all $a \in \mathcal{A}$, $s$, $h$, and any $\pi$, there exists some $\widetilde{a} \in \widetilde{\mathcal{A}}_s$ such that*

$$|Q_h^\pi(s,a) - Q_h^\pi(s,\widetilde{a})| \leq \delta, \quad |r_h(s,a) - r_h(s,\widetilde{a})| \leq \delta.$$

*Proof.* Let $\mathcal{N}$ be a $\delta/(4H\sqrt{d})$ cover of the unit ball. By Lemma A.1 we can bound $|\mathcal{N}| \leq (1 + 8H\sqrt{d}/\delta)^d$. Take any $s$ and let $\widetilde{\mathcal{A}}_s = \emptyset$. Then for each $\boldsymbol{\phi} \in \mathcal{N}$, choose any $a$ at random from the set $\{a \in \mathcal{A} : \|\boldsymbol{\phi}(s,a) - \boldsymbol{\phi}\|_2 \leq \delta/2\}$ and set $\widetilde{\mathcal{A}}_s \leftarrow \widetilde{\mathcal{A}}_s \cup \{a\}$. With this construction, we claim that for all $a \in \mathcal{A}$, there exists some $\widetilde{a} \in \widetilde{\mathcal{A}}_s$ such that $\|\boldsymbol{\phi}(s,a) - \boldsymbol{\phi}(s,\widetilde{a})\|_2 \leq \delta/(2H\sqrt{d})$. To see why this is, note that by construction of $\mathcal{N}$, there always exists some $\boldsymbol{\phi} \in \mathcal{N}$ such that $\|\boldsymbol{\phi}(s,a) - \boldsymbol{\phi}\|_2 \leq \delta/(4H\sqrt{d})$. Since $\widetilde{\mathcal{A}}_s$ will contain some $\widetilde{a}$ such that $\|\boldsymbol{\phi}(s,\widetilde{a}) - \boldsymbol{\phi}\|_2 \leq \delta/(4H\sqrt{d})$, the claim follows by the triangle inequality.

By Lemma A.10, we have that for any $\pi$, $\|\boldsymbol{w}_h^\pi\|_2 \leq 2H\sqrt{d}$. Take $a \in \mathcal{A}$ and let $\widetilde{a} \in \widetilde{\mathcal{A}}_s$ be the action such that $\|\boldsymbol{\phi}(s,a) - \boldsymbol{\phi}(s,\widetilde{a})\|_2 \leq \delta/(2H\sqrt{d})$. Then

$$|Q_h^\pi(s,a) - Q_h^\pi(s,\widetilde{a})| = |\langle \boldsymbol{\phi}(s,a) - \boldsymbol{\phi}(s,\widetilde{a}), w_h^\pi \rangle| \leq 2H\sqrt{d}\|\boldsymbol{\phi}(s,a) - \boldsymbol{\phi}(s,\widetilde{a})\|_2 \leq \delta.$$

The bound on $|r_h(s,a) - r_h(s,\widetilde{a})|$ follows analogously, since we assume our rewards are linear, and that $\|\boldsymbol{\theta}_h\|_2 \leq \sqrt{d}$. $\qquad\square$

**Definition A.1** (Linear Softmax Policy). We say a policy is a *linear softmax policy* with parameters $\eta$ and $\{\boldsymbol{w}_h\}_{h=1}^H$ if it can be written as

$$\pi_h(a|s) = \frac{e^{\eta\langle \boldsymbol{\phi}(s,a), \boldsymbol{w}_h \rangle}}{\sum_{a' \in \mathcal{A}} e^{\eta\langle \boldsymbol{\phi}(s,a'), \boldsymbol{w}_h \rangle}}$$

for some $\boldsymbol{w} = \{\boldsymbol{w}_h\}_{h=1}^H$. We will denote such a policy as $\pi^{\boldsymbol{w}}$.

**Definition A.2** (Restricted-Action Linear Softmax Policy). We say a policy is a *restricted-action linear softmax policy* with parameters $\eta$, $\{\boldsymbol{w}_h\}_{h=1}^H$, and $(\widetilde{\mathcal{A}}_s)_{s \in \mathcal{S}}$ if it can be written as

$$\widetilde{\pi}_h(a|s) = \frac{e^{\eta \langle \boldsymbol{\phi}(s,a), \boldsymbol{w}_h \rangle} \cdot \mathbb{I}\{a \in \widetilde{\mathcal{A}}_s\}}{\sum_{a' \in \widetilde{\mathcal{A}}_s} e^{\eta \langle \boldsymbol{\phi}(s,a'), \boldsymbol{w}_h \rangle}}$$

for some $\boldsymbol{w} = \{\boldsymbol{w}_h\}_{h=1}^H$. We will denote such a policy as $\widetilde{\pi}^{\boldsymbol{w}}$.

**Lemma A.12.** *For any restricted-action linear softmax policies $\pi^{\boldsymbol{w}}$ and $\pi^{\boldsymbol{u}}$ with identical restricted sets $(\widetilde{\mathcal{A}}_s)_{s \in \mathcal{S}}$, we can bound*

$$|V_0^{\pi^{\boldsymbol{w}}}(s_1) - V_0^{\pi^{\boldsymbol{u}}}(s_1)| \le 2dH\eta \sum_{h=1}^H \|\boldsymbol{w}_h - \boldsymbol{u}_h\|_2.$$

*Proof.* Note that for any policy $\pi$, the value of the policy can be expressed as

$$V_0^{\pi}(s_1) = \sum_{h=1}^H \langle \boldsymbol{\theta}_h, \boldsymbol{\phi}_{\pi,h} \rangle.$$

Thus,

$$|V_0^{\pi^{\boldsymbol{w}}}(s_1) - V_0^{\pi^{\boldsymbol{u}}}(s_1)| \le \sum_{h=1}^H |\langle \boldsymbol{\theta}_h, \boldsymbol{\phi}_{\pi^{\boldsymbol{w}},h} - \boldsymbol{\phi}_{\pi^{\boldsymbol{u}},h} \rangle|.$$

So it suffices to bound $|\langle \boldsymbol{\theta}_h, \boldsymbol{\phi}_{\pi^{\boldsymbol{w}},h} - \boldsymbol{\phi}_{\pi^{\boldsymbol{u}},h} \rangle|$. Using the same decomposition as in the proof of Lemma B.2, we have

$$\boldsymbol{\phi}_{\pi^{\boldsymbol{w}},h} - \boldsymbol{\phi}_{\pi^{\boldsymbol{u}},h} = \sum_{i=0}^{h-1} \left( \prod_{j=h-i+1}^h \mathcal{T}_{\pi^{\boldsymbol{w}},j} \right) (\mathcal{T}_{\pi^{\boldsymbol{w}},h-i} - \mathcal{T}_{\pi^{\boldsymbol{u}},h-i}) \boldsymbol{\phi}_{\pi^{\boldsymbol{u}},h-i-1}.$$

By definition,

$$\mathcal{T}_{\pi^{\boldsymbol{w}},h-i} - \mathcal{T}_{\pi^{\boldsymbol{u}},h-i} = \int (\boldsymbol{\phi}_{\pi^{\boldsymbol{w}},h-i}(s) - \boldsymbol{\phi}_{\pi^{\boldsymbol{u}},h-i}(s)) \mathrm{d}\boldsymbol{\mu}_{h-i-1}(s)^\top$$

where

$$\boldsymbol{\phi}_{\pi^{\boldsymbol{w}},h-i}(s) = \sum_{a \in \widetilde{\mathcal{A}}_s} \boldsymbol{\phi}(s,a) \pi_{h-i}^{\boldsymbol{w}}(a|s).$$

Now, for $a \in \widetilde{\mathcal{A}}_s$,

$$\nabla_{\boldsymbol{w}_h} \pi_h^{\boldsymbol{w}}(a|s) = \frac{\eta \boldsymbol{\phi}(s,a) e^{\eta \langle \boldsymbol{\phi}(s,a), \boldsymbol{w}_h \rangle} \cdot \sum_{a' \in \widetilde{\mathcal{A}}_s} e^{\eta \langle \boldsymbol{\phi}(s,a'), \boldsymbol{w}_h \rangle} - e^{\eta \langle \boldsymbol{\phi}(s,a), \boldsymbol{w}_h \rangle} \cdot \sum_{a' \in \widetilde{\mathcal{A}}_s} \eta \boldsymbol{\phi}(s,a') e^{\eta \langle \boldsymbol{\phi}(s,a'), \boldsymbol{w}_h \rangle}}{(\sum_{a' \in \widetilde{\mathcal{A}}_s} e^{\eta \langle \boldsymbol{\phi}(s,a'), \boldsymbol{w}_h \rangle})^2}$$

so

$$\|\nabla_{\boldsymbol{w}_h} \pi_h^{\boldsymbol{w}}(a|s)\|_2 \le \frac{2\eta e^{\eta \langle \boldsymbol{\phi}(s,a), \boldsymbol{w}_h \rangle}}{\sum_{a' \in \widetilde{\mathcal{A}}_s} e^{\eta \langle \boldsymbol{\phi}(s,a'), \boldsymbol{w}_h \rangle}}$$

Thus, by the Mean Value Theorem,

$$|\pi_h^{\boldsymbol{w}}(a|s) - \pi_h^{\boldsymbol{u}}(a|s)| \le \frac{2\eta e^{\eta \langle \boldsymbol{\phi}(s,a), \boldsymbol{w}_h \rangle}}{\sum_{a' \in \widetilde{\mathcal{A}}_s} e^{\eta \langle \boldsymbol{\phi}(s,a'), \boldsymbol{w}_h \rangle}} \cdot \|\boldsymbol{w}_h - \boldsymbol{u}_h\|_2$$

so

$$\|\boldsymbol{\phi}_{\pi^{\boldsymbol{w}},h-i}(s) - \boldsymbol{\phi}_{\pi^{\boldsymbol{u}},h-i}(s)\|_2 \le \sum_{a \in \widetilde{\mathcal{A}}_s} |\pi_{h-i}^{\boldsymbol{w}}(a|s) - \pi_{h-i}^{\boldsymbol{u}}(a|s)|$$

$$\le \sum_{a \in \widetilde{\mathcal{A}}_s} \frac{2\eta e^{\eta \langle \boldsymbol{\phi}(s,a), \boldsymbol{w}_h \rangle}}{\sum_{a' \in \widetilde{\mathcal{A}}_s} e^{\eta \langle \boldsymbol{\phi}(s,a'), \boldsymbol{w}_h \rangle}} \cdot \|\boldsymbol{w}_{h-i} - \boldsymbol{u}_{h-i}\|_2$$

$$\leq 2\eta\|\boldsymbol{w}_{h-1} - \boldsymbol{u}_{h-1}\|_2$$

which, with Definition 3.1, implies that

$$\|\mathcal{T}_{\pi^{\boldsymbol{w}},h-i} - \mathcal{T}_{\pi^{\boldsymbol{u}},h-i}\|_{\mathrm{op}} \leq \int \|\boldsymbol{\phi}_{\pi^{\boldsymbol{w}},h-i}(s) - \boldsymbol{\phi}_{\pi^{\boldsymbol{u}},h-i}(s)\|_2 \|\mathrm{d}\boldsymbol{\mu}_{h-i-1}(s)\|_2 \leq 2\sqrt{d}\eta\|\boldsymbol{w}_{h-i} - \boldsymbol{u}_{h-i}\|_2.$$

By Lemma A.8, we can bound $\|\boldsymbol{\theta}_h^\top \left(\prod_{j=h-i+1}^h \mathcal{T}_{\pi^{\boldsymbol{w}},j}\right)\|_2$. Thus, returning to the error decomposition given above, we have

$$\begin{aligned}
|V_0^{\pi^{\boldsymbol{w}}}(s_1) - V_0^{\pi^{\boldsymbol{u}}}(s_1)| &\leq \sum_{h=1}^H \sum_{i=0}^{h-1} \left|\boldsymbol{\theta}_h^\top \left(\prod_{j=h-i+1}^h \mathcal{T}_{\pi^{\boldsymbol{w}},j}\right) (\mathcal{T}_{\pi^{\boldsymbol{w}},h-i} - \mathcal{T}_{\pi^{\boldsymbol{u}},h-i})\boldsymbol{\phi}_{\pi^{\boldsymbol{u}},h-i-1}\right| \\
&\leq \sqrt{d} \sum_{h=1}^H \sum_{i=0}^{h-1} \|\mathcal{T}_{\pi^{\boldsymbol{w}},h-i} - \mathcal{T}_{\pi^{\boldsymbol{u}},h-i}\|_{\mathrm{op}} \|\boldsymbol{\phi}_{\pi^{\boldsymbol{u}},h-i-1}\|_2 \\
&\leq 2d\eta \sum_{h=1}^H \sum_{i=0}^{h-1} \|\boldsymbol{w}_{h-i} - \boldsymbol{u}_{h-i}\|_2 \\
&\leq 2dH\eta \sum_{h=1}^H \|\boldsymbol{w}_h - \boldsymbol{u}_h\|_2.
\end{aligned}$$

$\square$

**Lemma A.13.** *Let $\boldsymbol{w}^\star$ denote the weights such that $Q_h^\star(s,a) = \langle\boldsymbol{\phi}(s,a), \boldsymbol{w}_h^\star\rangle$, and $\pi^{\boldsymbol{w}^\star}$ the restricted-action linear softmax policy with action sets $(\widetilde{\mathcal{A}}_s)_{s\in\mathcal{S}}$ as defined in Lemma A.11 with $\delta = \frac{\epsilon}{3(3\sqrt{d})^H}$. Then*

$$|V_0^{\pi^{\boldsymbol{w}^\star}}(s_1) - V_0^\star(s_1)| \leq \epsilon$$

*as long as $\eta \geq 2dH\log(1 + 16Hd/\epsilon) \cdot \frac{(3\sqrt{d})^H}{\epsilon}$.*

*Proof.* We prove this by induction. Assume that at step $h$, for all $s$, we have $|V_h^\star(s) - V_h^{\pi^{\boldsymbol{w}^\star}}(s)| \leq \delta_h$ for some $\delta_h$. Then,

$$\begin{aligned}
|Q_{h-1}^{\pi^{\boldsymbol{w}^\star}}(s,a) - Q_{h-1}^\star(s,a)| &= \left|\int (V_h^{\pi^{\boldsymbol{w}^\star}}(s') - V_h^\star(s'))\mathrm{d}\boldsymbol{\mu}_{h-1}(s')^\top \boldsymbol{\phi}(s,a)\right| \\
&\leq \int |V_h^{\pi^{\boldsymbol{w}^\star}}(s') - V_h^\star(s')| \|\mathrm{d}\boldsymbol{\mu}_{h-1}(s')\|_2 \|\boldsymbol{\phi}(s,a)\|_2 \\
&\leq \sqrt{d}\delta_h
\end{aligned}$$

where we use the linear MDP assumption in the last inequality. Thus,

$$\begin{aligned}
V_{h-1}^{\pi^{\boldsymbol{w}^\star}}(s) &= \frac{\sum_{a\in\widetilde{\mathcal{A}}_s} e^{\eta\langle\boldsymbol{\phi}(s,a), \boldsymbol{w}_{h-1}^\star\rangle} Q_{h-1}^{\pi^{\boldsymbol{w}^\star}}(s,a)}{\sum_{a\in\widetilde{\mathcal{A}}_s} e^{\eta\langle\boldsymbol{\phi}(s,a), \boldsymbol{w}_{h-1}^\star\rangle}} \\
&= \frac{\sum_{a\in\widetilde{\mathcal{A}}_s} e^{\eta Q_{h-1}^\star(s,a)} Q_{h-1}^{\pi^{\boldsymbol{w}^\star}}(s,a)}{\sum_{a\in\widetilde{\mathcal{A}}_s} e^{\eta Q_{h-1}^\star(s,a)}} \\
&\geq \frac{\sum_{a\in\widetilde{\mathcal{A}}_s} e^{\eta Q_{h-1}^\star(s,a)} Q_{h-1}^\star(s,a)}{\sum_{a\in\widetilde{\mathcal{A}}_s} e^{\eta Q_{h-1}^\star(s,a)}} - \sqrt{d}\delta_h.
\end{aligned}$$

By Lemma A.3, as long as $\eta \geq \log|\widetilde{\mathcal{A}}_s|/(\sqrt{d}\delta_h)$, we can lower bound

$$\frac{\sum_{a\in\widetilde{\mathcal{A}}_s} e^{\eta Q_{h-1}^\star(s,a)} Q_{h-1}^\star(s,a)}{\sum_{a\in\widetilde{\mathcal{A}}_s} e^{\eta Q_{h-1}^\star(s,a)}} - \sqrt{d}\delta_h \geq \max_{a\in\widetilde{\mathcal{A}}_s} Q_{h-1}^\star(s,a) - 2\sqrt{d}\delta_h.$$

Furthermore, by Lemma A.11 and our choice of $\widetilde{\mathcal{A}}_s$, we have

$$\max_{a\in\widetilde{\mathcal{A}}_s} Q^\star_{h-1}(s,a) - 2\sqrt{d}\delta_h \geq \max_{a\in\mathcal{A}} Q^\star_{h-1}(s,a) - 2\sqrt{d}\delta_h - \frac{\epsilon}{3(3\sqrt{d})^H} = V^\star_{h-1}(s) - 2\sqrt{d}\delta_h - \frac{\epsilon}{3(3\sqrt{d})^H}.$$

Define recursively $\delta_{h-1} = 3\sqrt{d}\delta_h$ and $\delta_H = \frac{\epsilon}{(3\sqrt{d})^H}$. Then $\delta_{h-1} = \frac{\epsilon}{(3\sqrt{d})^{h-1}} \geq \frac{\epsilon}{(3\sqrt{d})^H}$, so

$$V^\star_{h-1}(s) - 2\sqrt{d}\delta_h - \frac{\epsilon}{3(3\sqrt{d})^H} \geq V^\star_{h-1}(s) - 2\sqrt{d}\delta_h - \delta_{h-1}/3 = V^\star_{h-1}(s) - \delta_{h-1}.$$

So $|V^\star_h(s) - V^{\pi^{w^\star}}_h(s)| \leq \delta_{h-1}$ for all $s$, which proves the inductive step.

For the base case, we have

$$\begin{aligned}
V^{\pi^{w^\star}}_H(s) - V^\star_H(s) &= \frac{\sum_{a\in\widetilde{\mathcal{A}}_s} e^{\eta Q^\star_H(s,a)}\nu_H(s,a)}{\sum_{a\in\widetilde{\mathcal{A}}_s} e^{\eta Q^\star_H(s,a)}} - \max_a \nu_H(s,a) \\
&\geq \max_{a\in\widetilde{\mathcal{A}}_s} \nu_H(s,a) - \max_a \nu_H(s,a) - \delta_H/2 \\
&\geq -\delta_H
\end{aligned}$$

where the first inequality holds by Lemma A.3 as long as $\eta \geq 2\log|\widetilde{\mathcal{A}}_s|/\delta_H$, and the second inequality holds by Lemma A.11 and our choice of $\widetilde{\mathcal{A}}_s$ and $\delta_H$. This proves the base case, since $V^{\pi^{w^\star}}_H(s) \leq V^\star_H(s)$.

Recursing this all the way back, we conclude that

$$V^{\pi^{w^\star}}_0(s_1) \geq V^\star_0(s_1) - \delta_0$$

for $\delta_0 = (3\sqrt{d})^H \delta_H = \epsilon$.

For this argument to hold, we must choose $\eta \geq 2\log|\widetilde{\mathcal{A}}_s|/\delta_H$ and $\eta \geq \log|\widetilde{\mathcal{A}}_s|/(\sqrt{d}\delta_h)$ for all $s$ and $h$. By Lemma A.11 and our choice of $\widetilde{\mathcal{A}}_s$, we can bound

$$|\widetilde{\mathcal{A}}_s| \leq (1 + 8H\sqrt{d}(2\sqrt{d})^H/\epsilon)^d \leq (1 + 16Hd/\epsilon)^{dH}$$

so it suffices that we take $\eta \geq 2dH\log(1 + 16Hd/\epsilon) \cdot \frac{(3\sqrt{d})^H}{\epsilon}$.
$\qquad\square$

**Lemma A.14.** *Let* $\eta = 2dH\log(1 + 16Hd/\epsilon) \cdot \frac{(3\sqrt{d})^H}{\epsilon}$ *and* $\mathcal{W}$ *an* $\frac{\epsilon}{4dH^2\eta}$-*net of* $\mathcal{B}^d(2H\sqrt{d})$. *Let* $\Pi$ *denote the set of restricted-action linear softmax policy with vectors* $w \in \mathcal{W}^H$, *parameter* $\eta$, *and action sets* $(\widetilde{\mathcal{A}}_s)_{s\in\mathcal{S}}$ *as defined in Lemma A.11 with* $\delta = \frac{\epsilon}{3(3\sqrt{d})^H}$. *Then for any MDP and reward function, there exists some* $\pi \in \Pi$ *such that* $|V^\pi_0 - V^\star_0| \leq \epsilon$, *and*

$$|\Pi| \leq \left(1 + \frac{32H^4 d^{5/2}\log(1 + 16Hd/\epsilon)}{\epsilon^2}\right)^{dH^2}.$$

*Proof.* Consider some MDP and reward function, and let $\{w^\star_h\}^H_{h=1}$ denote the optimal $Q$-function linear representation: $Q^\star_h(s,a) = \langle\phi(s,a), w^\star_h\rangle$. Let $\widetilde{w}$ denote the vector in $\mathcal{W}^H$ such that $\sum^H_{h=1}\|w^\star_h - \widetilde{w}_h\|_2$ is minimized. Then by Lemma A.12 and Lemma A.13, as long as $\eta \geq 2dH\log(1 + 16Hd/\epsilon) \cdot \frac{(3\sqrt{d})^H}{\epsilon}$, we have

$$\begin{aligned}
|V^{\pi^{\widetilde{w}}}_0(s_1) - V^\star_0(s_1)| &\leq |V^{\pi^{\widetilde{w}}}_0(s_1) - V^{\pi^{w^\star}}_0(s_1)| + |V^{\pi^{w^\star}}_0(s_1) - V^\star_0(s_1)| \\
&\leq 2dH\eta\sum^H_{h=1}\|w^\star_h - \widetilde{w}_h\|_2 + \epsilon/2.
\end{aligned}$$

The first conclusion then follows as long as we can find some $\widetilde{w}$ such that

$$2dH\eta\sum^H_{h=1}\|w^\star_h - \widetilde{w}_h\|_2 \leq \epsilon/2.$$

However, by Lemma A.10, we can bound $\|\boldsymbol{w}_h^\star\|_2 \le 2H\sqrt{d}$. Therefore, since $\mathcal{W}$ is a $\frac{\epsilon}{4dH^2\eta}$-net of $\mathcal{B}^d(2H\sqrt{d})$, for each $h$ there will exist some $\widetilde{\boldsymbol{w}}_h \in \mathcal{W}$ such that $\|\boldsymbol{w}_h^\star - \widetilde{\boldsymbol{w}}_h\|_2 \le \frac{\epsilon}{4dH^2\eta}$, which implies that we can find $\widetilde{\boldsymbol{w}} \in \mathcal{W}^d$ such that

$$2dH\eta \sum_{h=1}^{H} \|\boldsymbol{w}_h^\star - \widetilde{\boldsymbol{w}}_h\|_2 \le \epsilon/2,$$

which gives the first conclusion.

To bound the size of $\Pi$, we apply Lemma A.1 and our choice of $\eta$ to bound

$$|\mathcal{W}| \le (1 + \frac{16H^3 d^{3/2}\eta}{\epsilon})^d \le (1 + \frac{32H^4 d^{5/2}\log(1+16Hd/\epsilon)}{\epsilon^2})^{dH}.$$

The bound on $|\Pi|$ follows since $|\Pi| = |\mathcal{W}|^H$.

$\square$

# B  Policy Elimination

Throughout this section, assuming we have run for some number of episodes $K$, we let $(\mathcal{F}_\tau)_{\tau=1}^K$ the filtration on this, with $\mathcal{F}_\tau$ the filtration up to and including episode $\tau$. We also let $\mathcal{F}_{\tau,h}$ denote the filtration on all episodes $\tau' < \tau$, and on steps $h' = 1, \ldots, h$ of episode $\tau$.

## B.1  Estimating Feature-Visitations and Rewards

**Lemma B.1.** *Assume that we have collected some data* $\{(s_{h-1,\tau}, a_{h-1,\tau}, s_{h,\tau})\}_{\tau=1}^K$, *where, for each* $\tau'$, $s_{h,\tau'}|\mathcal{F}_{h-1,\tau'}$ *is independent of* $\{(s_{h-1,\tau}, a_{h-1,\tau}, s_{h,\tau})\}_{\tau \neq \tau'}$. *Denote* $\boldsymbol{\phi}_{h-1,\tau} = \boldsymbol{\phi}(s_{h-1,\tau}, a_{h-1,\tau})$ *and* $\boldsymbol{\Lambda}_{h-1} = \sum_{\tau=1}^K \boldsymbol{\phi}_{h-1,\tau}\boldsymbol{\phi}_{h-1,\tau}^\top + \lambda I$. *Fix* $\pi$ *and let*

$$\widehat{\mathcal{T}}_{\pi,h} = \left( \sum_{\tau=1}^K \boldsymbol{\phi}_{\pi,h}(s_{h,\tau})\boldsymbol{\phi}_{h-1,\tau}^\top \right) \boldsymbol{\Lambda}_{h-1}^{-1}.$$

*Fix* $\boldsymbol{v} \in \mathbb{R}^d$ *satisfying* $|\boldsymbol{v}^\top \boldsymbol{\phi}_{\pi,h}(s)| \le 1$ *for all* $s$ *and* $\boldsymbol{u} \in \mathbb{R}^d$. *Then with probability at least* $1 - \delta$, *we can bound*

$$|\boldsymbol{v}^\top(\mathcal{T}_{\pi,h} - \widehat{\mathcal{T}}_{\pi,h})\boldsymbol{u}| \le \left( 2\sqrt{\log 2/\delta} + \frac{\log 2/\delta}{\sqrt{\lambda_{\min}(\boldsymbol{\Lambda}_{h-1})}} + \sqrt{\lambda}\|\mathcal{T}_{\pi,h}^\top \boldsymbol{v}\|_2 \right) \cdot \|\boldsymbol{u}\|_{\boldsymbol{\Lambda}_{h-1}^{-1}}.$$

*Proof.* Let $\mathfrak{D} = \{(s_{h-1,\tau}, a_{h-1,\tau})\}_{\tau=1}^K$, our data collected at step $h-1$. Then by our assumption on the independence of $s_{h,\tau}$, we have that $s_{h,\tau}|\mathcal{F}_{h-1,\tau}$ has the same distribution as $s_{h,\tau}|(\mathcal{F}_{h-1,\tau}, \mathfrak{D})$. Conditioning on $\mathfrak{D}$, the $\boldsymbol{\phi}_{h-1,\tau}$ vectors are fixed, so $\boldsymbol{\Lambda}_{h-1}$ is also fixed. Note that

$$\mathcal{T}_{\pi,h} = \int \boldsymbol{\phi}_{\pi,h}(s)\mathrm{d}\boldsymbol{\mu}_{h-1}(s)^\top$$

$$= \int \boldsymbol{\phi}_{\pi,h}(s)\mathrm{d}\boldsymbol{\mu}_{h-1}(s)^\top \left( \sum_{\tau=1}^K \boldsymbol{\phi}_{h-1,\tau}\boldsymbol{\phi}_{h-1,\tau}^\top \right) \boldsymbol{\Lambda}_{h-1}^{-1} + \lambda \int \boldsymbol{\phi}_{\pi,h}(s)\mathrm{d}\boldsymbol{\mu}_{h-1}(s)^\top \boldsymbol{\Lambda}_{h-1}^{-1}$$

$$= \sum_{\tau=1}^K \left( \int \boldsymbol{\phi}_{\pi,h}(s)\mathrm{d}\boldsymbol{\mu}_{h-1}(s)^\top \boldsymbol{\phi}_{h-1,\tau} \right) \boldsymbol{\phi}_{h-1,\tau}^\top \boldsymbol{\Lambda}_{h-1}^{-1} + \lambda \int \boldsymbol{\phi}_{\pi,h}(s)\mathrm{d}\boldsymbol{\mu}_{h-1}(s)^\top \boldsymbol{\Lambda}_{h-1}^{-1}$$

$$= \sum_{\tau=1}^K \mathbb{E}[\boldsymbol{\phi}_{\pi,h}(s_{h,\tau})|\mathcal{F}_{h-1,\tau}]\boldsymbol{\phi}_{h-1,\tau}^\top \boldsymbol{\Lambda}_{h-1}^{-1} + \lambda \int \boldsymbol{\phi}_{\pi,h}(s)\mathrm{d}\boldsymbol{\mu}_{h-1}(s)^\top \boldsymbol{\Lambda}_{h-1}^{-1}$$

$$= \sum_{\tau=1}^K \mathbb{E}[\boldsymbol{\phi}_{\pi,h}(s_{h,\tau})|\mathcal{F}_{h-1,\tau}]\boldsymbol{\phi}_{h-1,\tau}^\top \boldsymbol{\Lambda}_{h-1}^{-1} + \lambda \mathcal{T}_{\pi,h}\boldsymbol{\Lambda}_{h-1}^{-1}$$

so

$$|\boldsymbol{v}^\top(\mathcal{T}_{\pi,h} - \widehat{\mathcal{T}}_{\pi,h})\boldsymbol{u}| \le \underbrace{\left| \sum_{\tau=1}^K \boldsymbol{v}^\top \left( \mathbb{E}[\boldsymbol{\phi}_{\pi,h}(s_{h,\tau})|\mathcal{F}_{h-1,\tau}] - \boldsymbol{\phi}_{\pi,h}(s_{h,\tau}) \right) \boldsymbol{\phi}_{h-1,\tau}^\top \boldsymbol{\Lambda}_{h-1}^{-1}\boldsymbol{u} \right|}_{(a)} + \underbrace{\left| \lambda \boldsymbol{v}^\top \mathcal{T}_{\pi,h}\boldsymbol{\Lambda}_{h-1}^{-1}\boldsymbol{u} \right|}_{(b)}.$$

Conditioned on $\mathfrak{D}$, $(a)$ is simply the sum of mean 0 random variables, where the $\tau$th random variable has magnitude bounded as

$$|\boldsymbol{v}^\top \left(\mathbb{E}[\boldsymbol{\phi}_{\pi,h}(s_{h,\tau})|\mathcal{F}_{h-1,\tau}] - \boldsymbol{\phi}_{\pi,h}(s_{h,\tau})\right) \boldsymbol{\phi}_{h-1,\tau}^\top \boldsymbol{\Lambda}_{h-1}^{-1} \boldsymbol{u}| \leq 2|\boldsymbol{\phi}_{h-1,\tau}^\top \boldsymbol{\Lambda}_{h-1}^{-1} \boldsymbol{u}|$$
$$\leq 2\|\boldsymbol{\phi}_{h-1,\tau}\|_{\boldsymbol{\Lambda}_{h-1}^{-1}} \|\boldsymbol{u}\|_{\boldsymbol{\Lambda}_{h-1}^{-1}}$$
$$\leq 2\|\boldsymbol{u}\|_{\boldsymbol{\Lambda}_{h-1}^{-1}}/\sqrt{\lambda_{\min}(\boldsymbol{\Lambda}_{h-1})}$$

Furthermore, the variance of each term in $(a)$ is bounded as

$$\mathrm{Var}\left[\boldsymbol{v}^\top \left(\mathbb{E}[\boldsymbol{\phi}_{\pi,h}(s_{h,\tau})|\mathcal{F}_{h-1,\tau}] - \boldsymbol{\phi}_{\pi,h}(s_{h,\tau})\right) \boldsymbol{\phi}_{h-1,\tau}^\top \boldsymbol{\Lambda}_{h-1}^{-1} \boldsymbol{u}|\mathcal{F}_{h-1}\right]$$
$$= \mathbb{E}\left[\left(\boldsymbol{v}^\top \left(\mathbb{E}[\boldsymbol{\phi}_{\pi,h}(s_{h,\tau})|\mathcal{F}_{h-1,\tau}] - \boldsymbol{\phi}_{\pi,h}(s_{h,\tau})\right) \boldsymbol{\phi}_{h-1,\tau}^\top \boldsymbol{\Lambda}_{h-1}^{-1} \boldsymbol{u}\right)^2 |\mathcal{F}_{h-1}\right]$$
$$\leq \boldsymbol{u}^\top \boldsymbol{\Lambda}_{h-1}^{-1} \boldsymbol{\phi}_{h-1,\tau} \boldsymbol{\phi}_{h-1,\tau}^\top \boldsymbol{\Lambda}_{h-1}^{-1} \boldsymbol{u}.$$

It follows that, by Bernstein's Inequality, we can bound, with probability at least $1 - \delta$ conditioned on $\mathfrak{D}$:

$$(a) \leq 2\sqrt{\sum_{\tau=1}^K \boldsymbol{u}^\top \boldsymbol{\Lambda}_{h-1}^{-1} \boldsymbol{\phi}_{h-1,\tau} \boldsymbol{\phi}_{h-1,\tau}^\top \boldsymbol{\Lambda}_{h-1}^{-1} \boldsymbol{u} \cdot \log \frac{2}{\delta}} + \frac{2\|\boldsymbol{u}\|_{\boldsymbol{\Lambda}_{h-1}^{-1}}}{\sqrt{\lambda_{\min}(\boldsymbol{\Lambda}_{h-1})}} \cdot \log \frac{2}{\delta}$$
$$\leq 2(\sqrt{\log 2/\delta} + \frac{\log 2/\delta}{\sqrt{\lambda_{\min}(\boldsymbol{\Lambda}_{h-1})}}) \cdot \|\boldsymbol{u}\|_{\boldsymbol{\Lambda}_{h-1}^{-1}}.$$

In other words,

$$\mathbb{P}\left[(a) \geq 2(\sqrt{\log 2/\delta} + \frac{\log 2/\delta}{\sqrt{\lambda_{\min}(\boldsymbol{\Lambda}_{h-1})}}) \cdot \|\boldsymbol{u}\|_{\boldsymbol{\Lambda}_{h-1}^{-1}}|\mathfrak{D}\right] \leq \delta$$

so, by the law of total probability, for any distribution $F$ over $\mathfrak{D}$,

$$\mathbb{P}\left[(a) \geq 2(\sqrt{\log 2/\delta} + \min\{1, \lambda^{-1}\} \log 2/\delta) \cdot \|\boldsymbol{u}\|_{\boldsymbol{\Lambda}_{h-1}^{-1}}\right]$$
$$= \int \mathbb{P}\left[(a) \geq 2(\sqrt{\log 2/\delta} + \min\{1, \lambda^{-1}\} \log 2/\delta) \cdot \|\boldsymbol{u}\|_{\boldsymbol{\Lambda}_{h-1}^{-1}}|\mathfrak{D}\right] dF(\mathfrak{D})$$
$$\leq \delta \int dF(\mathfrak{D})$$
$$= \delta.$$

We can also bound

$$(b) \leq \sqrt{\lambda}\|\boldsymbol{u}\|_{\boldsymbol{\Lambda}_{h-1}^{-1}} \|\mathcal{T}_{\pi,h}^\top \boldsymbol{v}\|_2.$$

Combining these gives the result. $\qquad\square$

**Lemma B.2.** *Fix $\pi$ and let*

$$\widehat{\boldsymbol{\phi}}_{\pi,h} = \widehat{\mathcal{T}}_{\pi,h} \widehat{\mathcal{T}}_{\pi,h-1} \ldots \widehat{\mathcal{T}}_{\pi,2} \widehat{\mathcal{T}}_{\pi,1} \boldsymbol{\phi}_{\pi,0}.$$

*Fix $\boldsymbol{u} \in \mathcal{S}^{d-1}$ or $\boldsymbol{u}$ a valid reward vector as defined by Definition 3.1. Then with probability at least $1 - \delta$:*

$$|\langle \boldsymbol{u}, \boldsymbol{\phi}_{\pi,h} - \widehat{\boldsymbol{\phi}}_{\pi,h} \rangle| \leq \sum_{i=1}^{h-1} \left(2\sqrt{\log \frac{2H}{\delta}} + \frac{\log \frac{2H}{\delta}}{\sqrt{\lambda_{\min}(\boldsymbol{\Lambda}_i)}} + \sqrt{d\lambda}\right) \cdot \|\widehat{\boldsymbol{\phi}}_{\pi,i}\|_{\boldsymbol{\Lambda}_i^{-1}}.$$

*Proof.* Note that

$$\boldsymbol{\phi}_{\pi,h} - \widehat{\boldsymbol{\phi}}_{\pi,h} = \mathcal{T}_{\pi,h} \boldsymbol{\phi}_{\pi,h-1} - \widehat{\mathcal{T}}_{\pi,h} \widehat{\boldsymbol{\phi}}_{\pi,h-1}$$
$$= \mathcal{T}_{\pi,h}(\boldsymbol{\phi}_{\pi,h-1} - \widehat{\boldsymbol{\phi}}_{\pi,h-1}) + (\mathcal{T}_{\pi,h} - \widehat{\mathcal{T}}_{\pi,h})\widehat{\boldsymbol{\phi}}_{\pi,h-1}.$$

Thus, unrolling this all the way back, we get

$$\boldsymbol{\phi}_{\pi,h} - \widehat{\boldsymbol{\phi}}_{\pi,h} = \sum_{i=1}^{h-1} \left( \prod_{j=h-i+1}^{h} \mathcal{T}_{\pi,j} \right) (\mathcal{T}_{\pi,h-i} - \widehat{\mathcal{T}}_{\pi,h-i}) \widehat{\boldsymbol{\phi}}_{\pi,h-i-1}$$

where we order the product $\prod_{j=h-i+1}^{h} \mathcal{T}_{\pi,j} = \mathcal{T}_{\pi,h}\mathcal{T}_{\pi,h-1} \ldots \mathcal{T}_{\pi,h-i+1}$. It follows that

$$|\langle \boldsymbol{u}, \boldsymbol{\phi}_{\pi,h} - \widehat{\boldsymbol{\phi}}_{\pi,h}\rangle| \le \sum_{i=1}^{h-1} \left| \boldsymbol{u}^\top \left( \prod_{j=h-i+1}^{h} \mathcal{T}_{\pi,j} \right) (\mathcal{T}_{\pi,h-i} - \widehat{\mathcal{T}}_{\pi,h-i}) \widehat{\boldsymbol{\phi}}_{\pi,h-i-1} \right|.$$

Denote $\boldsymbol{v}_i := \boldsymbol{u}^\top \left( \prod_{j=h-i+1}^{h} \mathcal{T}_{\pi,j} \right)$. By Lemma A.8 and our assumption on $\boldsymbol{u}$, we can bound $\|\boldsymbol{v}_i\|_2 \le \sqrt{d}$ and also have that for all $s, a$, $|\boldsymbol{v}_i^\top \boldsymbol{\phi}(s,a)| \le 1$, which implies

$$|\boldsymbol{v}_i^\top \boldsymbol{\phi}_{\pi,j}(s)| = \left| \sum_{a \in \mathcal{A}} \boldsymbol{v}_i^\top \boldsymbol{\phi}(s,a) \pi_h(a|s) \right| \le \sum_{a \in \mathcal{A}} \pi_h(a|s) = 1.$$

We can therefore apply Lemma B.1 to get that, with probability at least $1 - \delta$, for all $i$,

$$\left| \boldsymbol{v}_i^\top (\mathcal{T}_{\pi,h-i} - \widehat{\mathcal{T}}_{\pi,h-i}) \widehat{\boldsymbol{\phi}}_{\pi,h-i-1} \right| \le \left( 2\sqrt{\log \frac{2H}{\delta}} + \frac{\log \frac{2H}{\delta}}{\sqrt{\lambda_{\min}(\boldsymbol{\Lambda}_{h-i-1})}} + \sqrt{\lambda} \|\mathcal{T}_{\pi,h}^\top \boldsymbol{v}_i\|_2 \right) \cdot \|\widehat{\boldsymbol{\phi}}_{\pi,h-i-1}\|_{\boldsymbol{\Lambda}_{h-i-1}^{-1}}$$

By Lemma A.8, the definition of $\boldsymbol{v}_i$, and our assumption on $\boldsymbol{u}$, we can bound $\|\mathcal{T}_{\pi,h}^\top \boldsymbol{v}_i\|_2 \le \sqrt{d}$. Summing over $i$ proves the result. $\qquad \square$

**Lemma B.3.** *With probability at least $1 - \delta$:*

$$\|\widehat{\boldsymbol{\phi}}_{\pi,h} - \boldsymbol{\phi}_{\pi,h}\|_2 \le d \sum_{h'=1}^{h-1} \left( 2\sqrt{\log \frac{2Hd}{\delta}} + \frac{\log \frac{2Hd}{\delta}}{\sqrt{\lambda_{\min}(\boldsymbol{\Lambda}_{h'})}} + \sqrt{d\lambda} \right) \cdot \|\widehat{\boldsymbol{\phi}}_{\pi,h'}\|_{\boldsymbol{\Lambda}_{h'}^{-1}}.$$

*Proof.* We have:

$$\|\widehat{\boldsymbol{\phi}}_{\pi,h} - \boldsymbol{\phi}_{\pi,h}\|_2 \le \|\widehat{\boldsymbol{\phi}}_{\pi,h} - \boldsymbol{\phi}_{\pi,h}\|_1 = \sum_{i=1}^{d} |[\widehat{\boldsymbol{\phi}}_{\pi,h}]_i - [\boldsymbol{\phi}_{\pi,h}]_i| = \sum_{i=1}^{d} |\langle \boldsymbol{e}_i, \widehat{\boldsymbol{\phi}}_{\pi,h} - \boldsymbol{\phi}_{\pi,h}\rangle|.$$

Since $\boldsymbol{e}_i \in \mathcal{S}^{d-1}$, we can apply Lemma B.2 to bound, with probability $1 - \delta/d$,

$$|\langle \boldsymbol{e}_i, \widehat{\boldsymbol{\phi}}_{\pi,h} - \boldsymbol{\phi}_{\pi,h}\rangle| \le \sum_{h'=0}^{h-1} \left( 2\sqrt{\log \frac{2Hd}{\delta}} + \frac{\log \frac{2Hd}{\delta}}{\sqrt{\lambda_{\min}(\boldsymbol{\Lambda}_{h'})}} + \sqrt{d\lambda} \right) \cdot \|\widehat{\boldsymbol{\phi}}_{\pi,h'}\|_{\boldsymbol{\Lambda}_{h'}^{-1}}.$$

Summing over $i$ gives the result. $\qquad \square$

**Lemma B.4.** *Assume we have collected data $\{\boldsymbol{\phi}(s_{h,\tau}, a_{h,\tau}), r_h(s_{h,\tau}, a_{h,\tau})\}_{\tau=1}^{K}$ and that for each $\tau'$, $r_h(s_{h,\tau'}, a_{h,\tau'})|(s_{h,\tau'}, a_{h,\tau'})$ is independent of $\{(s_{h,\tau}, a_{h,\tau})\}_{\tau \ne \tau'}$. Let*

$$\widehat{\boldsymbol{\theta}}_h = \arg\min_{\boldsymbol{\theta}} \sum_{\tau=1}^{K} (r_{h,\tau} - \langle \boldsymbol{\phi}_{h,\tau}, \boldsymbol{\theta}\rangle)^2 + \lambda \|\boldsymbol{\theta}\|_2^2$$

*and fix $\boldsymbol{u} \in \mathbb{R}^d$ that is independent of $\{\boldsymbol{\phi}(s_{h,\tau}, a_{h,\tau}), r_h(s_{h,\tau}, a_{h,\tau})\}_{\tau=1}^{K}$. Then with probability at least $1 - \delta$:*

$$|\langle \boldsymbol{u}, \widehat{\boldsymbol{\theta}}_h - \boldsymbol{\theta}_h\rangle| \le \left( \sqrt{\log 2/\delta} + \frac{\log 2/\delta}{\sqrt{\lambda_{\min}(\boldsymbol{\Lambda}_h)}} + \sqrt{d\lambda} \right) \cdot \|\boldsymbol{u}\|_{\boldsymbol{\Lambda}_h^{-1}}.$$

*Proof.* Let $\mathfrak{D} = \{(s_{h,\tau}, a_{h,\tau})\}_{\tau=1}^{K}$. Then by our assumption on the independence of $r_{h,\tau}$, we have that $r_{h,\tau}|(s_{h,\tau}, a_{h,\tau})$ has the same distribution as $r_{h,\tau}|\mathfrak{D}$. Conditioning on $\mathfrak{D}$, the $\boldsymbol{\phi}_{h,\tau}$ vectors are fixed, so $\boldsymbol{\Lambda}_h$ is also fixed.

By construction we have

$$\widehat{\boldsymbol{\theta}}_h = \boldsymbol{\Lambda}_h^{-1} \sum_{\tau=1}^K \boldsymbol{\phi}_{h,\tau} r_{h,\tau}.$$

Furthermore:

$$\boldsymbol{\theta}_h = \boldsymbol{\Lambda}_h^{-1} \boldsymbol{\Lambda}_h \boldsymbol{\theta}_h = \boldsymbol{\Lambda}_h^{-1} \sum_{\tau=1}^K \boldsymbol{\phi}_{h,\tau} \mathbb{E}[r_{h,\tau}|\mathcal{F}_{h-1,\tau}] + \lambda \boldsymbol{\Lambda}_h^{-1} \boldsymbol{\theta}_h.$$

Thus,

$$|\langle \boldsymbol{u}, \widehat{\boldsymbol{\theta}}_h - \boldsymbol{\theta}_h \rangle| \le \underbrace{\left| \sum_{\tau=1}^K \boldsymbol{u}^\top \boldsymbol{\Lambda}_h^{-1} \boldsymbol{\phi}_{h,\tau}(r_{h,\tau} - \mathbb{E}[r_{h,\tau}|\mathcal{F}_{h-1,\tau}]) \right|}_{(a)} + \underbrace{\left| \lambda \boldsymbol{u}^\top \boldsymbol{\Lambda}_h^{-1} \boldsymbol{\theta}_h \right|}_{(b)}.$$

Since $R_{h,\tau} \in [0,1]$ almost surely, we can bound

$$|\boldsymbol{u}^\top \boldsymbol{\Lambda}_h^{-1} \boldsymbol{\phi}_{h,\tau}(r_{h,\tau} - \mathbb{E}[r_{h,\tau}|\mathcal{F}_{h-1,\tau}])| \le \|\boldsymbol{u}\|_{\boldsymbol{\Lambda}_h^{-1}} \|\boldsymbol{\phi}_{h,\tau}\|_{\boldsymbol{\Lambda}_h^{-1}} \le \|\boldsymbol{u}\|_{\boldsymbol{\Lambda}_h^{-1}} / \sqrt{\lambda_{\min}(\boldsymbol{\Lambda}_h)}.$$

Furthermore, we can bound

$$\begin{aligned}
\mathrm{Var} &\left[ \boldsymbol{u}^\top \boldsymbol{\Lambda}_h^{-1} \boldsymbol{\phi}_{h,\tau}(r_{h,\tau} - \mathbb{E}[r_{h,\tau}|\mathcal{F}_{h-1,\tau}])|\mathfrak{D} \right] \\
&= \mathbb{E}\left[ (\boldsymbol{u}^\top \boldsymbol{\Lambda}_h^{-1} \boldsymbol{\phi}_{h,\tau}(r_{h,\tau} - \mathbb{E}[r_{h,\tau}|\mathcal{F}_{h-1,\tau}]))^2|\mathfrak{D} \right] \\
&\le \boldsymbol{u}^\top \boldsymbol{\Lambda}_h^{-1} \boldsymbol{\phi}_{h,\tau} \boldsymbol{\phi}_{h,\tau}^\top \boldsymbol{\Lambda}_h^{-1} \boldsymbol{u}.
\end{aligned}$$

By Bernstein's inequality, we then have, with probability at least $1 - \delta$ conditioned on $\mathfrak{D}$:

$$\begin{aligned}
(a) &\le \sqrt{\sum_{\tau=1}^K \boldsymbol{u}^\top \boldsymbol{\Lambda}_h^{-1} \boldsymbol{\phi}_{h,\tau} \boldsymbol{\phi}_{h,\tau}^\top \boldsymbol{\Lambda}_h^{-1} \boldsymbol{u} \cdot \log 2/\delta} + \frac{\|\boldsymbol{u}\|_{\boldsymbol{\Lambda}_h^{-1}} \cdot \log 2/\delta}{\sqrt{\lambda_{\min}(\boldsymbol{\Lambda}_h)}} \\
&\le \left( \sqrt{\log 2/\delta} + \frac{\log 2/\delta}{\sqrt{\lambda_{\min}(\boldsymbol{\Lambda}_h)}} \right) \cdot \|\boldsymbol{u}\|_{\boldsymbol{\Lambda}_h^{-1}}.
\end{aligned}$$

Applying the Law of Total Probability as in Lemma B.1, we obtain

$$\mathbb{P}\left[ (a) \ge \left( \sqrt{\log 2/\delta} + \frac{\log 2/\delta}{\sqrt{\lambda_{\min}(\boldsymbol{\Lambda}_h)}} \right) \cdot \|\boldsymbol{u}\|_{\boldsymbol{\Lambda}_h^{-1}} \right] \le \delta.$$

By Definition 3.1, we can also bound

$$(b) \le \sqrt{\lambda} \|\boldsymbol{u}\|_{\boldsymbol{\Lambda}_h^{-1}} \|\boldsymbol{\theta}_h\|_2 \le \sqrt{d\lambda} \|\boldsymbol{u}\|_{\boldsymbol{\Lambda}_h^{-1}}.$$

Combining these proves the result. □

## B.2 Correctness and Sample Complexity of PEDEL

**Lemma B.5.** *Let $\mathcal{E}_{\mathrm{est}}^{\ell,h}$ denote the event on which, for all $\pi \in \Pi_\ell$:*

$$|\langle \boldsymbol{\theta}_{h+1}, \widehat{\boldsymbol{\phi}}_{\pi,h+1}^\ell - \boldsymbol{\phi}_{\pi,h+1} \rangle| \le \sum_{i=1}^h \left( 3\sqrt{\log \frac{4H^2|\Pi_\ell|\ell^2}{\delta}} + \frac{\log \frac{4H^2|\Pi_\ell|\ell^2}{\delta}}{\sqrt{\lambda_{\min}(\boldsymbol{\Lambda}_{i,\ell})}} \right) \cdot \|\widehat{\boldsymbol{\phi}}_{\pi,i}^\ell\|_{\boldsymbol{\Lambda}_{i,\ell}^{-1}},$$

$$\|\widehat{\boldsymbol{\phi}}_{\pi,h+1}^\ell - \boldsymbol{\phi}_{\pi,h+1}\|_2 \le d \sum_{i=1}^h \left( 3\sqrt{\log \frac{4H^2 d|\Pi_\ell|\ell^2}{\delta}} + \frac{\log \frac{4H^2 d|\Pi_\ell|\ell^2}{\delta}}{\sqrt{\lambda_{\min}(\boldsymbol{\Lambda}_{i,\ell})}} \right) \cdot \|\widehat{\boldsymbol{\phi}}_{\pi,i}^\ell\|_{\boldsymbol{\Lambda}_{i,\ell}^{-1}},$$

$$|\langle \widehat{\boldsymbol{\phi}}_{\pi,h}^\ell, \widehat{\boldsymbol{\theta}}_h - \boldsymbol{\theta}_h \rangle| \le \left( 2\sqrt{\log \frac{4H^2|\Pi_\ell|\ell^2}{\delta}} + \frac{\log \frac{4H^2|\Pi_\ell|\ell^2}{\delta}}{\sqrt{\lambda_{\min}(\boldsymbol{\Lambda}_{h,\ell})}} \right) \cdot \|\widehat{\boldsymbol{\phi}}_{\pi,h}^\ell\|_{\boldsymbol{\Lambda}_{h,\ell}^{-1}}.$$

*Then $\mathbb{P}[(\mathcal{E}_{\mathrm{est}}^{\ell,h})^c] \le \frac{\delta}{2H\ell^2}$.*

**Algorithm 3** **P**olicy **L**earning via **E**xperiment **De**sign in **L**inear MDPs (PEDEL, full version)

1: **input:** tolerance $\epsilon$, confidence $\delta$, policy set $\Pi$
2: $\ell_0 \leftarrow \lceil \log_2 \frac{d^{3/2}}{H} \rceil$, $\Pi_{\ell_0} \leftarrow \Pi$, $\widehat{\phi}^1_{\pi,1} \leftarrow \mathbb{E}_{a \sim \pi_1(\cdot|s_1)}[\phi(s_1, a)], \forall \pi \in \Pi$
3: **for** $\ell = \ell_0, \ell_0 + 1, \ldots, \lceil \log \frac{4}{\epsilon} \rceil$ **do**
4: $\quad \epsilon_\ell \leftarrow 2^{-\ell}$, $\beta_\ell \leftarrow 64H^4 \log \frac{4H^2|\Pi_\ell|\ell^2}{\delta}$
5: $\quad$ **for** $h = 1, 2, \ldots, H$ **do**
6: $\quad\quad$ Run procedure described in Theorem 9 with parameters

$$\epsilon_{\exp} \leftarrow \frac{\epsilon_\ell^2}{\beta_\ell}, \quad \delta \leftarrow \frac{\delta}{2H\ell^2}, \quad \underline{\lambda} \leftarrow \log \frac{4H^2|\Pi_\ell|\ell^2}{\delta}, \quad \Phi \leftarrow \Phi_{h,\ell} := \{\widehat{\phi}^\ell_{\pi,h} : \pi \in \Pi_\ell\}$$

$\quad\quad$ and denote returned data as $\{(s_{h,\tau}, a_{h,\tau}, r_{h,\tau}, s_{h+1,\tau})\}_{\tau=1}^{K_{h,\ell}}$, for $K_{h,\ell}$ total number of episodes run , and covariates

$$\Lambda_{h,\ell} \leftarrow \sum_{\tau=1}^{K_{h,\ell}} \phi(s_{h,\tau}, a_{h,\tau}) \phi(s_{h,\tau}, a_{h,\tau})^\top + 1/d \cdot I$$

7: $\quad\quad$ **for** $\pi \in \Pi_\ell$ **do** $\qquad$ // Estimate feature-visitations for active policies
8: $\quad\quad\quad \widehat{\phi}^\ell_{\pi,h+1} \leftarrow \left( \sum_{\tau=1}^{K_{h,\ell}} \phi_{\pi,h+1}(s_{h+1,\tau}) \phi_{h,\tau}^\top \Lambda_{h,\ell}^{-1} \right) \widehat{\phi}^\ell_{\pi,h}$
9: $\quad\quad \widehat{\theta}^\ell_h \leftarrow \Lambda_{h,\ell}^{-1} \sum_{\tau=1}^{K_{h,\ell}} \phi_{h,\tau} r_{h,\tau}$ $\qquad\qquad$ // Estimate reward vectors
10: $\quad$ // Remove provably suboptimal policies from active policy set

$$\Pi_{\ell+1} \leftarrow \Pi_\ell \setminus \left\{ \pi \in \Pi_\ell : \widehat{V}_0^\pi < \sup_{\pi' \in \Pi_\ell} \widehat{V}_0^{\pi'} - 2\epsilon_\ell \right\} \quad \text{for} \quad \widehat{V}_0^\pi := \sum_{h=1}^H \langle \widehat{\phi}^\ell_{\pi,h}, \widehat{\theta}^\ell_h \rangle$$

11: $\quad$ **if** $|\Pi_{\ell+1}| = 1$ **then return** $\pi \in \Pi_{\ell+1}$
12: **return** any $\pi \in \Pi_{\ell+1}$

---

*Proof.* Note that the data collection procedure outlined in Theorem 9 collects data that satisfies the independence requirement of Lemma B.1 and Lemma B.4, since Theorem 9 operates on the $h$-truncated-horizon MDP defined with respect to our original MDP (see Definition C.2 and following discussion), so by construction the data obtained at step $h$ is independent of $s_{h+1}$ and $r_h(s_h, a_h)$. Note also that $\widehat{\phi}^\ell_{\pi,h}$ is independent of $\{r^\ell_{h,\tau}\}_{\tau=1}^{K_{h,\ell}} | \{(s_{h,\tau}, a_{h,\tau})\}_{\tau=1}^{K_{h,\ell}}$, since we construct $\widehat{\phi}^\ell_{\pi,h}$ using only observations taken at step $h - 1$.

The result follows by Lemma B.2, Lemma B.3, and Lemma B.4, and setting $\lambda = 1/d$. $\qquad\square$

**Lemma B.6.** *Let $\mathcal{E}^{\ell,h}_{\exp}$ denote the event on which:*

- *The exploration procedure on Line 6 terminates after running for at most*

$$C \cdot \frac{\inf_{\Lambda \in \Omega_h} \max_{\phi \in \Phi_{\ell,h}} \|\phi\|^2_{A(\Lambda)^{-1}}}{\epsilon_\ell^2 / \beta_\ell} + \text{poly}\left(d, H, \log \frac{\ell^2}{\delta}, \frac{1}{\lambda^\star_{\min}}, \log |\Pi_\ell|\right)$$

*episodes.*

- *The covariates returned by Line 6 for any $(h, \ell)$, $\Lambda_{h,\ell}$, satisfy*

$$\max_{\phi \in \Phi_{\ell,h}} \|\phi\|^2_{\Lambda_{h,\ell}^{-1}} \leq \frac{\epsilon_\ell^2}{\beta_\ell}, \qquad \lambda_{\min}(\Lambda_{h,\ell}) \geq \log \frac{4H^2|\Pi_\ell|\ell^2}{\delta}.$$

*Then $\mathbb{P}[(\mathcal{E}^{\ell,h}_{\exp})^c \cap \mathcal{E}^{\ell,h-1}_{\text{est}} \cap (\cap_{i=1}^{h-1} \mathcal{E}^{\ell,i}_{\exp})] \leq \frac{\delta}{2H\ell^2}$.*

*Proof.* By Lemma B.7, on the event $\mathcal{E}^{\ell,h-1}_{\text{est}} \cap (\cap_{i=1}^{h-1} \mathcal{E}^{\ell,i}_{\exp})$ we can bound $\|\widehat{\phi}^\ell_{\pi,h} - \phi_{\pi,h}\|_2 \leq d\epsilon_\ell/2H$. By Lemma A.6, we can lower bound $\|\phi_{\pi,h}\|_2 \geq 1/\sqrt{d}$. By the reverse triangle inequality,

$$\|\widehat{\phi}^\ell_{\pi,h}\|_2 \geq \|\phi_{\pi,h}\|_2 - \|\widehat{\phi}^\ell_{\pi,h} - \phi_{\pi,h}\|_2 \geq 1/\sqrt{d} - d\epsilon_\ell/2H.$$

It follows that as long as $\epsilon_\ell \leq H/d^{3/2}$, that we can lower bound $\|\widehat{\phi}_{\pi,h}^\ell\|_2 \geq 1/(2\sqrt{d})$. Since we start $\ell$ at $\ell = \lceil \log_2 \frac{d^{3/2}}{H} \rceil$, we will have that $\epsilon_\ell = 2^{-\ell} \leq H/d^{3/2}$.

The result then follows by applying Theorem 9 with our chosen parameters and $\gamma_\Phi \leftarrow 1/(2\sqrt{d})$.
$\square$

**Lemma B.7.** *On the event* $\mathcal{E}_{\text{est}}^{\ell,h} \cap (\cap_{i=1}^h \mathcal{E}_{\text{exp}}^{\ell,i})$, *for all* $\pi \in \Pi_\ell$:
$$|\langle \boldsymbol{\theta}_{h+1}, \widehat{\phi}_{\pi,h+1}^\ell - \phi_{\pi,h+1}\rangle| \leq \epsilon_\ell/2H,$$
$$\|\widehat{\phi}_{\pi,h+1}^\ell - \phi_{\pi,h+1}\|_2 \leq d\epsilon_\ell/2H,$$
$$|\langle \widehat{\phi}_{\pi,h}^\ell, \widehat{\boldsymbol{\theta}}_h - \boldsymbol{\theta}_h \rangle| \leq \epsilon_\ell/2H.$$

*Proof.* On $\mathcal{E}_{\text{exp}}^{\ell,i}$, we can lower bound
$$\lambda_{\min}(\boldsymbol{\Lambda}_{i,\ell}) \geq \log \frac{4H^2|\Pi_\ell|\ell^2}{\delta}$$
which implies
$$3\sqrt{\log \frac{4H^2|\Pi_\ell|\ell^2}{\delta}} + \frac{\log \frac{4H^2|\Pi_\ell|\ell^2}{\delta}}{\sqrt{\lambda_{\min}(\boldsymbol{\Lambda}_{i,\ell})}} \leq 4\sqrt{\log \frac{4H^2|\Pi_\ell|\ell^2}{\delta}}.$$
Furthermore, on $\mathcal{E}_{\text{exp}}^{\ell,i}$, $\|\widehat{\phi}_{\pi,i}^\ell\|_{\boldsymbol{\Lambda}_{i,\ell}^{-1}} \leq \frac{\epsilon_\ell}{\sqrt{\beta_\ell}}$. Since $\beta_\ell = 64H^4 \log \frac{4H^2|\Pi_\ell|\ell^2}{\delta}$, on $\mathcal{E}_{\text{est}}^{\ell,h}$, we can then upper bound
$$|\langle \boldsymbol{\theta}_{h+1}, \widehat{\phi}_{\pi,h+1}^\ell - \phi_{\pi,h+1}\rangle| \leq \sum_{i=1}^h \left(3\sqrt{\log \frac{4H^2|\Pi_\ell|\ell^2}{\delta}} + \frac{\log \frac{4H^2|\Pi_\ell|\ell^2}{\delta}}{\sqrt{\lambda_{\min}(\boldsymbol{\Lambda}_{i,\ell})}}\right) \cdot \|\widehat{\phi}_{\pi,i}^\ell\|_{\boldsymbol{\Lambda}_{i,\ell}^{-1}}$$
$$\leq H4\sqrt{\log \frac{4H^2|\Pi_\ell|\ell^2}{\delta}} \frac{\epsilon_\ell}{\sqrt{\beta_\ell}}$$
$$\leq \epsilon_\ell/2H.$$
The same calculation gives the bounds on $\|\widehat{\phi}_{\pi,h}^\ell - \phi_{\pi,h}\|_2$ and $|\langle \widehat{\phi}_{\pi,h}^\ell, \widehat{\boldsymbol{\theta}}_h - \boldsymbol{\theta}_h \rangle|$.
$\square$

**Lemma B.8.** *Define* $\mathcal{E}_{\text{exp}} = \cap_\ell \cap_h \mathcal{E}_{\text{exp}}^{\ell,h}$ *and* $\mathcal{E}_{\text{est}} = \cap_\ell \cap_h \mathcal{E}_{\text{est}}^{\ell,h}$. *Then* $\mathbb{P}[\mathcal{E}_{\text{est}} \cap \mathcal{E}_{\text{exp}}] \geq 1 - 2\delta$ *and on* $\mathcal{E}_{\text{est}} \cap \mathcal{E}_{\text{exp}}$, *for all* $h, \ell$, *and* $\pi \in \Pi_\ell$,
$$|\langle \boldsymbol{\theta}_{h+1}, \widehat{\phi}_{\pi,h+1}^\ell - \phi_{\pi,h+1}\rangle| \leq \epsilon_\ell/2H,$$
$$\|\widehat{\phi}_{\pi,h+1}^\ell - \phi_{\pi,h+1}\|_2 \leq d\epsilon_\ell/2H,$$
$$|\langle \widehat{\phi}_{\pi,h}^\ell, \widehat{\boldsymbol{\theta}}_h - \boldsymbol{\theta}_h \rangle| \leq \epsilon_\ell/2H.$$

*Proof.* Clearly,
$$\mathcal{E}_{\text{est}}^c \cup \mathcal{E}_{\text{exp}}^c = \bigcup_{\ell=\ell_0}^{\lceil \log 4/\epsilon \rceil} \bigcup_{h=1}^H ((\mathcal{E}_{\text{est}}^{\ell,h})^c \cup (\mathcal{E}_{\text{exp}}^{\ell,h})^c)$$
$$= \bigcup_{\ell=\ell_0}^{\lceil \log 4/\epsilon \rceil} \bigcup_{h=1}^H (\mathcal{E}_{\text{est}}^{\ell,h})^c \backslash \left((\mathcal{E}_{\text{est}}^{\ell,h-1})^c \cup (\cup_{i=1}^{h-1}(\mathcal{E}_{\text{exp}}^{\ell,i})^c)\right) \cup \bigcup_{\ell=\ell_0}^{\lceil \log 4/\epsilon \rceil} \bigcup_{h=1}^H (\mathcal{E}_{\text{exp}}^{\ell,h})^c$$
$$= \bigcup_{\ell=\ell_0}^{\lceil \log 4/\epsilon \rceil} \bigcup_{h=1}^H (\mathcal{E}_{\text{est}}^{\ell,h})^c \cap \left(\mathcal{E}_{\text{est}}^{\ell,h-1} \cap (\cup_{i=1}^{h-1}\mathcal{E}_{\text{exp}}^{\ell,i})\right) \cup \bigcup_{\ell=\ell_0}^{\lceil \log 4/\epsilon \rceil} \bigcup_{h=1}^H (\mathcal{E}_{\text{exp}}^{\ell,h})^c.$$
The first conclusion follows by Lemma B.5, Lemma B.5, and since we can bound
$$\sum_\ell \sum_{h=1}^H 2 \cdot \frac{\delta}{2H\ell^2} \leq \frac{\pi^2}{6}\delta \leq 2\delta.$$
The second conclusion follows by Lemma B.7.
$\square$

**Lemma B.9.** *On the event $\mathcal{E}_{\text{est}} \cap \mathcal{E}_{\exp}$, for all $\ell > \ell_0$, every policy $\pi \in \Pi_\ell$ satisfies $V_0^\star(\Pi) - V_0^\pi \le 4\epsilon_\ell$ and $\widetilde{\pi}^\star \in \Pi_\ell$, for $\widetilde{\pi}^\star = \arg\max_{\pi \in \Pi} V_0^\pi$.*

*Proof.* The value of a policy $\pi$ is given by

$$\sum_{h=1}^H \langle \boldsymbol{\theta}_h, \boldsymbol{\phi}_{\pi,h} \rangle.$$

By Lemma B.8, for all $\pi \in \Pi_\ell$ we can bound

$$|\langle \widehat{\boldsymbol{\theta}}_h, \widehat{\boldsymbol{\phi}}_{\pi,h}^\ell \rangle - \langle \boldsymbol{\theta}_h, \boldsymbol{\phi}_{\pi,h} \rangle| \le |\langle \widehat{\boldsymbol{\theta}}_h - \boldsymbol{\theta}_h, \widehat{\boldsymbol{\phi}}_{\pi,h}^\ell \rangle| + |\langle \boldsymbol{\theta}_h, \widehat{\boldsymbol{\phi}}_{\pi,h}^\ell - \boldsymbol{\phi}_{\pi,h} \rangle| \le \epsilon_\ell/2H + \epsilon_\ell/2H = \epsilon_\ell/H.$$

Thus,

$$\left| \sum_{h=1}^H \langle \widehat{\boldsymbol{\theta}}_h^\ell, \widehat{\boldsymbol{\phi}}_{\pi,h}^\ell \rangle - \sum_{h=1}^H \langle \boldsymbol{\theta}_h, \boldsymbol{\phi}_{\pi,h} \rangle \right| \le \epsilon_\ell.$$

We will only include $\pi \in \Pi_{\ell+1}$ if $\pi \in \Pi_\ell$ and

$$\sum_{h=1}^H \langle \widehat{\boldsymbol{\phi}}_{\pi,h}^\ell, \widehat{\boldsymbol{\theta}}_h^\ell \rangle \ge \sup_{\pi' \in \Pi_\ell} \sum_{h=1}^H \langle \widehat{\boldsymbol{\phi}}_{\pi',h}^\ell, \widehat{\boldsymbol{\theta}}_h^\ell \rangle - 2\epsilon_\ell.$$

Using the estimation error given above, this implies that for any $\pi \in \Pi_\ell$,

$$V_0^\pi = \sum_{h=1}^H \langle \boldsymbol{\theta}_h, \boldsymbol{\phi}_{\pi,h} \rangle \ge \sup_{\pi' \in \Pi_\ell} \sum_{h=1}^H \langle \boldsymbol{\theta}_h, \boldsymbol{\phi}_{\pi',h} \rangle - 4\epsilon_\ell = \sup_{\pi' \in \Pi_\ell} V_0^{\pi'} - 4\epsilon_\ell.$$

Both claims then follow if we can show $\widetilde{\pi}^\star$ is always contained in the active set. Assume that $\widetilde{\pi}^\star \in \Pi_\ell$. Then

$$\sum_{h=1}^H \langle \widehat{\boldsymbol{\phi}}_{\widetilde{\pi}^\star,h}^\ell, \widehat{\boldsymbol{\theta}}_h^\ell \rangle \ge V_0^{\widetilde{\pi}^\star} - \epsilon_\ell, \quad \sup_{\pi' \in \Pi_\ell} \sum_{h=1}^H \langle \widehat{\boldsymbol{\phi}}_{\pi',h}^\ell, \widehat{\boldsymbol{\theta}}_h^\ell \rangle \le \sup_{\pi' \in \Pi_\ell} \sum_{h=1}^H \langle \boldsymbol{\phi}_{\pi',h}, \boldsymbol{\theta}_h \rangle + \epsilon_\ell = V_0^{\widetilde{\pi}^\star} + \epsilon_\ell.$$

Rearranging this gives

$$\sum_{h=1}^H \langle \widehat{\boldsymbol{\phi}}_{\widetilde{\pi}^\star,h}^\ell, \widehat{\boldsymbol{\theta}}_h^\ell \rangle \ge \sup_{\pi' \in \Pi_\ell} \sum_{h=1}^H \langle \widehat{\boldsymbol{\phi}}_{\pi',h}^\ell, \widehat{\boldsymbol{\theta}}_h^\ell \rangle - 2\epsilon_\ell$$

so $\widetilde{\pi}^\star \in \Pi_{\ell+1}$. $\qquad\square$

**Theorem 6.** *With probability at least $1 - 2\delta$, Algorithm 1 will terminate after collecting at most*

$$CH^4 \sum_{h=1}^H \sum_{\ell=\ell_0+1}^{\iota_0} \frac{\inf_{\boldsymbol{\Lambda} \in \boldsymbol{\Omega}_h} \max_{\pi \in \Pi(4\epsilon_\ell)} \|\boldsymbol{\phi}_{\pi,h}\|_{\boldsymbol{\Lambda}^{-1}}^2}{\epsilon_\ell^2} \cdot \log \frac{H|\Pi(4\epsilon_\ell)| \log \frac{1}{\epsilon}}{\delta} + \text{poly}\left(d, H, \frac{1}{\lambda_{\min}^\star}, \log \frac{1}{\delta}, \log |\Pi|, \log \frac{1}{\epsilon}\right)$$

$$+ CH^4 \sum_{h=1}^H \frac{\inf_{\boldsymbol{\Lambda} \in \boldsymbol{\Omega}_h} \max_{\pi \in \Pi} \|\boldsymbol{\phi}_{\pi,h}\|_{\boldsymbol{\Lambda}^{-1}}^2}{\epsilon_{\ell_0}^2} \cdot \log \frac{H|\Pi| \log(1/\epsilon)}{\delta}$$

*episodes for $\iota_0 := \min\{\lceil \log \frac{4}{\epsilon} \rceil, \log \frac{4}{\Delta_{\min}(\Pi)}\}$, and will output a policy $\widehat{\pi}$ such that*

$$V_0^{\widehat{\pi}} \ge \max_{\pi \in \Pi} V_0^\pi - \epsilon,$$

*where here $\Pi(4\epsilon_\ell) = \{\pi \in \Pi : V_0^\pi \ge \max_{\pi \in \Pi} V_0^\pi - 4\epsilon_\ell\}$.*

*Proof.* By Lemma B.8 the event $\mathcal{E}_{\text{est}} \cap \mathcal{E}_{\exp}$ occurs with probability at least $1 - 2\delta$. Henceforth we assume we are on this event.

Correctness follows by Lemma B.9, since upon termination, $\Pi_\ell$ will only contain policies $\pi$ satisfying $V_0^\pi \ge \max_{\pi \in \Pi} V_0^\pi - \epsilon$ (and will contain at least 1 policy since $\widetilde{\pi}^\star \in \Pi_\ell$ for all $\ell$).

Furthermore, by Lemma B.9, if $4\epsilon_\ell < \Delta_{\min}(\Pi)$, we must have that $\Pi_\ell = \{\widetilde{\pi}^\star\}$, and will therefore terminate on Line 11 since $|\Pi_\ell| = 1$. Thus, we can bound the number of number of epochs by

$$\iota_0 := \min\{\lceil \log \frac{4}{\epsilon} \rceil, \log \frac{4}{\Delta_{\min}(\Pi)}\}.$$

By Lemma B.6, the total number of episodes collected is bounded by

$$\sum_{h=1}^{H} \sum_{\ell=1}^{\iota_0} C \cdot \frac{\inf_{\mathbf{\Lambda} \in \mathbf{\Omega}_h} \max_{\phi \in \Phi_{\ell,h}} \|\phi\|_{\mathbf{A}(\mathbf{\Lambda})^{-1}}^2}{\epsilon_\ell^2 / \beta_\ell} + \mathrm{poly}\left(d, H, \log \frac{1}{\delta}, \frac{1}{\lambda_{\min}^\star}, \log |\Pi|, \log \frac{1}{\epsilon}\right)$$

$$\leq \sum_{h=1}^{H} \sum_{\ell=1}^{\iota_0} C \cdot \frac{\inf_{\mathbf{\Lambda} \in \mathbf{\Omega}_h} \max_{\phi \in \Phi_{\ell,h}} \|\phi\|_{\mathbf{A}(\mathbf{\Lambda})^{-1}}^2}{\epsilon_\ell^2} \cdot H^4 \log \frac{H|\Pi_\ell| \log(1/\epsilon)}{\delta} + \mathrm{poly}\left(d, H, \log \frac{1}{\delta}, \frac{1}{\lambda_{\min}^\star}, \log |\Pi|, \log \frac{1}{\epsilon}\right).$$

On $\mathcal{E}_{\mathrm{est}} \cap \mathcal{E}_{\mathrm{exp}}$, by Lemma B.8, for each $\pi \in \Pi_\ell$, we have $\|\widehat{\phi}_{\pi,h}^\ell - \phi_{\pi,h}\|_2 \leq d\epsilon_\ell/2H$. As $\Phi_{\ell,h} = \{\widehat{\phi}_{\pi,h}^\ell : \pi \in \Pi_\ell\}$, it follows that we can upper bound

$$\inf_{\mathbf{\Lambda} \in \mathbf{\Omega}_h} \max_{\phi \in \Phi_{\ell,h}} \|\phi\|_{\mathbf{A}(\mathbf{\Lambda})^{-1}}^2 = \inf_{\mathbf{\Lambda} \in \mathbf{\Omega}_h} \max_{\pi \in \Pi_\ell} \|\widehat{\phi}_{\pi,h}^\ell\|_{\mathbf{A}(\mathbf{\Lambda})^{-1}}^2$$

$$\leq \inf_{\mathbf{\Lambda} \in \mathbf{\Omega}_h} \max_{\pi \in \Pi_\ell} (2\|\phi_{\pi,h}\|_{\mathbf{A}(\mathbf{\Lambda})^{-1}}^2 + 2\|\widehat{\phi}_{\pi,h}^\ell - \phi_{\pi,h}\|_{\mathbf{A}(\mathbf{\Lambda})^{-1}}^2)$$

$$\leq \inf_{\mathbf{\Lambda} \in \mathbf{\Omega}_h} \max_{\pi \in \Pi_\ell} (2\|\phi_{\pi,h}\|_{\mathbf{A}(\mathbf{\Lambda})^{-1}}^2 + \frac{d^2 \epsilon_\ell^2}{2H^2 \lambda_{\min}(\mathbf{A}(\mathbf{\Lambda}))})$$

$$\leq \inf_{\mathbf{\Lambda} \in \mathbf{\Omega}_h} \max_{\pi \in \Pi_\ell} 4\|\phi_{\pi,h}\|_{\mathbf{A}(\mathbf{\Lambda})^{-1}}^2 + \inf_\pi \frac{d^2 \epsilon_\ell^2}{H^2 \lambda_{\min}(\mathbf{A}(\mathbf{\Lambda}))}$$

$$\leq \inf_{\mathbf{\Lambda} \in \mathbf{\Omega}_h} \max_{\pi \in \Pi_\ell} 4\|\phi_{\pi,h}\|_{\mathbf{\Lambda}^{-1}}^2 + \frac{d^2 \epsilon_\ell^2}{H^2 \lambda_{\min}^\star}$$

so

$$\frac{\inf_{\mathbf{\Lambda} \in \mathbf{\Omega}_h} \max_{\phi \in \Phi_{\ell,h}} \|\phi\|_{\mathbf{A}(\mathbf{\Lambda})^{-1}}^2}{\epsilon_\ell^2} \leq \frac{\inf_{\mathbf{\Lambda} \in \mathbf{\Omega}_h} \max_{\pi \in \Pi_\ell} 4\|\phi_{\pi,h}\|_{\mathbf{\Lambda}^{-1}}^2}{\epsilon_\ell^2} + \frac{d^2}{H^2 \lambda_{\min}^\star}.$$

Note also that, by Lemma B.9, for $\ell > \ell_0$, every policy $\pi \in \Pi_\ell$ will be $4\epsilon_\ell$ optimal, so we therefore have

$$\Pi_\ell \subseteq \{\pi \in \Pi \; : \; V_0^\pi \geq V_0^{\widetilde{\pi}^\star} - 4\epsilon_\ell\} =: \Pi(4\epsilon_\ell).$$

Putting this together, we can upper bound the complexity by

$$\sum_{h=1}^{H} \sum_{\ell=\ell_0+1}^{\iota_0} C \cdot \frac{\inf_{\mathbf{\Lambda} \in \mathbf{\Omega}_h} \max_{\pi \in \Pi(4\epsilon_\ell)} \|\phi_{\pi,h}\|_{\mathbf{\Lambda}^{-1}}^2}{\epsilon_\ell^2} \cdot H^4 \log \frac{H|\Pi(4\epsilon_\ell)| \log(1/\epsilon)}{\delta} + \mathrm{poly}\left(d, H, \log \frac{1}{\delta}, \frac{1}{\lambda_{\min}^\star}, \log |\Pi|, \log \frac{1}{\epsilon}\right)$$

$$+ C \cdot \frac{\inf_{\mathbf{\Lambda} \in \mathbf{\Omega}_h} \max_{\pi \in \Pi} \|\phi_{\pi,h}\|_{\mathbf{\Lambda}^{-1}}^2}{\epsilon_{\ell_0}^2} \cdot H^4 \log \frac{H|\Pi| \log(1/\epsilon)}{\delta}.$$

$\square$

**Corollary 5** (Full Statement of Theorem 1). *With probability at least $1 - \delta$, the complexity of Algorithm 1 can be bounded as*

$$CH^4 \log \frac{1}{\epsilon} \cdot \sum_{h=1}^{H} \inf_{\mathbf{\Lambda} \in \mathbf{\Omega}_h} \max_{\pi \in \Pi} \frac{\|\phi_{\pi,h}\|_{\mathbf{\Lambda}^{-1}}^2}{(V_0^\star(\Pi) - V_0^\pi)^2 \vee \epsilon^2 \vee \Delta_{\min}(\Pi)^2} \cdot \log \frac{H|\Pi| \log \frac{1}{\epsilon}}{\delta}$$

$$+ \mathrm{poly}\left(d, H, \frac{1}{\lambda_{\min}^\star}, \log \frac{1}{\delta}, \log |\Pi|, \log \frac{1}{\epsilon}\right)$$

*episodes, and Algorithm 1 will output a policy $\widehat{\pi}$ such that*

$$V_0^{\widehat{\pi}} \geq \max_{\pi \in \Pi} V_0^\pi - \epsilon.$$

*Proof.* By the definition of $\Pi(4\epsilon_\ell)$, for each $\pi \in \Pi(4\epsilon_\ell)$ we have

$$\epsilon_\ell^2 = \frac{1}{16}\left((V_0^\star(\Pi) - V_0^\pi)^2 \vee (4\epsilon_\ell)^2\right).$$

We can therefore upper bound

$$\sum_{\ell=\ell_0+1}^{\iota_0} \frac{\inf_{\mathbf{\Lambda}\in\mathbf{\Omega}_h} \max_{\pi\in\Pi(4\epsilon_\ell)} \|\boldsymbol{\phi}_{\pi,h}\|_{\mathbf{\Lambda}^{-1}}^2}{\epsilon_\ell^2} \cdot \log\frac{H|\Pi(4\epsilon_\ell)|\log\frac{1}{\epsilon}}{\delta}$$

$$\leq C \sum_{\ell=\ell_0+1}^{\iota_0} \inf_{\mathbf{\Lambda}\in\mathbf{\Omega}_h} \max_{\pi\in\Pi(4\epsilon_\ell)} \frac{\|\boldsymbol{\phi}_{\pi,h}\|_{\mathbf{\Lambda}^{-1}}^2}{(V_0^\star(\Pi) - V_0^\pi)^2 \vee \epsilon_\ell^2} \cdot \log\frac{H|\Pi(4\epsilon_\ell)|\log\frac{1}{\epsilon}}{\delta}$$

$$\leq C\log\frac{1}{\epsilon} \cdot \inf_{\mathbf{\Lambda}\in\mathbf{\Omega}_h} \max_{\pi\in\Pi} \frac{\|\boldsymbol{\phi}_{\pi,h}\|_{\mathbf{\Lambda}^{-1}}^2}{(V_0^\star(\Pi) - V_0^\pi)^2 \vee \epsilon^2 \vee \Delta_{\min}(\Pi)^2} \cdot \log\frac{H|\Pi|\log\frac{1}{\epsilon}}{\delta}.$$

Furthermore, since $\ell_0 = \lceil\log_2 d^{3/2}/H\rceil$, using Lemma B.10 we can also bound

$$C \cdot \frac{\inf_{\mathbf{\Lambda}\in\mathbf{\Omega}_h} \max_{\pi\in\Pi} \|\boldsymbol{\phi}_{\pi,h}\|_{\mathbf{\Lambda}^{-1}}^2}{\epsilon_{\ell_0}^2} \cdot H^4 \log\frac{H|\Pi|\log(1/\epsilon)}{\delta} \leq \mathrm{poly}\left(d, H, \log 1/\delta, \log|\Pi|, \log 1/\epsilon\right).$$

The result then follows by Theorem 6. $\qquad\square$

*Proof of Corollary 1.* By Lemma A.14, we can choose $\Pi_\epsilon$ to be the restricted-action linear softmax policy set constructed in Lemma A.14. Lemma A.14 shows that $\Pi_\epsilon$ will contain an $\epsilon$-optimal policy for any MDP and reward function, and that

$$|\Pi_\epsilon| \leq \left(1 + \frac{32H^4 d^{5/2}\log(1 + 16Hd/\epsilon)}{\epsilon^2}\right)^{dH^2}.$$

Combining this with the guarantee of Corollary 5 shows that $V_0^{\widehat{\pi}} \geq V_0^\star - 2\epsilon$ and that $V_0^\star(\Pi) - V_0^\pi$ is within a factor of $\epsilon$ of $V_0^\star - V_0^\pi$. To bound the complexity of this procedure, we apply the bound given in Corollary 5 with the bound on the cardinality of $\Pi_\epsilon$ given above. $\qquad\square$

## B.3 Interpreting the Complexity

**Lemma B.10.** *For any set of policies $\Pi$, we can bound*

$$\inf_{\mathbf{\Lambda}\in\mathbf{\Omega}_h} \sup_{\pi\in\Pi} \|\boldsymbol{\phi}_{\pi,h}\|_{\mathbf{\Lambda}^{-1}}^2 \leq d.$$

*Proof.* By Jensen's inequality, for any $\boldsymbol{v} \in \mathbb{R}^d$, we have

$$\boldsymbol{v}^\top \mathbf{\Lambda}_{\pi,h} \boldsymbol{v} = \mathbb{E}_\pi[(\boldsymbol{v}^\top\boldsymbol{\phi}_h)^2] \geq (\mathbb{E}_\pi[\boldsymbol{v}^\top\boldsymbol{\phi}_h])^2 = (\boldsymbol{v}^\top\boldsymbol{\phi}_{\pi,h})^2.$$

It follows that, for any $\pi$,

$$\mathbf{\Lambda}_{\pi,h} \succeq \boldsymbol{\phi}_{\pi,h}\boldsymbol{\phi}_{\pi,h}^\top.$$

Take $\mathbf{\Lambda} \in \mathbf{\Omega}_h$. Then,

$$\mathbf{\Lambda} = \mathbb{E}_{\pi\sim\omega}[\mathbf{\Lambda}_{\pi,h}] \succeq \mathbb{E}_{\pi\sim\omega}[\boldsymbol{\phi}_{\pi,h}\boldsymbol{\phi}_{\pi,h}^\top].$$

It follows that we can upper bound

$$\inf_{\mathbf{\Lambda}\in\mathbf{\Omega}_h} \sup_{\pi\in\Pi} \|\boldsymbol{\phi}_{\pi,h}\|_{\mathbf{\Lambda}^{-1}}^2 \leq \inf_{\lambda\in\triangle_\Pi} \sup_{\pi\in\Pi} \|\boldsymbol{\phi}_{\pi,h}\|_{A(\lambda)^{-1}}^2$$

where $A(\lambda) = \sum_\pi \lambda_\pi \boldsymbol{\phi}_{\pi,h}\boldsymbol{\phi}_{\pi,h}^\top$. By Kiefer-Wolfowitz (Lattimore & Szepesvári, 2020), this is upper bounded by $d$. $\qquad\square$

*Proof of Corollary 2.* This follows directly from Lemma B.10 and Corollary 1, by upper bounding:

$$\inf_{\mathbf{\Lambda}\in\mathbf{\Omega}_h} \max_{\pi\in\Pi_\epsilon} \frac{\|\boldsymbol{\phi}_{\pi,h}\|_{\mathbf{\Lambda}^{-1}}^2}{\max\{V_0^\star - V_0^\pi, \epsilon\}^2} \leq \inf_{\mathbf{\Lambda}\in\mathbf{\Omega}_h} \max_{\pi\in\Pi_\epsilon} \frac{\|\boldsymbol{\phi}_{\pi,h}\|_{\mathbf{\Lambda}^{-1}}^2}{\epsilon^2} \leq \frac{d}{\epsilon^2}.$$

$\qquad\square$

### B.3.1 Linear Contextual Bandits

Since we always assume the MDP starts in some state $s_1$, to encode a linear contextual bandit, the direct mapping of our linear MDP in Definition 3.1 would require considering an $H = 2$ MDP, where we encode the "context" in the transition to state $s$ at step $h = 2$. While we could run our algorithm directly on this, in the standard contextual bandit setting, the learner has no control over the context, and so their action before receiving that context has no effect. Thus, there is no need for the learner to explore at stage $h = 1$. To account for this, we can simply run our algorithm but ignore the exploration at stage $h = 1$, which will reduce the $h = 1$ term in the sample complexity.

### B.3.2 Tabular MDPs

**Lemma B.11.** *In the tabular MDP setting, assuming that $\Pi$ contains an optimal policy,*

$$
\inf_{\boldsymbol{\Lambda} \in \boldsymbol{\Omega}_h} \max_{\pi \in \Pi} \frac{\|\boldsymbol{\phi}_{\pi,h}\|_{\boldsymbol{\Lambda}^{-1}}^2}{(V_0^\star - V_0^\pi)^2 \vee \epsilon^2 \vee \Delta_{\min}(\Pi)^2}
$$
$$
\leq \inf_{\pi_{\exp}} \max_{\pi \in \Pi} \max_{s,a} \frac{1}{w_h^{\pi_{\exp}}(s,a)} \min \left\{ \frac{1}{w_h^\pi(s,a)\Delta_h(s,a)^2}, \frac{w_h^\pi(s,a)}{\epsilon^2 \vee \Delta_{\min}(\Pi)^2} \right\}
$$
$$
\leq \inf_{\pi_{\exp}} \max_{s,a} \frac{1}{w_h^{\pi_{\exp}}(s,a)} \cdot \frac{1}{\epsilon \max\{\Delta_h(s,a), \epsilon, \Delta_{\min}(\Pi)\}}.
$$

*Proof.* We have that $[\boldsymbol{\phi}_{\pi,h}]_{s,a} = w_h^\pi(s,a)$. Furthermore, $\boldsymbol{\phi}(s,a) = \boldsymbol{e}_{s,a}$, so for any $\boldsymbol{\Lambda} \in \boldsymbol{\Omega}_h$, $\boldsymbol{\Lambda}$ is diagonal with $[\boldsymbol{\Lambda}]_{sa,sa} = \mathbb{E}_{\pi \sim \omega}[w_h^\pi(s,a)]$. Furthermore, by the Performance-Difference Lemma, $V_0^\star - V_0^\pi = \sum_{s,a,h} w_h^\pi(s,a)\Delta_h(s,a)$. Thus,

$$
\inf_{\boldsymbol{\Lambda} \in \boldsymbol{\Omega}_h} \max_{\pi \in \Pi} \frac{\|\boldsymbol{\phi}_{\pi,h}\|_{\boldsymbol{\Lambda}^{-1}}^2}{(V_0^\star - V_0^\pi)^2 \vee \epsilon^2 \vee \Delta_{\min}(\Pi)^2} \leq \inf_{\pi_{\exp}} \max_{\pi \in \Pi} \frac{\sum_{s,a} \frac{w_h^\pi(s,a)^2}{w_h^{\pi_{\exp}}(s,a)}}{(\sum_{s',a',h'} w_{h'}^\pi(s',a')\Delta_{h'}(s',a'))^2 \vee \epsilon^2 \vee \Delta_{\min}(\Pi)^2}. \tag{B.1}
$$

We have

$$
\sum_{s,a} \frac{w_h^\pi(s,a)^2}{w_h^{\pi_{\exp}}(s,a)} \leq \left( \sum_{s,a} w_h^\pi(s,a) \right) \cdot \max_{s,a} \frac{w_h^\pi(s,a)}{w_h^{\pi_{\exp}}(s,a)} = \max_{s,a} \frac{w_h^\pi(s,a)}{w_h^{\pi_{\exp}}(s,a)}.
$$

Thus,

$$
(\text{B.1}) \leq \inf_{\pi_{\exp}} \max_{\pi \in \Pi} \max_{s,a} \frac{w_h^\pi(s,a)/w_h^{\pi_{\exp}}(s,a)}{(\sum_{s',a',h'} w_{h'}^\pi(s',a')\Delta_{h'}(s',a'))^2 \vee \epsilon^2 \vee \Delta_{\min}(\Pi)^2}
$$
$$
\leq \inf_{\pi_{\exp}} \max_{\pi \in \Pi} \max_{s,a} \frac{w_h^\pi(s,a)/w_h^{\pi_{\exp}}(s,a)}{(w_h^\pi(s,a)\Delta_h(s,a))^2 \vee \epsilon^2 \vee \Delta_{\min}(\Pi)^2}
$$
$$
= \inf_{\pi_{\exp}} \max_{\pi \in \Pi} \max_{s,a} \frac{1}{w_h^{\pi_{\exp}}(s,a)} \min \left\{ \frac{1}{w_h^\pi(s,a)\Delta_h(s,a)^2}, \frac{w_h^\pi(s,a)}{\epsilon^2 \vee \Delta_{\min}(\Pi)^2} \right\}. \tag{B.2}
$$

We can further upper bound

$$
\min \left\{ \frac{1}{w_h^\pi(s,a)\Delta_h(s,a)^2}, \frac{w_h^\pi(s,a)}{\epsilon^2 \vee \Delta_{\min}(\Pi)^2} \right\} \leq \frac{1}{\Delta_h(s,a)(\epsilon \vee \Delta_{\min}(\Pi))}
$$

so

$$
(\text{B.2}) \leq \inf_{\pi_{\exp}} \max_{\pi \in \Pi} \max_{s,a} \frac{1}{w_h^{\pi_{\exp}}(s,a)} \min \left\{ \frac{1}{\Delta_h(s,a)\epsilon}, \frac{w_h^\pi(s,a)}{\epsilon^2 \vee \Delta_{\min}(\Pi)^2} \right\}
$$
$$
\leq \inf_{\pi_{\exp}} \max_{s,a} \frac{1}{w_h^{\pi_{\exp}}(s,a)} \frac{1}{\epsilon \max\{\Delta_h(s,a), \epsilon, \Delta_{\min}(\Pi)\}}.
$$

$\square$

**Lemma B.12.** *If PEDEL is run with a set $\Pi$ that contains an optimal policy, the complexity of PEDEL is upper bounded as*

$$
\widetilde{\mathcal{O}} \left( H^4 \sum_{h=1}^H \sup_{\epsilon' \geq \max\{\epsilon, \Delta_{\min}(\Pi)/4\}} \inf_{\pi_{\exp}} \max_{\pi \in \Pi(\epsilon')} \max_{s,a} \frac{1}{w_h^{\pi_{\exp}}(s,a)} \min \left\{ \frac{1}{w_h^\pi(s,a)\Delta_h(s,a)^2}, \frac{w_h^\pi(s,a)}{(\epsilon')^2} \right\} \cdot \log \frac{|\Pi(\epsilon')|}{\delta} + C_0 \right)
$$

*for $\Pi(\epsilon') = \{\pi \in \Pi : V_0^\pi \geq V_0^\star(\Pi) - \epsilon\}$.*

*Proof.* Using an argument identical to that in Lemma B.11, we can upper bound

$$\frac{\inf_{\boldsymbol{\Lambda}\in\boldsymbol{\Omega}_h}\max_{\pi\in\Pi(4\epsilon_\ell)}\|\boldsymbol{\phi}_{\pi,h}\|_{\boldsymbol{\Lambda}^{-1}}^2}{\epsilon_\ell^2} \le \inf_{\boldsymbol{\Lambda}\in\boldsymbol{\Omega}_h}\max_{\pi\in\Pi(4\epsilon_\ell)}\frac{c\|\boldsymbol{\phi}_{\pi,h}\|_{\boldsymbol{\Lambda}^{-1}}^2}{(V_0^\star - V_0^\pi)^2 \vee \epsilon_\ell^2}$$

$$\le \inf_{\pi_{\exp}}\max_{\pi\in\Pi(4\epsilon_\ell)}\max_{s,a}\frac{1}{w_h^{\pi_{\exp}}(s,a)}\min\left\{\frac{1}{w_h^\pi(s,a)\Delta_h(s,a)^2}, \frac{w_h^\pi(s,a)}{\epsilon_\ell^2}\right\}.$$

The result then follows from Theorem 6, noting that we will never run for $\epsilon_\ell < \Delta_{\min}(\Pi)/4$. $\qquad\square$

*Proof of Corollary 3.* Note that in the tabular MDP setting, we can choose $\Pi$ to be the set of all deterministic policies, since this set is guaranteed to contain an optimal policy. We can then bound $|\Pi| \le A^{SH}$. The result then follows directly from Lemma B.11 and Theorem 1. $\qquad\square$

*Proof of Proposition 4.* We begin with an example where PEDEL has complexity smaller than the Gap-Visitation Complexity, and then turn to an example where the reverse is true.

**PEDEL Improves on Gap-Visitation Complexity.** Consider the tabular MDP with $|\mathcal{S}| = |\mathcal{A}| = N$, and where

$$P_h(s_1|s_1, a_1) = 1, \quad \nu_h(s_1, a_1) = 1, \forall h \in [H]$$
$$P_h(s_1|s_1, a_j) = 0, \quad \nu_h(s_1, a_j) = 0, \forall h \in [H], j \ne 1$$
$$P_h(s_1|s_i, a_j) = 0, \forall h \in [H], j \in [N], i \ne 1$$
$$P_h(s_i|s_j, a_i) = 1, \forall h \in [H], j \in [N], i \ne 1$$
$$r_h(s_i, a_1) = \epsilon, \forall h \in [H], i \ne 1, \quad \nu_h(s_i, a_j) = 0, \forall h \in [H], j \ne 1, i \ne 1.$$

In this MDP, the optimal policy simply plays action $a_1$ $H$ times and is always in state $s_1$. The total reward it collects is $H$. Any deterministic policy that does not play $a_1$ $H$ consecutive times has optimality gap of at least $1 - \epsilon$. Furthermore, every other state can be reached with probability 1. In this case, then, assuming that we take $\Pi$ to be the set of all deterministic policies, we have $\Delta_{\min}(\Pi) = 1 - \epsilon$ (note that since there always exists a deterministic policy that is optimal, it suffices to take $\Pi$ to be the set of all deterministic policies).

By Corollary 3, we can therefore upper bound the complexity of the leading-order term by $\widetilde{\mathcal{O}}(H^5 S^2 A)$, so PEDEL will identify the optimal policy (since $\Pi$ contains an optimal policy). Thus, the total complexity of PEDEL is $\mathcal{O}(\text{poly}(S, A, H, \log 1/\delta))$.

On this example, in every state $s_i, i \ne 1$, action $a_1$ still collects a reward of $\epsilon$. Thus, we have that $\Delta_h(s_i, a_j) = \epsilon$ for $j \ne 1$. The Gap-Visitation complexity is given by

$$\sum_{h=1}^H \inf_\pi \max_{s,a} \min\left\{\frac{1}{w_h^\pi(s,a)\Delta_h(s,a)^2}, \frac{W_h(s)^2}{\epsilon^2}\right\}.$$

Since $W_h(s) = 1$ for each $s$, we conclude that

$$\sum_{h=1}^H \inf_\pi \max_{s,a} \min\left\{\frac{1}{w_h^\pi(s,a)\Delta_h(s,a)^2}, \frac{W_h(s)^2}{\epsilon^2}\right\} \ge \sum_{h=1}^H \frac{1}{\epsilon^2}.$$

Thus, for small $\epsilon$, the Gap-Visitation complexity can be arbitrarily worse than the complexity of PEDEL.

**The Gap-Visitation Complexity Improves on PEDEL.** To show that the Gap-Visitation Complexity improves on the complexity of PEDEL, we consider the example in Instance Class 5.1 of Wagenmaker et al. (2021b). As shown by Proposition 6 of Wagenmaker et al. (2021b), on this example, for any $\epsilon$, the Gap-Visitation Complexity is $\widetilde{\mathcal{O}}(\text{poly}(S))$.

To bound the complexity of PEDEL on this example, we consider the complexity given in Theorem 6 with $\Pi$ the set of all deterministic policies, which is slightly tighter than the complexity of Corollary 3. Take $\epsilon \ge 2^{-S}$. Then, on this example, it follows that $\Delta_{\min}(\Pi) \le \mathcal{O}(\epsilon)$, since we can find a policy $\pi$ which is optimal on every state $s_i$ at step $h = 2$ for $i = \mathcal{O}(\log 1/\epsilon)$, which will give it a policy gap of $\mathcal{O}(\epsilon)$. Furthermore, any near-optimal policy will have $[\boldsymbol{\phi}_{\pi,2}]_{s_1,a_1} = w_2^\pi(s_1, a_1) = \mathcal{O}(1)$, so we always have $\inf_{\boldsymbol{\Lambda}\in\boldsymbol{\Omega}_2}\max_{\pi\in\Pi(4\epsilon_\ell)}\|\boldsymbol{\phi}_{\pi,h}\|_{\boldsymbol{\Lambda}^{-1}}^2 \ge \Omega(1)$. It follows that the complexity of PEDEL is lower bounded by $\Omega(1/\epsilon^2)$.

$\qquad\square$

### B.3.3 Deterministic, Tabular MDPs

**Lemma B.13.** *In the deterministic MDP setting,*

$$\inf_{\boldsymbol{\Lambda} \in \boldsymbol{\Omega}_h} \max_{\pi \in \Pi} \frac{\|\phi_{\pi,h}\|^2_{\boldsymbol{\Lambda}^{-1}}}{(V_0^\star - V_0^\pi)^2 \vee \epsilon^2} \leq \sum_{s,a} \frac{1}{\bar{\Delta}_h(s,a)^2 \vee \epsilon^2}.$$

*Proof.* Note that $[\phi_{\pi,h}]_{s_h^\pi, a_h^\pi} = 1$, and otherwise, for $(s,a) \neq (s_h^\pi, a_h^\pi)$, $[\phi_{\pi,h}]_{s,a} = 0$. Furthermore, $\boldsymbol{\Lambda}_{\pi_{\exp},h}$ will always be diagonal, with diagonal elements $w_h^\pi(s,a)$. We then have $\|\phi_{\pi,h}\|^2_{\boldsymbol{\Lambda}^{-1}_{\pi_{\exp},h}} = \frac{1}{w_h^{\pi_{\exp}}(s_h^\pi, a_h^\pi)}$, so

$$\begin{aligned}
\inf_{\boldsymbol{\Lambda} \in \boldsymbol{\Omega}_h} \max_{\pi \in \Pi} \frac{\|\phi_{\pi,h}\|^2_{\boldsymbol{\Lambda}^{-1}}}{(V_0^\star - V_0^\pi)^2 \vee \epsilon^2} &\leq \inf_{\pi_{\exp}} \max_{\pi \in \Pi} \frac{\|\phi_{\pi,h}\|^2_{\boldsymbol{\Lambda}^{-1}_{\pi_{\exp},h}}}{(V_0^\star - V_0^\pi)^2 \vee \epsilon^2} \\
&= \inf_{\pi_{\exp}} \max_{\pi \in \Pi} \frac{w_h^{\pi_{\exp}}(s_h^\pi, a_h^\pi)^{-1}}{(V_0^\star - V_0^\pi)^2 \vee \epsilon^2} \\
&\overset{(a)}{=} \inf_{\pi_{\exp}} \max_{s,a} \max_{\pi \in \Pi_{sah}} \frac{w_h^{\pi_{\exp}}(s_h^\pi, a_h^\pi)^{-1}}{(V_0^\star - V_0^\pi)^2 \vee \epsilon^2} \\
&\overset{(b)}{=} \inf_{\pi_{\exp}} \max_{s,a} \max_{\pi \in \Pi_{sah}} \frac{w_h^{\pi_{\exp}}(s, a)^{-1}}{(V_0^\star - V_0^\pi)^2 \vee \epsilon^2} \\
&= \inf_{\pi_{\exp}} \max_{s,a} \frac{w_h^{\pi_{\exp}}(s, a)^{-1}}{(V_0^\star - \max_{\pi \in \Pi_{sah}} V_0^\pi)^2 \vee \epsilon^2} \\
&\overset{(c)}{=} \inf_{\pi_{\exp}} \max_{s,a} \frac{w_h^{\pi_{\exp}}(s, a)^{-1}}{\bar{\Delta}_h(s,a)^2 \vee \epsilon^2}
\end{aligned}$$

where $(a)$ follows since $\Pi = \cup_{s,a} \Pi_{sah}$, $(b)$ follows since by definition, for any $\pi \in \Pi_{sah}$, $(s_h^\pi, a_h^\pi) = (s,a)$, and $(c)$ follows by the definition of $\bar{\Delta}_h(s,a)$.

Let $\pi^{sa}$ denote any policy such that $(s_h^\pi, a_h^\pi) = (s,a)$. Set

$$\lambda_{\pi^{sa}} = \frac{\max\{\bar{\Delta}_h(s,a), \epsilon\}^{-2}}{\sum_{s',a'} \max\{\bar{\Delta}_h(s',a'), \epsilon\}^{-2}}.$$

Note that this is a valid distribution. Let $\pi_{\exp} = \sum_{s,a} \lambda_{\pi^{sa}} \pi^{sa}$, then $w_h^{\pi_{\exp}}(s,a) = \lambda_{\pi^{sa}}$, so

$$\begin{aligned}
\inf_{\pi_{\exp}} \max_{s,a} \frac{w_h^{\pi_{\exp}}(s, a)^{-1}}{\bar{\Delta}_h(s,a)^2 \vee \epsilon^2} &\leq \max_{s,a} \frac{\lambda_{\pi^{sa}}^{-1}}{\bar{\Delta}_h(s,a)^2 \vee \epsilon^2} \\
&\leq \sum_{s,a} \frac{1}{\bar{\Delta}_h(s,a)^2 \vee \epsilon^2}
\end{aligned}$$

which proves the result. $\qquad\square$

*Proof of Corollary 4.* As in tabular MDPs, we can set $\Pi$ to correspond to the set of all deterministic policies. However, since our MDP is also deterministic, at any given $h$, we only need to specify $\pi_h(s)$ for a single $s$—the state we will end up in at step $h$ with probability 1. Thus, we can take $\Pi$ to be a set of cardinality $|\Pi| = A^H$. The result then follows directly from Lemma B.13 and Theorem 1. $\quad\square$

**Comparison to Lower Bound of Tirinzoni et al. (2022).** The precise definition for $\bar{\Delta}_{\min}^h$ is $\bar{\Delta}_{\min}^h := \min_{s,a: \bar{\Delta}_h(s,a) > 0} \bar{\Delta}_h(s,a)$ in the setting when every deterministic $\epsilon$-optimal policy will reach the same $(s,a)$ at step $h$, and $\bar{\Delta}_{\min}^h := 0$ otherwise.

The exact lower bound given in Tirinzoni et al. (2022) scales as $\varphi^\star(\underline{c})$ which does not have an explicit form. However, they show that

$$\max_{h \in [H]} \sum_{s \in \mathcal{S}} \sum_{a \in \mathcal{A}} \frac{\log(1/4\delta)}{4 \max\{\bar{\Delta}_h(s,a), \bar{\Delta}_{\min}^h, \epsilon\}^2} \leq \varphi^\star(\underline{c}) \leq \sum_{h \in [H]} \sum_{s \in \mathcal{S}} \sum_{a \in \mathcal{A}} \frac{\log(1/4\delta)}{4 \max\{\bar{\Delta}_h(s,a), \bar{\Delta}_{\min}^h, \epsilon\}^2}.$$

Up to $H$ factors, then, this matches the complexity of our upper bound in every term but the $\bar{\Delta}^h_{\min}$ term. $\bar{\Delta}^h_{\min} \geq \bar{\Delta}_{\min}$, so this lower bound is potentially smaller than our upper bound in this dependence. We remark, however, that the algorithm presented in Tirinzoni et al. (2022) obtains the same scaling as we do, depending on $\bar{\Delta}_{\min}$ instead of $\bar{\Delta}^h_{\min}$. Furthermore, in general we can think of these quantities as scaling in a similar manner, since they each quantify the minimum policy gap.

# C  Experiment Design via Online Frank-Wolfe

## C.1  Experiment Design in MDPs with General Objective Functions

While the experiment design in (5.1) is the natural design if our goal is to identify a near-optimal policy, in general we may be interested in collecting data to minimize some other objective; that is, solving an experiment design of the form:

$$\inf_{\mathbf{\Lambda}_{\exp} \in \mathbf{\Omega}_h} f(\mathbf{\Lambda}_{\exp})$$

for some function $f$ defined over the space of PSD matrices. For example, we could take $f(\mathbf{\Lambda}_{\exp}) = \|\mathbf{\Lambda}_{\exp}^{-1}\|_{\mathrm{op}} = \frac{1}{\lambda_{\min}(\mathbf{\Lambda}_{\exp})}$, and the above experiment design would correspond to maximizing the minimum eigenvalue of the collected covariates, or E-optimal design (Pukelsheim, 2006).

Motivated by this, in this section we generalize Theorem 5 and Algorithm 2 to handle a much broader class of experiment design problems. In particular, we consider all *smooth experiment design objectives*, which we define as follows.

**Definition C.1** (Smooth Experiment Design Objectives). We say that $f(\mathbf{\Lambda}) : \mathbb{S}^d_+ \to \mathbb{R}$ is a *smooth experiment design objective* if it satisfies the following conditions:

- $f$ is convex, differentiable, and $\beta$ smooth in the norm $\|\cdot\|$: $\|\nabla f(\mathbf{\Lambda}) - \nabla f(\mathbf{\Lambda}')\|_* \leq \beta\|\mathbf{\Lambda} - \mathbf{\Lambda}'\|$.
- $f$ is $L$-lipschitz in the operator norm: $|f(\mathbf{\Lambda}) - f(\mathbf{\Lambda}')| \leq L\|\mathbf{\Lambda} - \mathbf{\Lambda}'\|_{\mathrm{op}}$.
- Let $\Xi_{\mathbf{\Lambda}_0} := -\nabla_{\mathbf{\Lambda}} f(\mathbf{\Lambda})|_{\mathbf{\Lambda}=\mathbf{\Lambda}_0}$. Then $\Xi_{\mathbf{\Lambda}_0} \succeq 0$ and $\mathrm{tr}(\Xi_{\mathbf{\Lambda}_0}) \leq M$ for all $\mathbf{\Lambda}_0 \succeq 0$ satisfying $\|\mathbf{\Lambda}_0\|_{\mathrm{op}} \leq 1$.

We will often be interested in objectives $f$ that satisfy $f(a\mathbf{\Lambda}) = a^{-1} f(\mathbf{\Lambda})$ for a scalar $a$, in which case the guarantee $f(N^{-1}\widehat{\mathbf{\Sigma}}_N) \leq N\epsilon$ reduces to $f(\widehat{\mathbf{\Sigma}}_N) \leq \epsilon$. We note also that many typical experiment design objectives are non-smooth. As we show in Appendix D, however, it is often possible to derive smoothed versions of such objectives with negligible approximation error.

Through the remainder of Appendix C as well as Appendix D, we will be interested in the problem of data collection in linear MDPs. In general, we will seek to collect data for a particular $h \in [H]$. We will therefore consider the following truncation to our MDP.

**Definition C.2** (Truncated Horizon MDPs). Given some MDP $\mathcal{M}$ with horizon $H$, we define the $h$-truncated-horizon MDP $\mathcal{M}_{\mathrm{tr},h}$ to be the MDP that is identical to $\mathcal{M}$ for $h' \leq h$, but that terminates after reaching state $s_h$ and playing action $a_h$.

We can simulated a truncated-horizon MDP by playing in our standard MDP $\mathcal{M}$, and after taking an action at step $h$, $a_h$, taking random actions for $h' > h$ and ignoring all future observations.

The utility of considering truncated-horizon MDPs is that we can therefore guarantee the data we collect, $\{\{(s_{h',\tau}, a_{h',\tau})\}_{h'=1}^{h}\}_{\tau=1}^{K}$ is uncorrelated with the true next state and reward at step $h$ obtained in $\mathcal{M}$, $\{(s_{h+1,\tau}, r_{h,\tau})\}_{\tau=1}^{K}$. While we do not allow our algorithm to use $\{(s_{h+1,\tau}, r_{h,\tau})\}_{\tau=1}^{K}$ in its operation, it is allowed to store this data and return it.

For the remainder of Appendix C and Appendix D, then, we assume there is some fixed $h$ we are interested in, and that we are running our algorithms in the $h$-truncated-horizon MDP defined with respect to our original MDP. We will also drop the subscript of $h$ from observations, so $\mathbf{\Lambda}_\pi = \mathbf{\Lambda}_{\pi,h}$, $\phi_\tau = \phi_{\tau,h}$, and $\mathbf{\Omega} = \mathbf{\Omega}_h$.

Our main experiment design algorithm, OPTCOV, relies on a regret-minimization algorithm satisfying the following guarantee.

**Definition C.3** (Regret Minimization Algorithm). We say REGMIN is a regret minimization algorithm if it has regret scaling as, with probability at least $1 - \delta$,

$$\mathcal{R}_K := \sum_{k=1}^{K} (V_0^\star - V_0^{\pi_k}) \leq \sqrt{\mathcal{C}_1 K \log^{p_1}(HK/\delta)} + \mathcal{C}_2 \log^{p_2}(HK/\delta)$$

for any deterministic reward function $r_h(s, a) \in [0, 1]$.

Throughout this section, we will let $\mathbf{\Lambda}$ refer to covariates normalized by time, and $\mathbf{\Sigma}$ unnormalized covariates. So, for example, we might have $\mathbf{\Sigma} = \sum_{\tau=1}^{T} \phi_\tau \phi_\tau^\top$ and $\mathbf{\Lambda} = \frac{1}{T} \sum_{\tau=1}^{T} \phi_\tau \phi_\tau^\top$.

The rest of this section is organized as follows. First, in Appendix C.2 we show that a variant of the Frank-Wolfe algorithm that relies on only approximate updates enjoys a convergence rate similar to the standard Frank-Wolfe rate. Next, in Appendix C.3 we show that for a smooth experiment design objective, we can approximately optimize the objective in a linear MDP by approximating the Frank-Wolfe updates via a regret minimization algorithm. Finally, in Appendix C.4 we present our main experiment-design algorithm, OptCov, which relies on our online Frank-Wolfe procedure to collect covariates that minimize an online experimental design objective up to an arbitrarily tolerance.

## C.2 Approximate Frank-Wolfe

We will consider the following approximate variant of the Frank-Wolfe algorithm:

---

**Algorithm 4** Approximate Frank-Wolfe

1: **input**: function to optimize $f$, number of iterations to run $T$, starting iterate $\boldsymbol{x}_1$
2: **for** $t = 1, 2, \ldots, T$ **do**
3:     Set $\gamma_t \leftarrow \frac{1}{t+1}$
4:     Choose $\boldsymbol{y}_t$ to be any point such that

$$\nabla f(\boldsymbol{x}_t)^\top \boldsymbol{y}_t \le \min_{\boldsymbol{y} \in \mathcal{X}} \nabla f(\boldsymbol{x}_t)^\top \boldsymbol{y} + \epsilon_t$$

5:     $\boldsymbol{x}_{t+1} \leftarrow (1 - \gamma_t)\boldsymbol{x}_t + \gamma_t \boldsymbol{y}_t$
6: **return** $\boldsymbol{x}_{T+1}$

---

**Lemma C.1.** *Consider running Algorithm 4 with some convex function $f$ that is $\beta$-smooth with respect to some norm $\|\cdot\|$, and let $R := \sup_{\boldsymbol{x}, \boldsymbol{y} \in \mathcal{X}} \|\boldsymbol{x} - \boldsymbol{y}\|$. Then for $T \ge 2$, we have*

$$f(\boldsymbol{x}_{T+1}) - \min_{\boldsymbol{x} \in \mathcal{X}} f(\boldsymbol{x}) \le \frac{\beta R^2 (\log T + 1)}{2(T+1)} + \frac{1}{T+1} \sum_{t=1}^{T} \epsilon_t.$$

*Proof.* Let $\boldsymbol{x}^\star = \arg\min_{\boldsymbol{x} \in \mathcal{X}} f(\boldsymbol{x})$. Using that $f$ is $\beta$-smooth, the definition of $\boldsymbol{y}_s$, and the convexity of $f$, we have that for any $s$,

$$
\begin{aligned}
f(\boldsymbol{x}_{s+1}) - f(\boldsymbol{x}_s) &\le \nabla f(\boldsymbol{x}_s)^\top (\boldsymbol{x}_{s+1} - \boldsymbol{x}_s) + \frac{\beta}{2} \|\boldsymbol{x}_{s+1} - \boldsymbol{x}_s\|^2 \\
&\le \gamma_s \nabla f(\boldsymbol{x}_s)^\top (\boldsymbol{y}_s - \boldsymbol{x}_s) + \frac{\beta}{2} \gamma_s^2 R^2 \\
&\le \gamma_s \nabla f(\boldsymbol{x}_s)^\top (\boldsymbol{x}^\star - \boldsymbol{x}_s) + \gamma_s \epsilon_s + \frac{\beta}{2} \gamma_s^2 R^2 \\
&\le \gamma_s (f(\boldsymbol{x}^\star) - f(\boldsymbol{x}_s)) + \gamma_s \epsilon_s + \frac{\beta}{2} \gamma_s^2 R^2.
\end{aligned}
$$

Letting $\delta_s = f(\boldsymbol{x}_s) - f(\boldsymbol{x}^\star)$, this implies that

$$\delta_{s+1} \le (1 - \gamma_s)\delta_s + \gamma_s \epsilon_s + \frac{\beta}{2} \gamma_s^2 R^2.$$

Unrolling this backwards gives

$$
\begin{aligned}
\delta_{T+1} &\le (1 - \gamma_T)\delta_T + \gamma_T \epsilon_T + \frac{\beta}{2} \gamma_T^2 R^2 \\
&\le (1 - \gamma_T)(1 - \gamma_{T-1})\delta_{T-1} + (1 - \gamma_T)(\gamma_{T-1}\epsilon_{T-1} + \frac{\beta}{2}\gamma_{T-1}^2 R^2) + \gamma_T \epsilon_T + \frac{\beta}{2}\gamma_T^2 R^2 \\
&\le \sum_{t=1}^{T} \left( \prod_{s=t+1}^{T} (1 - \gamma_s) \right) (\gamma_t \epsilon_t + \frac{\beta}{2}\gamma_t^2 R^2).
\end{aligned}
$$

We can write

$$\prod_{s=t+1}^{T}(1-\gamma_s) = \prod_{s=t+1}^{T}\frac{s}{s+1} = \frac{t+1}{T+1}$$

so

$$\sum_{t=1}^{T}\left(\prod_{s=t+1}^{T}(1-\gamma_s)\right)\frac{\beta}{2}\gamma_t^2 R^2 = \sum_{t=1}^{T}\frac{t+1}{T+1}\frac{\beta}{2}\frac{1}{(t+1)^2}R^2$$

$$= \frac{\beta R^2}{2(T+1)}\sum_{t=1}^{T}\frac{1}{t+1}$$

$$\leq \frac{\beta R^2(\log T + 1)}{2(T+1)}$$

and

$$\sum_{t=1}^{T}\left(\prod_{s=t+1}^{T}(1-\gamma_s)\right)\gamma_t\epsilon_t = \sum_{t=1}^{T}\frac{t+1}{T+1}\frac{1}{t+1}\epsilon_t$$

$$= \frac{1}{T+1}\sum_{t=1}^{T}\epsilon_t$$

which proves the result.

$\square$

**Lemma C.2.** *When running Algorithm 4, we have*

$$\boldsymbol{x}_{T+1} = \frac{1}{T+1}\left(\sum_{t=1}^{T}\boldsymbol{y}_t + \boldsymbol{x}_1\right).$$

*Proof.* We have:

$$\boldsymbol{x}_{T+1} = \sum_{t=1}^{T}\left(\prod_{s=t+1}^{T}(1-\gamma_s)\right)\gamma_t\boldsymbol{y}_t + \left(\prod_{s=1}^{T}(1-\gamma_s)\right)\boldsymbol{x}_1$$

$$= \sum_{t=1}^{T}\frac{t+1}{T+1}\frac{1}{t+1}\boldsymbol{y}_t + \frac{1}{T+1}\boldsymbol{x}_1$$

$$= \frac{1}{T+1}\sum_{t=1}^{T}\boldsymbol{y}_t + \frac{1}{T+1}\boldsymbol{x}_1.$$

$\square$

### C.3 Online Frank-Wolfe via Regret Minimization

---
**Algorithm 5** Online Frank-Wolfe via Regret Minimization (FWREGRET)

---
1: **input**: function to optimize $f$, number of iterates $T$, episodes per iterate $K$
2: Play any policy for $K$ episodes, denote collected covariates as $\boldsymbol{\Gamma}_0$, collected data as $\mathfrak{D}_0$
3: $\boldsymbol{\Lambda}_1 \leftarrow K^{-1}\boldsymbol{\Gamma}_0$
4: **for** $t = 1, 2, \ldots, T$ **do**
5:     Set $\gamma_t \leftarrow \frac{1}{t+1}$
6:     Run REGMIN on reward $r_h^t(s,a) = \mathrm{tr}(\boldsymbol{\Xi}_{\boldsymbol{\Lambda}_t} \cdot \boldsymbol{\phi}(s,a)\boldsymbol{\phi}(s,a)^\top)/M$ for $K$ episodes, denote collected covariates as $\boldsymbol{\Gamma}_t$, collected data as $\mathfrak{D}_t$
7:     $\boldsymbol{\Lambda}_{t+1} \leftarrow (1-\gamma_t)\boldsymbol{\Lambda}_t + \gamma_t K^{-1}\boldsymbol{\Gamma}_t$
8: **return** $\boldsymbol{\Lambda}_{T+1}, \cup_{t=0}^{T}\mathfrak{D}_t$

---

**Lemma C.3.** *Consider running Algorithm 5 with a function $f$ satisfying Definition C.1 and a regret minimization algorithm satisfying Definition C.3. Denote $K_0(T, \beta, M, \delta)$ the minimum integer value of $K$ satisfying*

$$K \geq \max\left\{ \frac{72T^2M^2\log(4T/\delta)}{\beta^2 R^4}, \frac{8T^2M^2\mathcal{C}_1\log^{p_1}(2HKT/\delta)}{\beta^2 R^4}, \frac{3TM\mathcal{C}_2\log^{p_2}(2HKT/\delta)}{\beta R^2} \right\}.$$

*Then as long as $K \geq K_0(T, \beta, M, \delta)$, we have that, with probability at least $1 - \delta$,*

$$f(\mathbf{\Lambda}_{T+1}) - \inf_{\mathbf{\Lambda}\in\mathbf{\Omega}} f(\mathbf{\Lambda}) \leq \frac{\beta R^2(\log T + 3)}{2(T+1)}$$

*for $R = \sup_{\pi,\pi'} \|\mathbf{\Lambda}_\pi - \mathbf{\Lambda}_{\pi'}\|$.*

*Proof.* Note that by Lemma C.2 and since $\|\phi(s,a)\|_2 \leq 1$, we can bound $\|\mathbf{\Lambda}_t\|_{\mathrm{op}} \leq 1$ and $\|\phi(s,a)\phi(s,a)^\top\|_{\mathrm{op}} \leq 1$. Definition C.1 it follows that $r_h^t(s,a) \in [0,1]$ for all $s,a$, since $\mathrm{tr}(\Xi_{\mathbf{\Lambda}_t} \cdot \phi(s,a)\phi(s,a)^\top) \leq \|\phi(s,a)\phi(s,a)^\top\|_{\mathrm{op}} \cdot \mathrm{tr}(\Xi_{\mathbf{\Lambda}_t}) \leq \mathrm{tr}(\Xi_{\mathbf{\Lambda}_t}) \leq M$, and $\mathrm{tr}(\Xi_{\mathbf{\Lambda}_t} \cdot \phi(s,a)\phi(s,a)^\top) \geq 0$ since $\Xi_{\mathbf{\Lambda}_t} \succeq 0$. If we run REGMIN for $K$ episodes on reward function $r_h^t$, by Definition C.1 and Definition C.3 we then have that, with probability at least $1 - \delta/2T$,

$$\sqrt{\mathcal{C}_1 K\log^{p_1}(2HKT/\delta)} + \mathcal{C}_2\log^{p_2}(2HKT/\delta) \geq K\sup_\pi \mathbb{E}_\pi[\mathrm{tr}(\Xi_{\mathbf{\Lambda}_t} \cdot \phi\phi^\top)/M] - \sum_{k=1}^K \mathbb{E}_{\pi_k}[\mathrm{tr}(\Xi_{\mathbf{\Lambda}_t} \cdot \phi\phi^\top)/M]$$

$$= K\sup_\pi \mathrm{tr}(\Xi_{\mathbf{\Lambda}_t}\mathbf{\Lambda}_\pi)/M - K\mathrm{tr}(\Xi_{\mathbf{\Lambda}_t} \cdot K^{-1}\sum_{k=1}^K \mathbf{\Lambda}_{\pi_k})/M$$

which implies

$$\sqrt{\frac{M^2\mathcal{C}_1\log^{p_1}(2HKT/\delta)}{K}} + \frac{M\mathcal{C}_2\log^{p_2}(2HKT/\delta)}{K} \geq \sup_\pi \mathrm{tr}(\Xi_{\mathbf{\Lambda}_t}\mathbf{\Lambda}_\pi) - \mathrm{tr}(\Xi_{\mathbf{\Lambda}_t} \cdot K^{-1}\sum_{k=1}^K \mathbf{\Lambda}_{\pi_k}).$$

Furthermore, we have that

$$\left| \mathrm{tr}(\Xi_{\mathbf{\Lambda}_t} \cdot K^{-1}\sum_{k=1}^K \mathbf{\Lambda}_{\pi_k}) - \mathrm{tr}(\Xi_{\mathbf{\Lambda}_t} \cdot K^{-1}\mathbf{\Gamma}_t) \right| = \left| \frac{1}{K}\sum_{k=1}^K \mathrm{tr}(\Xi_{\mathbf{\Lambda}_t}\mathbf{\Lambda}_{\pi_k}) - \frac{1}{K}\sum_{k=1}^K \mathrm{tr}(\Xi_{\mathbf{\Lambda}_t}\phi_k\phi_k^\top) \right|.$$

Note that $\mathbb{E}_{\pi_k}[\mathrm{tr}(\Xi_{\mathbf{\Lambda}_t}\phi_k\phi_k^\top)] = \mathrm{tr}(\Xi_{\mathbf{\Lambda}_t}\mathbf{\Lambda}_{\pi_k})$, $\mathrm{tr}(\Xi_{\mathbf{\Lambda}_t}\phi_k\phi_k^\top) \in [0, M]$, and $\pi_k$ is $\mathcal{F}_{k-1}$-measurable. We can therefore apply Azuma-Hoeffding (Lemma A.4) to get that, with probability at least $1 - \delta/2T$,

$$\left| \mathrm{tr}(\Xi_{\mathbf{\Lambda}_t} \cdot K^{-1}\sum_{k=1}^K \mathbf{\Lambda}_{\pi_k}) - \mathrm{tr}(\Xi_{\mathbf{\Lambda}_t} \cdot K^{-1}\mathbf{\Gamma}_t) \right| \leq \sqrt{\frac{8M^2\log(4T/\delta)}{K}}.$$

Therefore,

$$\sqrt{\frac{8M^2\log(4T/\delta)}{K}} + \sqrt{\frac{M^2\mathcal{C}_1\log^{p_1}(2HKT/\delta)}{K}} + \frac{M\mathcal{C}_2\log^{p_2}(2HKT/\delta)}{K}$$
$$\geq \sup_\pi \mathrm{tr}(\Xi_{\mathbf{\Lambda}_t}\mathbf{\Lambda}_\pi) - \mathrm{tr}(\Xi_{\mathbf{\Lambda}_t} \cdot K^{-1}\mathbf{\Gamma}_t).$$

Given our condition on $K$, we have

$$\sqrt{\frac{8M^2\log(4T/\delta)}{K}} + \sqrt{\frac{M^2\mathcal{C}_1\log^{p_1}(2HKT/\delta)}{K}} + \frac{M\mathcal{C}_2\log^{p_2}(2HKT/\delta)}{K} \leq \frac{\beta R^2}{T}$$

which implies

$$\sup_\pi \mathrm{tr}(\Xi_{\mathbf{\Lambda}_t}\mathbf{\Lambda}_\pi) - \mathrm{tr}(\Xi_{\mathbf{\Lambda}_t} \cdot K^{-1}\mathbf{\Gamma}_t) \leq \frac{\beta R^2}{T}. \tag{C.1}$$

Note that, for any $\mathbf{\Lambda} \in \mathbf{\Omega}$, we have

$$\mathrm{tr}(\Xi_{\mathbf{\Lambda}_t}\mathbf{\Lambda}) = \mathrm{tr}(\Xi_{\mathbf{\Lambda}_t}\mathbb{E}_{\pi\sim\omega}[\mathbf{\Lambda}_\pi]) = \mathbb{E}_{\pi\sim\omega}[\mathrm{tr}(\Xi_{\mathbf{\Lambda}_t}\mathbf{\Lambda}_\pi)]$$

so

$$\sup_{\boldsymbol{\Lambda} \in \boldsymbol{\Omega}} \operatorname{tr}(\Xi_{\boldsymbol{\Lambda}_t} \boldsymbol{\Lambda}) = \sup_{\omega \in \boldsymbol{\Omega}_\pi} \mathbb{E}_{\pi \sim \omega}[\operatorname{tr}(\Xi_{\boldsymbol{\Lambda}_t} \boldsymbol{\Lambda}_\pi)] = \sup_\pi \operatorname{tr}(\Xi_{\boldsymbol{\Lambda}_t} \boldsymbol{\Lambda}_\pi)$$

By definition, $\Xi_{\boldsymbol{\Lambda}_t} = -\nabla_{\boldsymbol{\Lambda}} f(\boldsymbol{\Lambda})|_{\boldsymbol{\Lambda}=\boldsymbol{\Lambda}_t}$, so it follows that

$$-\sup_{\boldsymbol{\Lambda}' \in \boldsymbol{\Omega}} \operatorname{tr}(\Xi_{\boldsymbol{\Lambda}_t} \boldsymbol{\Lambda}') = \inf_{\boldsymbol{\Lambda}' \in \boldsymbol{\Omega}} \operatorname{tr}(\nabla_{\boldsymbol{\Lambda}} f(\boldsymbol{\Lambda})|_{\boldsymbol{\Lambda}=\boldsymbol{\Lambda}_t} \cdot \boldsymbol{\Lambda}').$$

It follows that (C.1) is precisely the guarantee required on $\boldsymbol{y}_t$ by Algorithm 4 with $\epsilon_t = \frac{\beta R^2}{T}$. Since $f$ is $\beta$-smooth by Definition C.1 and since the set $\boldsymbol{\Omega}$ is convex and compact by Lemma A.9, we can apply Lemma C.1 with a union bound over $t$ to get the result. $\qquad\square$

## C.4 Data Collection via Online Frank-Wolfe

---
**Algorithm 6** Collect Optimal Covariates (OPTCOV)

---
1: **input**: functions to optimize $(f_i)_i$, constraint tolerance $\epsilon$, confidence $\delta$
2: **for** $i = 1, 2, 3, \ldots$ **do**
3: $\quad$ $T_i \leftarrow 2^i$, $K_i \leftarrow 2^i T_i^2$
4: $\quad$ **if** $K_i \geq \widetilde{K}_0(T_i, \beta_i, M_i, \frac{\delta}{4i^2}) T_i^2 + \widetilde{K}_1(T_i, \beta_i, M_i, \frac{\delta}{4i^2}) T_i$ for $\widetilde{K}_0$ and $\widetilde{K}_1$ as in Lemma C.5
$\quad$ **then**
5: $\quad\quad$ $\widehat{\boldsymbol{\Lambda}}, \mathfrak{D}_i \leftarrow$ FWREGRET$(f_i, T_i - 1, K_i)$
6: $\quad\quad$ **if** $f_i(\widehat{\boldsymbol{\Lambda}}) \leq K_i T_i \epsilon$ and $f_i(\widehat{\boldsymbol{\Lambda}}) \geq \frac{\beta_i R^2 (\log T_i + 3)}{T_i}$ **then**
7: $\quad\quad\quad$ **return** $\widehat{\boldsymbol{\Lambda}}, K_i T_i, \mathfrak{D}_i$

---

**Theorem 7.** *Let $(f_i)_i$ denote some sequence of functions which satisfy Definition C.1 with constants $(\beta_i, L_i, M_i)$ and assume $\beta_i \geq 1$. Let $(\beta, L, M)$ be some values such that $\beta_i \leq \beta$, $L_i \leq L$, $M_i \leq M$ for al $i$, and let $f$ be some function such that $f_i(\boldsymbol{\Lambda}) \leq f(\boldsymbol{\Lambda})$ for all $i$ and $\boldsymbol{\Lambda} \succeq 0$. Denote $f_{\min}$ a lower bound on all $f_i$: $\min_i \inf_{\boldsymbol{\Lambda} \in \boldsymbol{\Omega}} f_i(\boldsymbol{\Lambda}) \geq f_{\min}$.*

*Define*

$$N^\star(\epsilon; f) := \frac{\inf_{\boldsymbol{\Lambda} \in \boldsymbol{\Omega}} f(\boldsymbol{\Lambda})}{\epsilon}. \tag{C.2}$$

*Then, if we run Algorithm 6 on $(f_i)_i$ with constraint tolerance $\epsilon$ and confidence $\delta$, we have that with probability at least $1 - \delta$, it will run for at most*

$$5N^\star(\epsilon; f) + \operatorname{poly}\left(2^{p_1 + p_2}, \mathcal{C}_1, \mathcal{C}_2, M, \beta, R, L, f_{\min}^{-1}, \log 1/\delta\right)$$

*episodes, and will return data $\{\boldsymbol{\phi}_\tau\}_{\tau=1}^N$ with covariance $\widehat{\boldsymbol{\Sigma}}_N = \sum_{\tau=1}^N \boldsymbol{\phi}_\tau \boldsymbol{\phi}_\tau^\top$ such that*

$$f_{\widehat{i}}(N^{-1}\widehat{\boldsymbol{\Sigma}}_N) \leq N\epsilon,$$

*where $\widehat{i}$ is the iteration on which OPTCOV terminates.*

**Corollary 6.** *Instantiating REGMIN with the computationally efficient version of the FORCE algorithm of [Wagenmaker et al. (2021a)](#), we obtain a complexity of*

$$5N^\star(\epsilon; f) + \operatorname{poly}\left(d, H, M, \beta, R, L, f_{\min}^{-1}, \log 1/\delta\right).$$

*Proof.* This result is immediate since FORCE satisfies Definition C.3 with

$$\mathcal{C}_1 = c_1 d^4 H^4, \quad \mathcal{C}_2 = c_2 d^4 H^3, \quad p_1 = 3, \quad p_2 = 7/2$$

for universal numerical constants $c_1$ and $c_2$. $\qquad\square$

*Proof of Theorem 8.* We first show that the condition $f_i(\widehat{\boldsymbol{\Lambda}}) \geq \frac{\beta R^2 (\log T_i + 3)}{T_i}$ is sufficient to ensure a 2-approximate minimum of $f_i$, and then show a sufficient condition on $K_i$ and $T_i$ that will guarantee the condition on Line 6 is met.

**Guaranteeing 2-optimality.** We first show that for a fixed $i$, the condition $f_i(\widehat{\mathbf{\Lambda}}) \geq \frac{\beta_i R^2(\log T_i + 3)}{T_i}$ will only be met once

$$f_i(\widehat{\mathbf{\Lambda}}) \leq 2 \cdot \inf_{\mathbf{\Lambda} \in \mathbf{\Omega}} f_i(\mathbf{\Lambda})$$

and that it will take at most

$$T_i \geq \frac{2\beta R^2(\log T_i + 3)}{\inf_{\mathbf{\Lambda} \in \mathbf{\Omega}} f_i(\mathbf{\Lambda})}$$

iterations to do so, as long as

$$T_i K_i \geq \frac{L^2}{2(d\log(1 + 8\sqrt{T_i K_i}) + \log(4i^2/\delta)) \cdot (\inf_{\mathbf{\Lambda} \in \mathbf{\Omega}} f_i(\mathbf{\Lambda}))^2}.$$

The first part follows by applying Lemma C.3. Note that the if statement on Line 4 will only be met once

$$K_i \geq K_0(T_i, \beta_i, M_i, \delta/4i^2).$$

This follows by Lemma C.5. Thus, the condition on $K_i$ required by Lemma C.3 will be met, so it follows that with probability at least $1 - \delta/(4i^2)$,

$$f_i(\widehat{\mathbf{\Lambda}}) - \inf_{\mathbf{\Lambda} \in \mathbf{\Omega}} f_i(\mathbf{\Lambda}) \leq \frac{\beta_i R^2(\log T_i + 3)}{2T_i}.$$

Therefore, if $f_i(\widehat{\mathbf{\Lambda}}) \geq \frac{\beta_i R^2(\log T_i + 3)}{T_i}$, we have

$$f_i(\widehat{\mathbf{\Lambda}}) - \inf_{\mathbf{\Lambda} \in \mathbf{\Omega}} f_i(\mathbf{\Lambda}) \leq \frac{1}{2} f_i(\widehat{\mathbf{\Lambda}}) \implies \frac{1}{2} f_i(\widehat{\mathbf{\Lambda}}) \leq \inf_{\mathbf{\Lambda} \in \mathbf{\Omega}} f_i(\mathbf{\Lambda})$$

$$\implies f_i(\widehat{\mathbf{\Lambda}}) \leq 2 \cdot \inf_{\mathbf{\Lambda} \in \mathbf{\Omega}} f_i(\mathbf{\Lambda}).$$

We will show a sufficient condition for $f_i(\widehat{\mathbf{\Lambda}}) \geq \frac{\beta R^2(\log T_i + 3)}{T_i}$, which implies that $f_i(\widehat{\mathbf{\Lambda}}) \geq \frac{\beta_i R^2(\log T_i + 3)}{T_i}$ since $\beta_i \leq \beta$. By Lemma C.2 and the procedure run by Algorithm 5, we have that $\widehat{\mathbf{\Lambda}} = \frac{1}{T_i K_i} \sum_{\tau=1}^{T_i K_i} \boldsymbol{\phi}_\tau \boldsymbol{\phi}_\tau^\top$ where at episodes $\tau$ we run some $\mathcal{F}_{\tau-1}$-measurable policy $\pi_\tau$ to acquire $\boldsymbol{\phi}_\tau$. Now if $\widehat{\mathbf{\Lambda}} = \widetilde{\mathbf{\Lambda}}$ for some $\widetilde{\mathbf{\Lambda}} \in \mathbf{\Omega}$, then the second part follows trivially since $\inf_{\mathbf{\Lambda} \in \mathbf{\Omega}} f_i(\mathbf{\Lambda}) \leq f_i(\widetilde{\mathbf{\Lambda}})$, so a sufficient condition for $f_i(\widehat{\mathbf{\Lambda}}) \geq \frac{\beta R^2(\log T_i + 3)}{T_i}$ is that $\inf_{\mathbf{\Lambda} \in \mathbf{\Omega}} f_i(\mathbf{\Lambda}) \geq \frac{\beta R^2(\log T_i + 3)}{T_i}$. However, since $\widehat{\mathbf{\Lambda}}$ is stochastic, we may not have that $\widehat{\mathbf{\Lambda}} \in \mathbf{\Omega}$. Let $\widetilde{\mathbf{\Lambda}} := \frac{1}{T_i K_i} \sum_{\tau=1}^{T_i K_i} \mathbf{\Lambda}_{\pi_\tau}$ and note that $\widetilde{\mathbf{\Lambda}} \in \mathbf{\Omega}$. Applying Lemma C.4, we have that with probability at least $1 - \delta/(4i^2)$,

$$\left\| \widetilde{\mathbf{\Lambda}} - \widehat{\mathbf{\Lambda}} \right\|_{\text{op}} \leq \sqrt{\frac{8d\log(1 + 8\sqrt{T_i K_i}) + 8\log(4i^2/\delta)}{T_i K_i}}$$

for $\widetilde{\pi}$ the uniform mixture of $\{\pi_\tau\}_{\tau=1}^{T_i K_i}$. By the Lipschitz condition of Definition C.1, this implies

$$\begin{aligned}
f_i(\widehat{\mathbf{\Lambda}}) &\geq f_i(\widetilde{\mathbf{\Lambda}}) - L_i \|\widehat{\mathbf{\Lambda}} - \widetilde{\mathbf{\Lambda}}\|_{\text{op}} \\
&\geq f_i(\widetilde{\mathbf{\Lambda}}) - L\|\widehat{\mathbf{\Lambda}} - \widetilde{\mathbf{\Lambda}}\|_{\text{op}} \\
&\geq f_i(\widetilde{\mathbf{\Lambda}}) - L\sqrt{\frac{8d\log(1 + 8\sqrt{T_i K_i}) + 8\log(4i^2/\delta)}{T_i K_i}} \\
&\geq \inf_{\mathbf{\Lambda} \in \mathbf{\Omega}} f_i(\mathbf{\Lambda}) - L\sqrt{\frac{8d\log(1 + 8\sqrt{T_i K_i}) + 8\log(4i^2/\delta)}{T_i K_i}}.
\end{aligned}$$

Thus, a sufficient condition for $f_i(\widehat{\mathbf{\Lambda}}) \geq \frac{\beta R^2(\log T_i + 3)}{T_i}$ is that

$$\inf_{\mathbf{\Lambda} \in \mathbf{\Omega}} f_i(\mathbf{\Lambda}) - L\sqrt{\frac{8d\log(1 + 8\sqrt{T_i K_i}) + 8\log(4i^2/\delta)}{T_i K_i}} \geq \frac{\beta R^2(\log T_i + 3)}{T_i}$$

$$\iff T_i \geq \frac{\beta R^2(\log T_i + 3)}{\inf_{\mathbf{\Lambda} \in \mathbf{\Omega}} f_i(\mathbf{\Lambda}) - L\sqrt{\frac{8d\log(1+8\sqrt{T_i K_i}) + 8\log(4i^2/\delta)}{T_i K_i}}}.$$

If

$$T_i K_i \geq \frac{L^2}{2(d\log(1+8\sqrt{T_i K_i}) + \log(4i^2/\delta)) \cdot (\inf_{\mathbf{\Lambda} \in \mathbf{\Omega}} f_i(\mathbf{\Lambda}))^2}$$

it follows that a sufficient condition is

$$T_i \geq \frac{2\beta R^2(\log T_i + 3)}{\inf_{\mathbf{\Lambda} \in \mathbf{\Omega}} f_i(\mathbf{\Lambda})}.$$

Union bounding over the events considered above for all $i$, we have that the total probability of failure is bounded as

$$\sum_{i=1}^{\infty} \left( \frac{\delta}{4i^2} + \frac{\delta}{4i^2} \right) = \frac{\pi^2}{12}\delta \leq \delta.$$

**Termination Guarantee.** We next show a sufficient condition to ensure that the if statements on Line 4 and Line 6 are met.

Assume the if statement on Line 4 has been met and that we are in the regime where

$$T_i K_i \geq \frac{L^2}{2(d\log(1+8\sqrt{T_i K_i}) + \log(4i^2/\delta)) \cdot f_{\min}^2}, \quad T_i \geq \frac{2\beta R^2(\log T_i + 3)}{f_{\min}}. \tag{C.3}$$

By the argument above and since $\inf_{\mathbf{\Lambda} \in \mathbf{\Omega}} f_i(\mathbf{\Lambda}) \geq f_{\min}$, these conditions are sufficient to guarantee a 2-optimal solutions has been found, that is,

$$f_i(\widehat{\mathbf{\Lambda}}) \leq 2 \cdot \inf_{\mathbf{\Lambda} \in \mathbf{\Omega}} f_i(\mathbf{\Lambda}),$$

and that the condition $f_i(\widehat{\mathbf{\Lambda}}) \geq \frac{\beta R^2(\log T_i + 3)}{T_i}$ has been met. Thus, if (C.3) holds, a sufficient condition for $f_i(\widehat{\mathbf{\Lambda}}) \leq T_i K_i \epsilon$ is

$$2 \cdot \inf_{\mathbf{\Lambda} \in \mathbf{\Omega}} f_i(\mathbf{\Lambda}) \leq T_i K_i \epsilon.$$

It follows that this condition will be met (assuming (C.3) holds) once $T_i K_i \geq N^\star(\frac{\epsilon}{2}; f_i)$. Since $f_i \leq f$, $N^\star(\frac{\epsilon}{2}; f_i) \leq N^\star(\frac{\epsilon}{2}; f)$, so a sufficient condition is that $T_i K_i \geq N^\star(\frac{\epsilon}{2}; f)$.

To upper bound the total complexity, it suffices then to guarantee that we run for enough epochs so that

$$K_i = 2^{3i} \geq \widetilde{K}_0(T_i, \beta_i, M_i, \frac{\delta}{4i^2})T_i^2 + \widetilde{K}_1(T_i, \beta_i, M_i, \frac{\delta}{4i^2})T_i \tag{C.4}$$

$$T_i K_i = 2^{4i} \geq \frac{L^2}{2(d\log(1+8\sqrt{T_i K_i}) + \log(4i^2/\delta)) \cdot f_{\min}^2} \tag{C.5}$$

$$T_i = 2^i \geq \frac{2\beta R^2(\log T_i + 3)}{f_{\min}} \tag{C.6}$$

$$T_i K_i = 2^{4i} \geq N^\star(\frac{\epsilon}{2}; f). \tag{C.7}$$

Here (C.4) guarantees the if statement on Line 4 is met, and (C.5)-(C.7) guarantee the if statement on line Line 6 is met.

By assumption, $M_i \leq M$ and $\beta_i \geq 1$, and note that $\widetilde{K}_0(T_i, \beta_i, M_i, \frac{\delta}{4i^2})$ and $\widetilde{K}_1(T_i, \beta_i, M_i, \frac{\delta}{4i^2})$ are both increasing in $M_i$ and decreasing in $\beta_i$. Thus, a sufficient condition to ensure (C.4) is met is

$$2^{3i} \geq \widetilde{K}_0(2^i, 1, M, \frac{\delta}{4i^2})2^{2i} + \widetilde{K}_1(2^i, 1, M, \frac{\delta}{4i^2})2^i. \tag{C.8}$$

Some calculation shows that

$$\widetilde{K}_0(2^i, 1, M, \frac{\delta}{4i^2}) \leq (5i)^{p_1}\widetilde{K}_0(2, 1, M, \frac{\delta}{4}), \quad \widetilde{K}_1(2^i, 1, M, \frac{\delta}{4i^2}) \leq (4i)^{p_2}\widetilde{K}_1(2, 1, M, \frac{\delta}{4})$$

so a sufficient condition to meet (C.8) is

$$2^i \geq 2(5i)^{p_1} \widetilde{K}_0(2, 1, M, \frac{\delta}{4}), \quad 2^{2i} \geq 2(4i)^{p_2} \widetilde{K}_1(2, 1, M, \frac{\delta}{4}).$$

By Lemma A.2 and some calculation, this will be met once

$$i \geq \max \left\{ 4p_1 \log_2(2p_1) + 2 \log_2(2(5)^{p_1} \widetilde{K}_0(2, 1, M, \frac{\delta}{4})), 2p_2 \log_2(p_2) + 2 \log_2(2(4)^{p_2} \widetilde{K}_1(2, 1, M, \frac{\delta}{4})) \right\} =: i_0.$$

To meet (C.5) it suffices to take

$$i \geq \frac{1}{4} \log_2 \frac{L^2}{df_{\min}^2} =: i_1$$

By Lemma A.2, a sufficient condition to meet (C.6) is that

$$T_i \geq \max \left\{ \frac{6\beta R^2}{f_{\min}}, \frac{4\beta R^2}{f_{\min}} \log \frac{4\beta R^2}{f_{\min}} \right\}$$

so it suffices that

$$i \geq \log_2 \left( \frac{6\beta R^2}{f_{\min}} \log \frac{4\beta R^2}{f_{\min}} \right) =: i_2.$$

Finally, to meet (C.7), it suffices that

$$i \geq \frac{1}{4} \log_2 N^\star(\epsilon/2; f) =: i_3.$$

If we terminate at epoch $\widehat{i}$, the total sample complexity will be bounded by

$$\sum_{i=1}^{\widehat{i}} T_i K_i = \sum_{i=1}^{\widehat{i}} 2^{4i} \leq \frac{16}{15} \cdot 2^{4\widehat{i}}.$$

By the above argument, we can bound $\widehat{i} \leq \lceil \max\{i_0, i_1, i_2, i_3\} \rceil$. Furthermore, we see that

$$2^{4\lceil i_0 \rceil} = \mathrm{poly}\left( 2^{p_1}, 2^{p_2}, M, \mathcal{C}_1, \mathcal{C}_2, \log 1/\delta \right)$$
$$2^{4\lceil i_1 \rceil} = \mathrm{poly}(L, f_{\min}^{-1})$$
$$2^{4\lceil i_2 \rceil} = \mathrm{poly}(\beta, R, f_{\min}^{-1})$$
$$2^{4\lceil i_3 \rceil} \leq 2N^\star(\epsilon/2; f)$$

so we can bound the total sample complexity by

$$\frac{16}{15} \cdot 2^{4\lceil \max\{i_0, i_1, i_2, i_3\} \rceil} \leq \frac{32}{15} N^\star(\epsilon/2; f) + \mathrm{poly}\left( 2^{p_1}, 2^{p_2}, \beta, R, L, f_{\min}^{-1}, M, \mathcal{C}_1, \mathcal{C}_2, \log 1/\delta \right).$$

This completes the proof since $N^\star(\frac{\epsilon}{2}; f) = 2N^\star(\epsilon; f)$ and since, by Lemma C.2, $\widehat{\Lambda}$ is simply the average of the observed feature vectors: $\widehat{\Lambda} = \frac{1}{T_i K_i} \sum_{\tau=1}^{T_i K_i} \phi_\tau \phi_\tau^\top$.

$\square$

**Lemma C.4.** *Let $\Lambda_K$ denote the time-normalized covariates obtained by playing policies $\{\pi_k\}_{k=1}^K$, where $\pi_k$ is $\mathcal{F}_{k-1}$-measurable. Then, with probability at least $1 - \delta$,*

$$\left\| \frac{1}{K} \sum_{k=1}^K \Lambda_{\pi_k} - \Lambda_K \right\|_{\mathrm{op}} \leq \sqrt{\frac{8d \log(1 + 8\sqrt{K}) + 8 \log 1/\delta}{K}}.$$

*Proof.* Let $\mathcal{V}$ denote an $\epsilon$-net of $\mathcal{S}^{d-1}$, for some $\epsilon$ to be chosen. Then,

$$\left\| \frac{1}{K} \sum_{k=1}^K \Lambda_{\pi_k} - \Lambda_K \right\|_{\mathrm{op}} = \sup_{v \in \mathcal{S}^{d-1}} \left| v^\top \left( \frac{1}{K} \sum_{k=1}^K \Lambda_{\pi_k} - \Lambda_K \right) v \right|$$

$$\leq \sup_{\widetilde{\boldsymbol{v}} \in \mathcal{V}} \underbrace{\left| \widetilde{\boldsymbol{v}}^\top \left( \frac{1}{K} \sum_{k=1}^{K} \boldsymbol{\Lambda}_{\pi_k} - \boldsymbol{\Lambda}_K \right) \widetilde{\boldsymbol{v}} \right|}_{(a)}$$

$$+ \underbrace{\sup_{\boldsymbol{v} \in \mathcal{S}^{d-1}} \inf_{\widetilde{\boldsymbol{v}} \in \mathcal{V}} \left| \boldsymbol{v}^\top \left( \frac{1}{K} \sum_{k=1}^{K} \boldsymbol{\Lambda}_{\pi_k} - \boldsymbol{\Lambda}_K \right) \boldsymbol{v} - \widetilde{\boldsymbol{v}}^\top \left( \frac{1}{K} \sum_{k=1}^{K} \boldsymbol{\Lambda}_{\pi_k} - \boldsymbol{\Lambda}_K \right) \widetilde{\boldsymbol{v}} \right|}_{(b)}.$$

Via a union bound over $\mathcal{V}$ and application of Azuma-Hoeffding, we can bound, with probability at least $1 - \delta$,

$$(a) \leq \sqrt{\frac{2 \log |\mathcal{V}|/\delta}{K}}.$$

We can bound $(b)$ as

$$(b) \leq \sup_{\boldsymbol{v} \in \mathcal{S}^{d-1}} \inf_{\widetilde{\boldsymbol{v}} \in \mathcal{V}} 2 \left| \boldsymbol{v}^\top \left( \frac{1}{K} \sum_{k=1}^{K} \boldsymbol{\Lambda}_{\pi_k} - \boldsymbol{\Lambda}_K \right) (\boldsymbol{v} - \widetilde{\boldsymbol{v}}) \right|$$

$$\leq \sup_{\boldsymbol{v} \in \mathcal{S}^{d-1}} \inf_{\widetilde{\boldsymbol{v}} \in \mathcal{V}} 2 \|\boldsymbol{v} - \widetilde{\boldsymbol{v}}\|_2 \left\| \frac{1}{K} \sum_{k=1}^{K} \boldsymbol{\Lambda}_{\pi_k} - \boldsymbol{\Lambda}_K \right\|_{\mathrm{op}}$$

$$\leq 4\epsilon$$

where the last inequality follows since $\|\frac{1}{K} \sum_{k=1}^{K} \boldsymbol{\Lambda}_{\pi_k}\|_{\mathrm{op}} \leq 1$, and $\|\frac{1}{K} \boldsymbol{\Lambda}_K\|_{\mathrm{op}} \leq 1$, and since $\mathcal{V}$ is an $\epsilon$-net. Setting $\epsilon = 1/(4\sqrt{K})$, Lemma A.1 gives that $|\mathcal{V}| \leq (1 + 8\sqrt{K})^d$, and we conclude that with probability at least $1 - \delta$:

$$\left\| \frac{1}{K} \sum_{k=1}^{K} \boldsymbol{\Lambda}_{\pi_k} - \boldsymbol{\Lambda}_K \right\|_{\mathrm{op}} \leq \sqrt{\frac{2 \log |\mathcal{V}|/\delta}{K}} + 4\epsilon$$

$$\leq \sqrt{\frac{2d \log(1 + 8\sqrt{K}) + 2 \log 1/\delta}{K}} + \frac{1}{\sqrt{K}}$$

$$\leq 2\sqrt{\frac{2d \log(1 + 8\sqrt{K}) + 2 \log 1/\delta}{K}}.$$

$\square$

**Lemma C.5.** *We can bound*
$$K_0(T, \beta, M, \delta) \leq \widetilde{K}_0(T, \beta, M, \delta)T^2 + \widetilde{K}_1(T, \beta, M, \delta)T$$

*for*

$$\widetilde{K}_0(T, \beta, M, \delta) := \max \left\{ \frac{72M^2 \log(4T/\delta)}{\beta^2 R^4}, \frac{8M^2 \mathcal{C}_1}{\beta^2 R^4} \cdot (2p_1)^{p_1} \log^{p_1} \left( \frac{32p_1 H T^3 M^2 \mathcal{C}_1}{\beta^2 R^4 \delta} \right) \right\}$$

$$\widetilde{K}_1(T, \beta, M, \delta) := \frac{3M \mathcal{C}_2}{\beta R^2} \cdot (2p_2)^{p_2} \log^{p_2} \left( \frac{12p_2 H T^2 M \mathcal{C}_2}{\beta R^2 \delta} \right),$$

*Proof.* By definition $K_0(T, \beta, M, \delta)$ is the smallest integer value of $K$ that satisfies:

$$K \geq \max \left\{ \frac{72T^2 M^2 \log(4T/\delta)}{\beta^2 R^4}, \frac{8T^2 M^2 \mathcal{C}_1 \log^{p_1}(2HKT/\delta)}{\beta^2 R^4}, \frac{3TM \mathcal{C}_2 \log^{p_2}(2HKT/\delta)}{\beta R^2} \right\}.$$
(C.9)

By Lemma A.2, we have that if

$$K \geq \frac{8T^2 M^2 \mathcal{C}_1}{\beta^2 R^4} \cdot (2p_1)^{p_1} \log^{p_1} \left( \frac{8T^2 M^2 \mathcal{C}_1}{\beta^2 R^4} \cdot \frac{4p_1 H T}{\delta} \right), \quad K \geq \frac{3TM \mathcal{C}_2}{\beta R^2} \cdot (2p_2)^{p_2} \log^{p_2} \left( \frac{3TM \mathcal{C}_2}{\beta R^2} \cdot \frac{4p_2 H T}{\delta} \right)$$

and

$$K \geq \frac{72T^2 M^2 \log(4T/\delta)}{\beta^2 R^4}$$

then Equation (C.9) will be satisfied. Some algebra gives the result. $\square$

# D  XY-Optimal Design

We are interested in optimizing the function

$$\mathsf{XY}_{\mathrm{opt}}(\boldsymbol{\Lambda}) = \max_{\phi \in \Phi} \|\phi\|^2_{\mathbf{A}(\boldsymbol{\Lambda})^{-1}} \quad \text{for} \quad \mathbf{A}(\boldsymbol{\Lambda}) = \boldsymbol{\Lambda} + \boldsymbol{\Lambda}_0$$

with $\boldsymbol{\Lambda}_0 \succ 0$ some fixed regularizer. This objective, however, is not smooth, so we relax it to the following:

$$\widetilde{\mathsf{XY}}_{\mathrm{opt}}(\boldsymbol{\Lambda}) := \mathrm{LogSumExp}\left(\{e^{\eta \|\phi\|^2_{\mathbf{A}(\boldsymbol{\Lambda})^{-1}}}\}_{\phi \in \Phi}; \eta\right) = \frac{1}{\eta} \log\left(\sum_{\phi \in \Phi} e^{\eta \|\phi\|^2_{\mathbf{A}(\boldsymbol{\Lambda})^{-1}}}\right). \tag{D.1}$$

We first offer some properties on how well $\widetilde{\mathsf{XY}}_{\mathrm{opt}}(\boldsymbol{\Lambda})$ approximates $\mathsf{XY}_{\mathrm{opt}}(\boldsymbol{\Lambda})$, and then show that we can bound the smoothness constant of $\widetilde{\mathsf{XY}}_{\mathrm{opt}}(\boldsymbol{\Lambda})$. Throughout this section, we will denote $\gamma_\Phi := \max_{\phi \in \Phi} \|\phi\|_2$ and let $f(\boldsymbol{\Lambda}) := \widetilde{\mathsf{XY}}_{\mathrm{opt}}(\boldsymbol{\Lambda})$.

## D.1  Approximating Non-Smooth Optimal Design with Smooth Optimal Design

**Lemma D.1.**

$$|\mathsf{XY}_{\mathrm{opt}}(\boldsymbol{\Lambda}) - \widetilde{\mathsf{XY}}_{\mathrm{opt}}(\boldsymbol{\Lambda})| \leq \frac{\log |\Phi|}{\eta}, \qquad \mathsf{XY}_{\mathrm{opt}}(\boldsymbol{\Lambda}) \leq \widetilde{\mathsf{XY}}_{\mathrm{opt}}(\boldsymbol{\Lambda}).$$

*Proof.* This result is standard but we include the proof for completeness. We prove it for some generic sequence $(a_i)_{i=1}^n$. Take $\eta > 0$. Clearly,

$$\exp(\max_i \eta a_i) \leq \sum_{i=1}^n \exp(\eta a_i) \leq n \exp(\max_i \eta a_i)$$

so

$$\max_i \eta a_i \leq \log\left(\sum_{i=1}^n \exp(\eta a_i)\right) \leq \log n + \max_i \eta a_i.$$

The result follows by rearranging and dividing by $\eta$. $\qquad \square$

**Lemma D.2.** *If $\eta \geq \widetilde{\eta} \geq 0$, then $\widetilde{\mathsf{XY}}_{\mathrm{opt}}(\boldsymbol{\Lambda}; \eta) \leq \widetilde{\mathsf{XY}}_{\mathrm{opt}}(\boldsymbol{\Lambda}; \widetilde{\eta})$.*

*Proof.* We will prove this for some generic sequence $(a_i)_{i=1}^n$, $a_i \geq 0$. Note that,

$$\frac{\mathrm{d}}{\mathrm{d}\eta} \frac{1}{\eta} \log\left(\sum_i e^{\eta a_i}\right) = -\frac{1}{\eta^2} \log\left(\sum_i e^{\eta a_i}\right) + \frac{1}{\eta} \frac{1}{\sum_i e^{\eta a_i}} \cdot \sum_i a_i e^{\eta a_i}.$$

We are done if we can show this is non-positive. Note that,

$$\log\left(\sum_i e^{\eta a_i}\right) \geq \max_i \log(e^{\eta a_i}) = \max_i \eta a_i$$

so

$$-\frac{1}{\eta^2} \log\left(\sum_i e^{\eta a_i}\right) + \frac{1}{\eta} \frac{1}{\sum_i e^{\eta a_i}} \cdot \sum_i a_i e^{\eta a_i} \leq -\frac{1}{\eta} \max_i a_i + \frac{1}{\eta} \frac{1}{\sum_i e^{\eta a_i}} \cdot \sum_i a_i e^{\eta a_i}$$

$$\leq -\frac{1}{\eta} \max_i a_i + \frac{1}{\eta} \max_i a_i$$

$$= 0.$$

The result follows since $\widetilde{\mathsf{XY}}_{\mathrm{opt}}$ has this form. $\qquad \square$

**Lemma D.3.** *We have,*

$$\inf_{\boldsymbol{\Lambda} \succeq 0, \|\boldsymbol{\Lambda}\|_{\mathrm{op}} \leq 1} \mathsf{XY}_{\mathrm{opt}}(\boldsymbol{\Lambda}) \geq \frac{\gamma_\Phi}{1 + \|\boldsymbol{\Lambda}_0\|_{\mathrm{op}}}.$$

*Proof.* Note that $\|\mathbf{A}(\boldsymbol{\Lambda})\|_{\mathrm{op}} \le 1 + \|\boldsymbol{\Lambda}_0\|_{\mathrm{op}}$, so

$$\inf_{\boldsymbol{\Lambda} \succeq 0, \|\boldsymbol{\Lambda}\|_{\mathrm{op}} \le 1} \max_{\boldsymbol{\phi} \in \Phi} \|\boldsymbol{\phi}\|_{\mathbf{A}(\boldsymbol{\Lambda})^{-1}}^2 \ge \inf_{\boldsymbol{\Lambda} \succeq 0, \|\boldsymbol{\Lambda}\|_{\mathrm{op}} \le 1 + \|\boldsymbol{\Lambda}_0\|_{\mathrm{op}}} \|\boldsymbol{\phi}\|_{\mathbf{A}(\boldsymbol{\Lambda})^{-1}}^2 \ge \frac{\max_{\boldsymbol{\phi} \in \Phi} \|\boldsymbol{\phi}\|_2}{1 + \|\boldsymbol{\Lambda}_0\|_{\mathrm{op}}}.$$

$\square$

**Lemma D.4.** *Assume that we set $\eta \ge \frac{2}{\gamma_\Phi}(1 + \|\boldsymbol{\Lambda}_0\|_{\mathrm{op}}) \cdot \log |\Phi|$. Then*

$$N^\star(\epsilon; \widetilde{\mathsf{XY}}_{\mathrm{opt}}(\boldsymbol{\Lambda})) \le 2N^\star(\epsilon; \mathsf{XY}_{\mathrm{opt}}(\boldsymbol{\Lambda})).$$

*Proof.* Denote $f(\boldsymbol{\Lambda}) \leftarrow \mathrm{LogSumExp}\left(\{e^{\eta \|\boldsymbol{\phi}\|_{\mathbf{A}(\boldsymbol{\Lambda})^{-1}}^2}\}_{\boldsymbol{\phi} \in \Phi}; \eta\right)$. By Lemma D.1 and Lemma D.3, we have

$$|\max_{\boldsymbol{\phi} \in \Phi} \|\boldsymbol{\phi}\|_{\mathbf{A}(\boldsymbol{\Lambda})^{-1}}^2 - f(\boldsymbol{\Lambda})| \le \frac{\log |\Phi|}{\eta} \le \frac{\gamma_\Phi}{2(1 + \|\boldsymbol{\Lambda}_0\|_{\mathrm{op}})} \le \min_{\boldsymbol{\Lambda} \succeq 0, \|\boldsymbol{\Lambda}\|_{\mathrm{op}} \le 1} \frac{1}{2} f(\boldsymbol{\Lambda})$$
$$\implies f(\boldsymbol{\Lambda}) \le 2 \max_{\boldsymbol{\phi} \in \Phi} \|\boldsymbol{\phi}\|_{\mathbf{A}(\boldsymbol{\Lambda})^{-1}}^2.$$

Let $\boldsymbol{\Lambda}^\star$ denote the matrix that minimizes $\max_{\boldsymbol{\phi} \in \Phi} \|\boldsymbol{\phi}\|_{\mathbf{A}(\boldsymbol{\Lambda})^{-1}}^2$ over the constraint set: $\max_{\boldsymbol{\phi} \in \Phi} \|\boldsymbol{\phi}\|_{\mathbf{A}(\boldsymbol{\Lambda}^\star)^{-1}}^2 = \inf_{\boldsymbol{\Lambda} \in \Omega} \max_{\boldsymbol{\phi} \in \Phi} \|\boldsymbol{\phi}\|_{\mathbf{A}(\boldsymbol{\Lambda})^{-1}}^2$. Then it follows that, by definition of $N^\star(\epsilon; \max_{\boldsymbol{\phi} \in \Phi} \|\boldsymbol{\phi}\|_{\mathbf{A}(\boldsymbol{\Lambda})^{-1}}^2)$:

$$\max_{\boldsymbol{\phi} \in \Phi} \|\boldsymbol{\phi}\|_{\mathbf{A}(\boldsymbol{\Lambda}^\star)^{-1}}^2 \le \epsilon \cdot N^\star(\epsilon; \max_{\boldsymbol{\phi} \in \Phi} \|\boldsymbol{\phi}\|_{\mathbf{A}(\boldsymbol{\Lambda})^{-1}}^2).$$

However, this implies

$$\frac{1}{2} f(\boldsymbol{\Lambda}^\star) \le \epsilon \cdot N^\star(\epsilon; \max_{\boldsymbol{\phi} \in \Phi} \|\boldsymbol{\phi}\|_{\mathbf{A}(\boldsymbol{\Lambda})^{-1}}^2),$$

so $(\boldsymbol{\Lambda}^\star, 2N^\star(\epsilon; \max_{\boldsymbol{\phi} \in \Phi} \|\boldsymbol{\phi}\|_{\mathbf{A}(\boldsymbol{\Lambda})^{-1}}^2))$ is a feasible solution to the optimization (C.2) for $f$. As $N^\star(\epsilon; f)$ is the minimum solution, it follows that $N^\star(\epsilon; f) \le 2N^\star(\epsilon; \max_{\boldsymbol{\phi} \in \Phi} \|\boldsymbol{\phi}\|_{\mathbf{A}(\boldsymbol{\Lambda})^{-1}}^2)$. $\square$

### D.2 Bounding the Smoothness

**Lemma D.5.** $f(\boldsymbol{\Lambda}) = \widetilde{\mathsf{XY}}_{\mathrm{opt}}(\boldsymbol{\Lambda})$ *satisfies all conditions of Definition C.1 with*

$$L = \|\boldsymbol{\Lambda}_0^{-1}\|_{\mathrm{op}}^2, \quad \beta = 2\|\boldsymbol{\Lambda}_0^{-1}\|_{\mathrm{op}}^3(1 + \eta\|\boldsymbol{\Lambda}_0^{-1}\|_{\mathrm{op}}), \quad M = \|\boldsymbol{\Lambda}_0^{-1}\|_{\mathrm{op}}^2$$

$$\nabla_{\boldsymbol{\Lambda}} f(\boldsymbol{\Lambda}) = \left(\sum_{\boldsymbol{\phi} \in \Phi} e^{\eta \|\boldsymbol{\phi}\|_{\mathbf{A}(\boldsymbol{\Lambda})^{-1}}^2}\right)^{-1} \cdot \sum_{\boldsymbol{\phi} \in \Phi} e^{\eta \|\boldsymbol{\phi}\|_{\mathbf{A}(\boldsymbol{\Lambda})^{-1}}^2} \mathbf{A}(\boldsymbol{\Lambda})^{-1} \boldsymbol{\phi}\boldsymbol{\phi}^\top \mathbf{A}(\boldsymbol{\Lambda})^{-1} =: \Xi_{\boldsymbol{\Lambda}}.$$

*Proof.* Using Lemma D.6, the gradient of $f(\boldsymbol{\Lambda})$ with respect to $\boldsymbol{\Lambda}_{ij}$ is

$$\nabla_{\boldsymbol{\Lambda}_{ij}} f(\boldsymbol{\Lambda}) = -\left(\sum_{\boldsymbol{\phi} \in \Phi} e^{\eta \|\boldsymbol{\phi}\|_{\mathbf{A}(\boldsymbol{\Lambda})^{-1}}^2}\right)^{-1} \cdot \sum_{\boldsymbol{\phi} \in \Phi} e^{\eta \|\boldsymbol{\phi}\|_{\mathbf{A}(\boldsymbol{\Lambda})^{-1}}^2} \boldsymbol{\phi}^\top \mathbf{A}(\boldsymbol{\Lambda})^{-1} e_i e_j^\top \mathbf{A}(\boldsymbol{\Lambda})^{-1} \boldsymbol{\phi}$$

from which the expression for $\nabla_{\boldsymbol{\Lambda}} f(\boldsymbol{\Lambda})$ follows directly.

To bound the Lipschitz constant of $f$, by the Mean Value Theorem it suffices to bound

$$\sup_{\boldsymbol{\Lambda}, \widetilde{\boldsymbol{\Lambda}} \succeq 0, \|\boldsymbol{\Lambda}\|_{\mathrm{op}} \le 1, \|\widetilde{\boldsymbol{\Lambda}}\|_{\mathrm{op}} \le 1} |\mathrm{tr}(\nabla f(\boldsymbol{\Lambda})^\top \widetilde{\boldsymbol{\Lambda}})| \le \left(\sum_{\boldsymbol{\phi} \in \Phi} e^{\eta \|\boldsymbol{\phi}\|_{\mathbf{A}(\boldsymbol{\Lambda})^{-1}}^2}\right)^{-1} \cdot \sum_{\boldsymbol{\phi} \in \Phi} e^{\eta \|\boldsymbol{\phi}\|_{\mathbf{A}(\boldsymbol{\Lambda})^{-1}}^2} \|\mathbf{A}(\boldsymbol{\Lambda})^{-1}\|_{\mathrm{op}}^2 \|\widetilde{\boldsymbol{\Lambda}}\|_{\mathrm{op}}$$
$$\le \|\boldsymbol{\Lambda}_0^{-1}\|_{\mathrm{op}}^2$$

where the last inequality follows since $\mathbf{A}(\boldsymbol{\Lambda}) \succeq \boldsymbol{\Lambda}_0$ for all $\boldsymbol{\Lambda}$. This also suffices as a bound on $M$.

To bound the smoothness, again by the Mean Value Theorem it suffices to bound the operator norm of the Hessian. Standard calculus gives that, using $\nabla^2 f(\mathbf{\Lambda})[\widetilde{\mathbf{\Lambda}}, \bar{\mathbf{\Lambda}}]$ to denote the Hessian of $f$ in direction $(\widetilde{\mathbf{\Lambda}}, \bar{\mathbf{\Lambda}})$:

$$\nabla^2 f(\mathbf{\Lambda})[\widetilde{\mathbf{\Lambda}}, \bar{\mathbf{\Lambda}}] = -\frac{d}{dt}\left(\sum_{\phi \in \Phi} e^{\eta\|\phi\|^2_{\mathbf{A}(\mathbf{\Lambda}+t\bar{\mathbf{\Lambda}})^{-1}}}\right)^{-1} \cdot \sum_{\phi \in \Phi} e^{\eta\|\phi\|^2_{\mathbf{A}(\mathbf{\Lambda}+t\bar{\mathbf{\Lambda}})^{-1}}} \phi^\top \mathbf{A}(\mathbf{\Lambda}+t\bar{\mathbf{\Lambda}})^{-1}\widetilde{\mathbf{\Lambda}}\mathbf{A}(\mathbf{\Lambda}+t\bar{\mathbf{\Lambda}})^{-1}\phi$$

$$= -\eta\left(\sum_{\phi \in \Phi} e^{\eta\|\phi\|^2_{\mathbf{A}(\mathbf{\Lambda})^{-1}}}\right)^{-2}\left(\sum_{\phi \in \Phi} e^{\eta\|\phi\|^2_{\mathbf{A}(\mathbf{\Lambda})^{-1}}}\phi^\top \mathbf{A}(\mathbf{\Lambda})^{-1}\bar{\mathbf{\Lambda}}\mathbf{A}(\mathbf{\Lambda})^{-1}\phi\right)\left(\sum_{\phi \in \Phi} e^{\eta\|\phi\|^2_{\mathbf{A}(\mathbf{\Lambda})^{-1}}}\phi^\top \mathbf{A}(\mathbf{\Lambda})^{-1}\widetilde{\mathbf{\Lambda}}\mathbf{A}(\mathbf{\Lambda})^{-1}\phi\right)$$

$$+ \eta\left(\sum_{\phi \in \Phi} e^{\eta\|\phi\|^2_{\mathbf{A}(\mathbf{\Lambda})^{-1}}}\right)^{-1}\sum_{\phi \in \Phi} e^{\eta\|\phi\|^2_{\mathbf{A}(\mathbf{\Lambda})^{-1}}}\left(\phi^\top \mathbf{A}(\mathbf{\Lambda})^{-1}\bar{\mathbf{\Lambda}}\mathbf{A}(\mathbf{\Lambda})^{-1}\phi\right)\left(\phi^\top \mathbf{A}(\mathbf{\Lambda})^{-1}\widetilde{\mathbf{\Lambda}}\mathbf{A}(\mathbf{\Lambda})^{-1}\phi\right)$$

$$+ \left(\sum_{\phi \in \Phi} e^{\eta\|\phi\|^2_{\mathbf{A}(\mathbf{\Lambda})^{-1}}}\right)^{-1}\sum_{\phi \in \Phi} e^{\eta\|\phi\|^2_{\mathbf{A}(\mathbf{\Lambda})^{-1}}}\phi^\top \mathbf{A}(\mathbf{\Lambda})^{-1}\bar{\mathbf{\Lambda}}\mathbf{A}(\mathbf{\Lambda})^{-1}\widetilde{\mathbf{\Lambda}}\mathbf{A}(\mathbf{\Lambda})^{-1}\phi$$

$$+ \left(\sum_{\phi \in \Phi} e^{\eta\|\phi\|^2_{\mathbf{A}(\mathbf{\Lambda})^{-1}}}\right)^{-1}\sum_{\phi \in \Phi} e^{\eta\|\phi\|^2_{\mathbf{A}(\mathbf{\Lambda})^{-1}}}\phi^\top \mathbf{A}(\mathbf{\Lambda})^{-1}\widetilde{\mathbf{\Lambda}}\mathbf{A}(\mathbf{\Lambda})^{-1}\bar{\mathbf{\Lambda}}\mathbf{A}(\mathbf{\Lambda})^{-1}\phi.$$

We can bound this as

$$\sup_{\mathbf{\Lambda}, \widetilde{\mathbf{\Lambda}}, \bar{\mathbf{\Lambda}} \succeq 0, \|\mathbf{\Lambda}\|_{\mathrm{op}} \leq 1, \|\widetilde{\mathbf{\Lambda}}\|_{\mathrm{op}} \leq 1, \|\bar{\mathbf{\Lambda}}\|_{\mathrm{op}} \leq 1} |\nabla^2 f(\mathbf{\Lambda})[\widetilde{\mathbf{\Lambda}}, \bar{\mathbf{\Lambda}}]| \leq 2\eta\|\mathbf{\Lambda}_0^{-1}\|^4_{\mathrm{op}} + 2\|\mathbf{\Lambda}_0^{-1}\|^3_{\mathrm{op}}.$$

Convexity of $f(\mathbf{\Lambda})$ follows since it is the composition of a convex function with a strictly increasing convex function, so it is itself convex. $\qquad\square$

**Lemma D.6.** *For $\mathbf{\Lambda}$ invertible, $\frac{d}{dt}(\mathbf{\Lambda} + t\mathbf{e}_i\mathbf{e}_j^\top)^{-1} = -\mathbf{\Lambda}^{-1}\mathbf{e}_i\mathbf{e}_j^\top \mathbf{\Lambda}^{-1}$.*

*Proof.* We can compute the gradient as

$$\frac{d}{dt}(\mathbf{\Lambda} + t\mathbf{e}_i\mathbf{e}_j^\top)^{-1} = \lim_{t \to 0} \frac{(\mathbf{\Lambda} + t\mathbf{e}_i\mathbf{e}_j^\top)^{-1} - \mathbf{\Lambda}^{-1}}{t}.$$

By the Sherman-Morrison formula,

$$(\mathbf{\Lambda} + t\mathbf{e}_i\mathbf{e}_j^\top)^{-1} = \mathbf{\Lambda}^{-1} - \frac{t\mathbf{\Lambda}^{-1}\mathbf{e}_i\mathbf{e}_j^\top \mathbf{\Lambda}^{-1}}{1 + t\mathbf{e}_j^\top \mathbf{\Lambda}^{-1}\mathbf{e}_i}$$

so as $t \to 0$,

$$(\mathbf{\Lambda} + t\mathbf{e}_i\mathbf{e}_j^\top)^{-1} \to \mathbf{\Lambda}^{-1} - t\mathbf{\Lambda}^{-1}\mathbf{e}_i\mathbf{e}_j^\top \mathbf{\Lambda}^{-1}$$

Thus,

$$\lim_{t \to 0}\frac{(\mathbf{\Lambda} + t\mathbf{e}_i\mathbf{e}_j^\top)^{-1} - \mathbf{\Lambda}^{-1}}{t} = \lim_{t \to 0}\frac{\mathbf{\Lambda}^{-1} - t\mathbf{\Lambda}^{-1}\mathbf{e}_i\mathbf{e}_j^\top \mathbf{\Lambda}^{-1} - \mathbf{\Lambda}^{-1}}{t} = -\mathbf{\Lambda}^{-1}\mathbf{e}_i\mathbf{e}_j^\top \mathbf{\Lambda}^{-1}.$$

$\qquad\square$

### D.3 Obtaining Well-Conditioned Covariates

**Lemma D.7.** *Consider running policies $(\pi_\tau)_{\tau=1}^T$, for $\pi_\tau$ $\mathcal{F}_{\tau-1}$-measurable, and collecting covariance $\mathbf{\Sigma}_T = \sum_{\tau=1}^T \phi_\tau \phi_\tau^\top$. Then as long as*

$$\lambda_{\min}(\mathbf{\Sigma}_T) \geq 12544d \log \frac{2 + 32T}{\delta}.$$

*with probability at least $1 - \delta$, if we rerun each $(\pi_\tau)_{\tau=1}^T$, we will collect covariates $\widetilde{\mathbf{\Sigma}}_T$ such that*

$$\lambda_{\min}(\widetilde{\mathbf{\Sigma}}_T) \geq \frac{1}{2}\lambda_{\min}(\mathbf{\Sigma}_T).$$

---
**Algorithm 7** Collect Well-Conditioned Covariates (CONDITIONEDCOV)
---
1: **input**: Scale $N$, minimum eigenvalue $\underline{\lambda}$, confidence $\delta$
2: **for** $j = 1, 2, 3, \ldots$ **do**
3:     $T_j \leftarrow \mathrm{poly}(2^j, d, H, \log 1/\delta)$
4:     $\epsilon_j \leftarrow 2^{-j}, \gamma_j^2 \leftarrow \dfrac{2^{-j}}{\max\{12544d\log\frac{2N(2+32T_j)}{\delta}, \underline{\lambda}\}}, \delta_j \leftarrow \delta/(4j^2)$
5:     Run Algorithm 5 of Wagenmaker et al. (2022) with parameters $(\epsilon_j, \gamma_j^2, \delta_j)$, obtain covariates $\widetilde{\Sigma}$ and store policies run as $\widetilde{\Pi}$
6:     **if** $\lambda_{\min}(\widetilde{\Lambda}) \geq \max\{12544d\log\frac{2N(2+32T_j)}{\delta}, \underline{\lambda}\}$ **then**
7:         **break**
8: Rerun every policy $\pi \in \widetilde{\Pi} \lceil N/|\widetilde{\Pi}| \rceil$ times, collect covariates $\bar{\Sigma}$
9: **return** $\widetilde{\Sigma} + \bar{\Sigma}$
---

*Proof.* Let $\mathcal{N}$ be an $\frac{1}{8T}$-net of $\mathcal{S}^{d-1}$. Let $\Sigma \succeq 0$ be any matrix with $\|\Sigma\|_{\mathrm{op}} \leq T$ and let $v$ be the minimum eigenvalue of $\Sigma$. Let $\widetilde{v} \in \mathcal{N}$ be the element of $\mathcal{N}$ closest to $v$ in the $\ell_2$ norm. Then:

$$\lambda_{\min}(\Sigma) = v^\top \Sigma v = \widetilde{v}^\top \Sigma \widetilde{v} + (v^\top \Sigma v - \widetilde{v}^\top \Sigma \widetilde{v})$$
$$\geq \widetilde{v}^\top \Sigma \widetilde{v} - |v^\top \Sigma v - v^\top \Sigma \widetilde{v}| - |v^\top \Sigma \widetilde{v} - \widetilde{v}^\top \Sigma \widetilde{v}|$$
$$\geq \widetilde{v}^\top \Sigma \widetilde{v} - 2\|\Sigma\|_{\mathrm{op}}\|\widetilde{v} - v\|_2.$$

By the construction of $\mathcal{N}$ and since $\|\Sigma\|_{\mathrm{op}} \leq T$, we can bound $2\|\Sigma\|_{\mathrm{op}}\|\widetilde{v} - v\|_2 \leq 1/4$, so

$$\widetilde{v}^\top \Sigma \widetilde{v} - 2\|\Sigma\|_{\mathrm{op}}\|\widetilde{v} - v\|_2 \geq \widetilde{v}^\top \Sigma \widetilde{v} - 1/4$$

which implies

$$\lambda_{\min}(\Sigma) + 1/4 \geq \widetilde{v}^\top \Sigma \widetilde{v} \geq \min_{\widetilde{v} \in \mathcal{N}} \widetilde{v}^\top \Sigma \widetilde{v}. \tag{D.2}$$

By Lemma A.1, we can bound $|\mathcal{N}| \leq (1 + 16T)^d$.

Note that $\mathrm{Var}[v^\top \phi_\tau | \mathcal{F}_{\tau-1}] \leq \mathbb{E}_{\pi_\tau}[(v^\top \phi_\tau)^2]$ so $\sum_{\tau=1}^{T} \mathrm{Var}[v^\top \phi_\tau | \mathcal{F}_{\tau-1}] \leq v^\top \mathbb{E}[\Sigma_T | \pi_1, \ldots, \pi_T]v$ for $\mathbb{E}[\Sigma_T | \pi_1, \ldots, \pi_T] = \sum_{\tau=1}^{T} \mathbb{E}_{\pi_\tau}[\phi_\tau \phi_\tau^\top]$. By Freedman's Inequality (Lemma A.5), for all $v \in \mathcal{N}$ simultaneously, we will have, with probability at least $1 - \delta$,

$$\left| v^\top \Sigma_T v - v^\top \mathbb{E}[\Sigma_T | \pi_1, \ldots, \pi_T]v \right| \leq 2\sqrt{v^\top \mathbb{E}[\Sigma_T | \pi_1, \ldots, \pi_T]v \log \frac{2|\mathcal{N}|}{\delta}} + \log \frac{2|\mathcal{N}|}{\delta} \tag{D.3}$$

$$\left| v^\top \widetilde{\Sigma}_T v - v^\top \mathbb{E}[\Sigma_T | \pi_1, \ldots, \pi_T]v \right| \leq 2\sqrt{v^\top \mathbb{E}[\Sigma_T | \pi_1, \ldots, \pi_T]v \log \frac{2|\mathcal{N}|}{\delta}} + \log \frac{2|\mathcal{N}|}{\delta}. \tag{D.4}$$

Rearranging (D.3), some algebra shows that

$$v^\top \mathbb{E}[\Sigma_T | \pi_1, \ldots, \pi_T]v \leq v^\top \Sigma_T v + 3\log \frac{2|\mathcal{N}|}{\delta} + 2\sqrt{v^\top \Sigma_T v \log \frac{2|\mathcal{N}|}{\delta} + 2\log^2 \frac{2|\mathcal{N}|}{\delta}}$$
$$\leq v^\top \Sigma_T v + 6\log \frac{2|\mathcal{N}|}{\delta} + 2\sqrt{v^\top \Sigma_T v \log \frac{2|\mathcal{N}|}{\delta}}$$
$$\leq 3v^\top \Sigma_T v + 8\log \frac{2|\mathcal{N}|}{\delta}$$

where the last inequality uses $\sqrt{ab} \leq \max\{a, b\}$. Thus, if (D.3) and (D.4) hold, we have

$$v^\top \widetilde{\Sigma}_T v \geq v^\top \Sigma_T v - 4\sqrt{v^\top \mathbb{E}[\Sigma_T | \pi_1, \ldots, \pi_T]v \log \frac{2|\mathcal{N}|}{\delta}} - 2\log \frac{2|\mathcal{N}|}{\delta}$$
$$\geq v^\top \Sigma_T v - 4\sqrt{3v^\top \Sigma_T v \log \frac{2|\mathcal{N}|}{\delta}} - 14\log \frac{2|\mathcal{N}|}{\delta}$$

Therefore, as long as

$$v^\top \Sigma_T v \geq 12544\log \frac{2|\mathcal{N}|}{\delta},$$

we can lower bound

$$\boldsymbol{v}^\top \boldsymbol{\Sigma}_T \boldsymbol{v} - 4\sqrt{3\boldsymbol{v}^\top \boldsymbol{\Sigma}_T \boldsymbol{v} \log \frac{2|\mathcal{N}|}{\delta}} - 14 \log \frac{2|\mathcal{N}|}{\delta} \geq \frac{3}{4}\boldsymbol{v}^\top \boldsymbol{\Sigma}_T \boldsymbol{v} \geq \frac{3}{4}\lambda_{\min}(\boldsymbol{\Sigma}_T)$$

so, for all $\boldsymbol{v} \in \mathcal{N}$,

$$\boldsymbol{v}^\top \widetilde{\boldsymbol{\Sigma}}_T \boldsymbol{v} \geq \frac{3}{4}\lambda_{\min}(\boldsymbol{\Sigma}_T).$$

By assumption, $\lambda_{\min}(\boldsymbol{\Sigma}_T) \geq 12544d \log \frac{2+32T}{\delta}$, which implies, since $|\mathcal{N}| \leq (1 + 16T)^d$, that for all $\boldsymbol{v} \in \mathcal{S}^{d-1}$, $\boldsymbol{v}^\top \boldsymbol{\Sigma}_T \boldsymbol{v} \geq 12544 \log \frac{2|\mathcal{N}|}{\delta}$, so the above condition will be met.

Since $\|\widetilde{\boldsymbol{\Sigma}}_T\|_{\mathrm{op}} \leq T$, we can apply (D.2) to then get that

$$\lambda_{\min}(\widetilde{\boldsymbol{\Sigma}}_T) \geq \frac{3}{4}\lambda_{\min}(\boldsymbol{\Sigma}_T) - 1/4 \geq \frac{1}{2}\lambda_{\min}(\boldsymbol{\Sigma}_T) + \frac{1}{4}(\lambda_{\min}(\boldsymbol{\Sigma}_T) - 1).$$

Since we have already establishes that $\lambda_{\min}(\boldsymbol{\Sigma}_T) \geq 12544d \log \frac{2+32T}{\delta}$, we have $\lambda_{\min}(\boldsymbol{\Sigma}_T) - 1 \geq 0$, so we can lower bound

$$\lambda_{\min}(\widetilde{\boldsymbol{\Sigma}}_T) \geq \frac{1}{2}\lambda_{\min}(\boldsymbol{\Sigma}_T).$$

$\square$

**Lemma D.8.** *With probability at least $1 - \delta$, Algorithm 7 will terminate after at most*

$$N + \mathrm{poly} \log \left( \frac{1}{\sup_\pi \lambda_{\min}(\boldsymbol{\Sigma}_\pi)}, d, H, \underline{\lambda}, \log \frac{N}{\delta} \right) \cdot \left( \frac{d \max\{d \log \frac{N}{\delta}, \underline{\lambda}\}}{\sup_\pi \lambda_{\min}(\boldsymbol{\Sigma}_\pi)^2} + \frac{d^4 H^3 \log^{7/2} \frac{1}{\delta}}{\sup_\pi \lambda_{\min}(\boldsymbol{\Sigma}_\pi)} \right)$$

*episodes, and will return covariates $\boldsymbol{\Sigma}$ such that*

$$\lambda_{\min}(\boldsymbol{\Sigma}) \geq N \cdot \min \left\{ \frac{\sup_\pi \lambda_{\min}(\boldsymbol{\Sigma}_\pi)^2}{d}, \frac{\sup_\pi \lambda_{\min}(\boldsymbol{\Sigma}_\pi)}{d^3 H^3 \log^{7/2} 1/\delta} \right\} \cdot \mathrm{poly} \log \left( \frac{1}{\sup_\pi \lambda_{\min}(\boldsymbol{\Sigma}_\pi)}, d, H, \underline{\lambda}, \log \frac{N}{\delta} \right)^{-1}$$
$$+ \max\{d \log 1/\delta, \underline{\lambda}\}$$

*and*

$$\|\boldsymbol{\Sigma}\|_{\mathrm{op}} \leq N + \mathrm{poly} \log \left( \frac{1}{\sup_\pi \lambda_{\min}(\boldsymbol{\Sigma}_\pi)}, d, H, \underline{\lambda}, \log \frac{N}{\delta} \right) \cdot \left( \frac{d \max\{d \log \frac{N}{\delta}, \underline{\lambda}\}}{\sup_\pi \lambda_{\min}(\boldsymbol{\Sigma}_\pi)^2} + \frac{d^4 H^3 \log^{7/2} \frac{1}{\delta}}{\sup_\pi \lambda_{\min}(\boldsymbol{\Sigma}_\pi)} \right).$$

*Proof.* By Theorem 4 of Wagenmaker et al. (2022), as long as Algorithm 5 of Wagenmaker et al. (2022) is run with parameters $\epsilon$ and $\gamma^2$, it will terminate after at most

$$c_1 \cdot \frac{1}{\epsilon} \max \left\{ \frac{dm}{\gamma^2} \log \frac{dm}{\epsilon\gamma^2}, d^4 H^3 m^{7/2} \log^{3/2}(d/\gamma^2) \log^{7/2} \frac{c_2 m dH \log(d/\gamma^2)}{\delta} \right\}$$

episodes for $m = \lceil \log(2/\epsilon) \rceil$ (to get the slightly more precise bound on the number of episodes collected than that given in Theorem 4 of Wagenmaker et al. (2022), we use the precise definition of $K_i$ given at the start of Appendix B). Furthermore, if $\epsilon \leq \sup_\pi \lambda_{\min}(\boldsymbol{\Sigma}_\pi)$, with probability at least $1 - \delta$ it will collect covariates $\widetilde{\boldsymbol{\Sigma}}$ satisfying $\lambda_{\min}(\widetilde{\boldsymbol{\Sigma}}) \geq \epsilon/\gamma^2$.

It follows that, by our choice of $\epsilon_j = 2^{-j}$, $\gamma_j^2 = \frac{2^{-j}}{\max\{12544d \log \frac{2N(2+32T_j)}{\delta}, \underline{\lambda}\}}$, and $\delta_j = \delta/(4j^2)$, for every $j$ we will collect at most

$$c_1 \cdot 2^j \max \left\{ 2^j dj^2 \max\{d \log \frac{2N(2+32T_j)}{\delta}, \underline{\lambda}\} \log(dja_j), d^4 H^3 j^5 \log^{3/2}(da_j) \log^{7/2} \frac{c_2 j^4 dH \log(da_j)}{\delta} \right\}$$

episodes, where we denote $a_j := \max\{12544d \log \frac{2N(2+32T_j)}{\delta}, \underline{\lambda}\}$. Note that $T_j$ is an upper bound on this complexity. Furthermore, once $j$ is large enough that $2^{-j} \leq \sup_\pi \lambda_{\min}(\boldsymbol{\Sigma}_\pi)$, Theorem 4 of Wagenmaker et al. (2022) implies that the condition $\lambda_{\min}(\widetilde{\boldsymbol{\Sigma}}) \geq \epsilon_j/\gamma_j^2$ will be met. By our choice of $\gamma_j^2$ and $\epsilon_j$, it follows that the if condition on Line 6 will be met once $2^{-j} \leq \sup_\pi \lambda_{\min}(\boldsymbol{\Sigma}_\pi)$.

Since $2^{-j}$ decreases by a factor of 2 each time, it follows that the if statement on Line 6 will have terminated once $2^{-j} \geq \sup_\pi \lambda_{\min}(\boldsymbol{\Sigma}_\pi)/2$. This implies that the total number of episodes collected before the if statement on Line 6 is met is bounded as

$$\text{poly}\log\left(\frac{1}{\sup_\pi \lambda_{\min}(\boldsymbol{\Sigma}_\pi)}, d, H, \underline{\lambda}, \log\frac{N}{\delta}\right) \cdot \left(\frac{d\max\{d\log\frac{N}{\delta}, \underline{\lambda}\}}{\sup_\pi \lambda_{\min}(\boldsymbol{\Sigma}_\pi)^2} + \frac{d^4 H^3 \log^{7/2}\frac{1}{\delta}}{\sup_\pi \lambda_{\min}(\boldsymbol{\Sigma}_\pi)}\right) \quad \text{(D.5)}$$

By Lemma D.7, since $\lambda_{\min}(\widetilde{\boldsymbol{\Sigma}}) \geq \max\{12544d\log\frac{2N(2+32T_j)}{\delta}, \underline{\lambda}\}$ and $T_j$ is an upper bound on the number of episodes run at epoch $j$, every time we run all policies $\pi \in \widetilde{\Pi}$, with probability at least $1 - \delta/(2N)$, we will collect covariates $\boldsymbol{\Sigma}$ such that

$$\lambda_{\min}(\boldsymbol{\Sigma}) \geq \lambda_{\min}(\widetilde{\boldsymbol{\Sigma}})/2 \geq \frac{1}{2}\max\{12544d\log\frac{2N(2+32T_j)}{\delta}, \underline{\lambda}\}.$$

Thus, if we rerun every policy $\lceil N/|\widetilde{\Pi}|\rceil$ times to create covariates $\bar{\boldsymbol{\Sigma}}$, with probability at least $1 - \delta/2$, we have

$$\lambda_{\min}(\bar{\boldsymbol{\Sigma}}) \geq \frac{N}{2|\widetilde{\Pi}|}\max\{12544d\log\frac{2N(2+32T_j)}{\delta}, \underline{\lambda}\}.$$

Note that this procedure will complete after at most $N + |\widetilde{\Pi}|$ episodes. Furthermore, $|\widetilde{\Pi}| \leq$ (D.5), so we can lower bound

$$\lambda_{\min}(\bar{\boldsymbol{\Sigma}}) \geq N \cdot \min\left\{\frac{\sup_\pi \lambda_{\min}(\boldsymbol{\Sigma}_\pi)^2}{d}, \frac{\sup_\pi \lambda_{\min}(\boldsymbol{\Sigma}_\pi)}{d^3 H^3 \log^{7/2} 1/\delta}\right\} \cdot \text{poly}\log\left(\frac{1}{\sup_\pi \lambda_{\min}(\boldsymbol{\Sigma}_\pi)}, d, H, \underline{\lambda}, \log\frac{N}{\delta}\right)^{-1}.$$

The final lower bound on the returned covariates follows since we return $\bar{\boldsymbol{\Sigma}} + \widetilde{\boldsymbol{\Sigma}}$, and we know that $\lambda_{\min}(\widetilde{\boldsymbol{\Sigma}}) \geq \max\{12544d\log\frac{2N(2+32T_j)}{\delta}, \underline{\lambda}\}$. The upper bound on $\|\bar{\boldsymbol{\Sigma}} + \widetilde{\boldsymbol{\Sigma}}\|_{\text{op}}$ follows since every feature vector encountered has norm of at most 1.

The failure probability of each call to Algorithm 5 of Wagenmaker et al. (2022) is $\delta/(4j^2)$, so the total failure probability of Algorithm 7 is

$$\sum_{j=1}^\infty \frac{\delta}{4j^2} = \frac{\pi^2}{24}\delta \leq \delta/2.$$

$\square$

## D.4 Online $\mathsf{XY}$-Optimal Design

**Theorem 8** (Full version of Theorem 5). *Consider running* OPTCOV *with some $\epsilon > 0$ and functions*

$$f_i(\boldsymbol{\Lambda}) \leftarrow \widetilde{\mathsf{XY}}_{\text{opt}}(\boldsymbol{\Lambda})$$

*for $\boldsymbol{\Lambda}_0 \leftarrow (T_i K_i)^{-1}\boldsymbol{\Sigma}_i =: \boldsymbol{\Lambda}_i$ and*

$$\eta_i = \frac{2}{\gamma_\Phi} \cdot (1 + \|\boldsymbol{\Lambda}_i\|_{\text{op}}) \cdot \log|\Phi|$$

$$L_i = \|\boldsymbol{\Lambda}_i^{-1}\|_{\text{op}}^2, \quad \beta_i = 2\|\boldsymbol{\Lambda}_i^{-1}\|_{\text{op}}^3(1 + \eta_i\|\boldsymbol{\Lambda}_i^{-1}\|_{\text{op}}), \quad M_i = \|\boldsymbol{\Lambda}_i^{-1}\|_{\text{op}}^2$$

*where $\boldsymbol{\Sigma}_i$ is the matrix returned by running* CONDITIONEDCOV *with $N \leftarrow T_i K_i$, $\delta \leftarrow \delta/(2i^2)$, and some $\underline{\lambda} \geq 0$. Then with probability $1 - 2\delta$, this procedure will collect at most*

$$20 \cdot \frac{\inf_{\boldsymbol{\Lambda}\in\boldsymbol{\Omega}}\max_{\phi\in\Phi}\|\phi\|_{\mathbf{A}(\boldsymbol{\Lambda})^{-1}}^2}{\epsilon_{\exp}} + \text{poly}\left(d, H, \log 1/\delta, \frac{1}{\lambda_{\min}^\star}, \frac{1}{\gamma_\Phi}, \underline{\lambda}, \log|\Phi|, \log\frac{1}{\epsilon_{\exp}}\right)$$

*episodes, where*

$$\mathbf{A}(\boldsymbol{\Lambda}) = \boldsymbol{\Lambda} + \min\left\{\frac{(\lambda_{\min}^\star)^2}{d}, \frac{\lambda_{\min}^\star}{d^3 H^3 \log^{7/2} 1/\delta}\right\} \cdot \text{poly}\log\left(\frac{1}{\lambda_{\min}^\star}, d, H, \underline{\lambda}, \log\frac{1}{\delta}\right)^{-1} \cdot I,$$

*and will produce covariates $\widehat{\boldsymbol{\Sigma}} + \boldsymbol{\Sigma}_i$ such that*

$$\max_{\phi\in\Phi}\|\phi\|_{(\widehat{\boldsymbol{\Sigma}}+\boldsymbol{\Sigma}_i)^{-1}}^2 \leq \epsilon_{\exp}$$

*and*

$$\lambda_{\min}(\widehat{\boldsymbol{\Sigma}} + \boldsymbol{\Sigma}_i) \geq \max\{d\log 1/\delta, \underline{\lambda}\}.$$

*Proof.* Note that the total failure probability of our calls to CONDITIONEDCOV is at most

$$\sum_{i=1}^{\infty} \frac{\delta}{2i^2} = \frac{\pi^2}{12}\delta \le \delta.$$

For the remainder of the proof, we will then assume that we are on the success event of CONDITIONEDCOV, as defined in Lemma D.8.

By Lemma D.5, $f_i(\boldsymbol{\Lambda})$ satisfies Definition C.1 with constants

$$L_i = \|\boldsymbol{\Lambda}_i^{-1}\|_{\mathrm{op}}^2, \quad \beta_i = 2\|\boldsymbol{\Lambda}_i^{-1}\|_{\mathrm{op}}^3(1 + \eta_i\|\boldsymbol{\Lambda}_i^{-1}\|_{\mathrm{op}}), \quad M_i = \|\boldsymbol{\Lambda}_i^{-1}\|_{\mathrm{op}}^2$$

for $\boldsymbol{\Lambda}_i \leftarrow (T_iK_i)^{-1}\boldsymbol{\Sigma}_i$.

By Lemma D.8, on the success event of Lemma D.8 we have that

$$\lambda_{\min}(\boldsymbol{\Lambda}_i) \ge \min\left\{\frac{(\lambda_{\min}^\star)^2}{d}, \frac{\lambda_{\min}^\star}{d^3H^3\log^{7/2}1/\delta}\right\} \cdot \mathrm{poly}\log\left(\frac{1}{\lambda_{\min}^\star}, d, H, \underline{\lambda}, i, \log\frac{1}{\delta}\right)^{-1}$$

(note that the $\mathrm{poly}\log(i)^{-1}$ dependence arises because we take $N \leftarrow T_iK_i = 2^{4i}$). Thus, we can bound, for all $i$ (using the upper bound on $\|\boldsymbol{\Sigma}_i\|_{\mathrm{op}}$ given in Lemma D.8 to upper bound $\eta_i$),

$$L_i = M_i \le \max\left\{\frac{d^2}{(\lambda_{\min}^\star)^4}, \frac{d^6H^6\log^7 1/\delta}{(\lambda_{\min}^\star)^2}\right\} \cdot \mathrm{poly}\log\left(\frac{1}{\lambda_{\min}^\star}, d, H, \underline{\lambda}, i, \log\frac{1}{\delta}\right),$$

$$\beta_i \le \mathrm{poly}\left(d, H, \log 1/\delta, \frac{1}{\lambda_{\min}^\star}, \frac{1}{\gamma_\Phi}, \underline{\lambda}, i, \log|\Phi|\right).$$

Assume that the termination condition of OPTCOV for $\widehat{i}$ satisfying

$$\widehat{i} \le \log\left(\mathrm{poly}\left(\frac{1}{\epsilon_{\exp}}, d, H, \log 1/\delta, \frac{1}{\lambda_{\min}^\star}, \frac{1}{\gamma_\Phi}, \underline{\lambda}, \log|\Phi|\right)\right). \tag{D.6}$$

We assume this holds and justify it at the conclusion of the proof. For notational convenience, define

$$\iota := \mathrm{poly}\left(\log\frac{1}{\epsilon_{\exp}}, d, H, \log 1/\delta, \frac{1}{\lambda_{\min}^\star}, \frac{1}{\gamma_\Phi}, \underline{\lambda}, \log|\Phi|\right).$$

Given this upper bound on $\widehat{i}$, set

$$L = M := \max\left\{\frac{d^2}{(\lambda_{\min}^\star)^4}, \frac{d^6H^6\log^7 1/\delta}{(\lambda_{\min}^\star)^2}\right\} \cdot \mathrm{poly}\log\iota, \qquad \beta := \iota.$$

With this choice of $L, M, \beta$, we have $L_i \le L, M_i \le M, \beta_i \le \beta$ for all $i \le \widehat{i}$.

Now take $f(\boldsymbol{\Lambda}) \leftarrow \widetilde{\mathsf{XY}}_{\mathrm{opt}}(\boldsymbol{\Lambda}; \eta, \boldsymbol{\Lambda}_0)$ with

$$\boldsymbol{\Lambda}_0 \leftarrow \min\left\{\frac{(\lambda_{\min}^\star)^2}{d}, \frac{\lambda_{\min}^\star}{d^3H^3\log^{7/2}1/\delta}\right\} \cdot \frac{1}{\mathrm{poly}\log\iota} \cdot I \tag{D.7}$$

and

$$\eta = \frac{2\log|\Phi|}{\gamma_\Phi} \cdot \left(1 + \min\left\{\frac{(\lambda_{\min}^\star)^2}{d}, \frac{\lambda_{\min}^\star}{d^3H^3\log^{7/2}1/\delta}\right\} \cdot \frac{1}{\mathrm{poly}\log\iota}\right).$$

Note that in this case, we have $\|\boldsymbol{\Lambda}_0\|_{\mathrm{op}} \le \lambda_{\min}(\boldsymbol{\Lambda}_i)$ for all $i$, so $\boldsymbol{\Lambda}_0 \preceq \boldsymbol{\Lambda}_i$ and $\eta \le \eta_i$. By the construction of $\widetilde{\mathsf{XY}}_{\mathrm{opt}}$ and Lemma D.2, it follows that $f(\boldsymbol{\Lambda}) \ge f_i(\boldsymbol{\Lambda})$ for all $\boldsymbol{\Lambda} \succeq 0$, so this is a valid choice of $f$, as required by Theorem 8. Furthermore, we can set $R = 2$, since $\|\boldsymbol{\Lambda}_\pi\|_{\mathrm{F}} \le 1$ for all $\pi$.

To apply Theorem 8, it remains only to find a suitable value of $f_{\min}$. By Lemma D.1 and Lemma D.3, we can lower bound $f_i$ by $\frac{\gamma_\Phi}{1+\|\boldsymbol{\Lambda}_i\|_{\mathrm{op}}}$. By Lemma D.8, we can lower bound

$$\frac{\gamma_\Phi}{1+\|\boldsymbol{\Lambda}_i\|_{\mathrm{op}}} \ge \frac{\gamma_\Phi}{2 + \mathrm{poly}\log\iota \cdot \left(\frac{d\max\{d\log\frac{1}{\delta}, \underline{\lambda}\}}{(\lambda_{\min}^\star)^2} + \frac{d^4H^3\log^{7/2}\frac{1}{\delta}}{\lambda_{\min}^\star}\right)}.$$

We then take this as our choice of $f_{\min}$.

We can now apply Theorem 8, using the complexity for OPTCOV instantiated with FORCE given in Corollary 6, and get that with probability at least $1 - \delta$, OPTCOV will terminate in

$$N \leq 5N^\star(\epsilon_{\exp}/2; f) + \iota$$

episodes, and will return (time-normalized) covariates $\widehat{\boldsymbol{\Lambda}}$ such that

$$f_{\widehat{i}}(\widehat{\boldsymbol{\Lambda}}) \leq N\epsilon_{\exp}.$$

By Lemma D.4, our choice of $\eta$ and $\boldsymbol{\Lambda}_0$, we can upper bound

$$N^\star(\epsilon_{\exp}/2; f) \leq 2N^\star(\epsilon_{\exp}/2; \mathsf{XY}_{\mathrm{opt}}) = \frac{4 \inf_{\boldsymbol{\Lambda} \in \boldsymbol{\Omega}} \max_{\boldsymbol{\phi} \in \Phi} \|\boldsymbol{\phi}\|^2_{\mathbf{A}(\boldsymbol{\Lambda})^{-1}}}{\epsilon_{\exp}}$$

where here $\mathbf{A}(\boldsymbol{\Lambda}) = \boldsymbol{\Lambda} + \boldsymbol{\Lambda}_0$ for $\boldsymbol{\Lambda}_0$ as in (D.7). Furthermore, by Lemma D.1 we have

$$\max_{\boldsymbol{\phi} \in \Phi} \|\boldsymbol{\phi}\|^2_{(\widehat{\boldsymbol{\Sigma}} + \boldsymbol{\Lambda}_0)^{-1}} \leq f_{\widehat{i}}(\widehat{\boldsymbol{\Lambda}}).$$

The final upper bound on the number of episodes collected and the lower bound on the minimum eigenvalue of the covariates follows from Lemma D.8.

It remains to justify our bound on $\widehat{i}$, (D.6). Note that by definition of OPTCOV, if we run for a total of $\bar{N}$ episodes, we can bound $\widehat{i} \leq \frac{1}{4} \log_2(\bar{N})$. However, we see that the bound on $\widehat{i}$ given in (D.6) upper bounds $\frac{1}{4} \log_2(\bar{N})$ for $\bar{N}$ the upper bound on the number of samples collected by OPTCOV stated above. Thus, our bound on $\widehat{i}$ is valid. $\qquad\square$

# E   Suboptimality of Optimistic Algorithms

## E.1   Linear Bandit Construction

In the linear bandit setting, at each time step $t$, the learner chooses some $\boldsymbol{z}_t \in \mathcal{Z}$, and observes $y_t$. We will consider the case when the noise is Bernoulli so that $y_t \sim \mathrm{Bernoulli}(\langle \boldsymbol{\theta}_\star, \boldsymbol{z}_t \rangle + 1/2)$, and will set

$$\boldsymbol{\theta}_\star = \boldsymbol{e}_1, \quad \mathcal{Z} = \{\xi \boldsymbol{e}_1, \boldsymbol{e}_2, \ldots, \boldsymbol{e}_d, \boldsymbol{x}_2, \ldots, \boldsymbol{x}_d\}, \quad \boldsymbol{x}_i = (\xi - \Delta)\boldsymbol{e}_1 + \gamma \boldsymbol{e}_i$$

for some $\xi, \Delta, \alpha$ to be chosen. In this setting, the optimal arm is $\boldsymbol{z}^\star = \xi \boldsymbol{e}_1$, and $\Delta(\boldsymbol{e}_i) = \xi, i \geq 2$, $\Delta(\boldsymbol{x}_i) = \Delta$.

We will assume:

$$\frac{1}{52d} \geq \xi \geq \max\{\gamma/\sqrt{d}, \sqrt{\Delta}\}, \quad \max\left\{\zeta := \frac{2\mathcal{C}_1}{(d/\Delta^2)^{1-\alpha}} + \frac{2\mathcal{C}_2\Delta^2}{d}, \Delta\right\} \leq \gamma^2. \tag{E.1}$$

We provide explicit values for $\xi, \Delta$, and $\gamma$ that satisfy this in Lemma E.3.

**Definition E.1** ($\delta$-correct). We say a stopping rule $\tau$ is $\delta$-correct if $\mathbb{P}[\widehat{\boldsymbol{z}}_\tau = \boldsymbol{z}^\star] \geq 1 - \delta$, where $\widehat{\boldsymbol{z}}_\tau$ is the arm recommended at time $\tau$.

**Lemma E.1.** *Consider running some low-regret algorithm satisfying Definition 4.1 on the linear bandit instance described above and let $\tau$ be some stopping time. Then if $\tau$ is $\delta$-correct, we must have that*

$$\mathbb{E}[\tau] \geq \frac{d-1}{48\Delta^2} \cdot \log \frac{1}{2.4\delta}.$$

*Proof.* This proof follows closely the proof of Theorem 1 of Fiez et al. (2019) and relies on the Transportation Lemma of Kaufmann et al. (2016).

**Bounding the number of pulls to $\{\boldsymbol{e}_2, \ldots, \boldsymbol{e}_d\}$.** By assumption, we collect data with a low-regret algorithm satisfying Definition 4.1. Every time we pull $\boldsymbol{e}_i, i \geq 2$, we incur a loss of $1/2$. Thus, we can lower bound

$$\mathbb{E}[V_0^\star - V_0^{\pi_k}] \geq \frac{1}{2} \sum_{i=2}^d \mathbb{E}[\mathbb{P}_{\pi_k}[\boldsymbol{z}_k = \boldsymbol{x}_i]]$$

so, letting $T(\boldsymbol{x}_i)$ denote the total number of pulls to $\boldsymbol{x}_i$, we have

$$\mathcal{C}_1 K^\alpha + \mathcal{C}_2 \geq \sum_{k=1}^K \mathbb{E}[V_0^\star - V_0^{\pi_k}] \geq \frac{1}{2} \sum_{k=1}^K \sum_{i=2}^d \mathbb{E}[\mathbb{P}_{\pi_k}[\boldsymbol{z}_k = \boldsymbol{x}_i]] = \frac{1}{2} \sum_{i=2}^d \mathbb{E}[T(\boldsymbol{x}_i)]. \tag{E.2}$$

**Applying the Transportation Lemma.** Let $\Theta_{\mathrm{alt}}$ denote the set of $\boldsymbol{\theta}$ vectors such that $\xi \boldsymbol{e}_1$ is not the optimal arm, that is, $\max_{\boldsymbol{z}\in\mathcal{X}}\langle\boldsymbol{\theta},\boldsymbol{z}\rangle > \langle\boldsymbol{\theta},\xi\boldsymbol{e}_1\rangle$. Let $\nu_{\boldsymbol{\theta},\boldsymbol{z}} = \mathrm{Bernoulli}(\langle\boldsymbol{\theta},\boldsymbol{z}\rangle + 1/2)$. Then by the Transportation Lemma of Kaufmann et al. (2016), for any $\boldsymbol{\theta}\in\Theta_{\mathrm{alt}}$, assuming our stopping rule is $\delta$-correct, we have

$$\sum_{\boldsymbol{z}\in\mathcal{Z}} \mathbb{E}[T(\boldsymbol{z})]\mathrm{KL}(\nu_{\boldsymbol{\theta}_\star,\boldsymbol{z}}||\nu_{\boldsymbol{\theta},\boldsymbol{z}}) \geq \log\frac{1}{2.4\delta}.$$

Combining this with our constraint (E.2), it follows that $\sum_{\boldsymbol{z}\in\mathcal{Z}}\mathbb{E}[T(\boldsymbol{z})] \geq \sum_{\boldsymbol{z}\in\mathcal{Z}} t_{\boldsymbol{z}}$ for any $(t_{\boldsymbol{z}})_{\boldsymbol{z}\in\mathcal{Z}}$ that is a feasible solution to

$$\min \sum_{\boldsymbol{z}\in\mathcal{Z}} t_{\boldsymbol{z}} \quad \text{s.t.} \quad \min_{\boldsymbol{\theta}\in\Theta_{\mathrm{alt}}} \sum_{\boldsymbol{z}\in\mathcal{Z}} t_{\boldsymbol{z}}\mathrm{KL}(\nu_{\boldsymbol{\theta}_\star,\boldsymbol{z}}||\nu_{\boldsymbol{\theta},\boldsymbol{z}}) \geq \log\frac{1}{2.4\delta}, \mathcal{C}_1(\sum_{\boldsymbol{z}\in\mathcal{Z}} t_{\boldsymbol{z}})^\alpha + \mathcal{C}_2 \geq \frac{1}{2}\sum_{i=2}^d t_{\boldsymbol{x}_i}.$$
$$\text{(E.3)}$$

We can rearrange the second constraint to

$$\frac{2\mathcal{C}_1}{(\sum_{\boldsymbol{z}\in\mathcal{Z}} t_{\boldsymbol{z}})^{1-\alpha}} + \frac{2\mathcal{C}_2}{\sum_{\boldsymbol{z}\in\mathcal{Z}} t_{\boldsymbol{z}}} \geq \frac{\sum_{i=2}^d t_{\boldsymbol{x}_i}}{\sum_{\boldsymbol{z}\in\mathcal{Z}} t_{\boldsymbol{z}}}.$$

Assume that the optimal value of (E.3) satisfies $\sum_{\boldsymbol{z}\in\mathcal{Z}} t_{\boldsymbol{z}} \geq \frac{d}{\Delta^2}$, then this constraint can be weakened to

$$\zeta := \frac{2\mathcal{C}_1}{(d/\Delta^2)^{1-\alpha}} + \frac{2\mathcal{C}_2\Delta^2}{d} \geq \frac{\sum_{i=2}^d t_{\boldsymbol{x}_i}}{\sum_{\boldsymbol{z}\in\mathcal{Z}} t_{\boldsymbol{z}}}.$$

It follows then that if the optimal value to

$$\min \sum_{\boldsymbol{z}\in\mathcal{Z}} t_{\boldsymbol{z}} \quad \text{s.t.} \quad \min_{\boldsymbol{\theta}\in\Theta_{\mathrm{alt}}} \sum_{\boldsymbol{z}\in\mathcal{Z}} t_{\boldsymbol{z}}\mathrm{KL}(\nu_{\boldsymbol{\theta}_\star,\boldsymbol{z}}||\nu_{\boldsymbol{\theta},\boldsymbol{z}}) \geq \log\frac{1}{2.4\delta}, \zeta \geq \frac{\sum_{i=2}^d t_{\boldsymbol{x}_i}}{\sum_{\boldsymbol{z}\in\mathcal{Z}} t_{\boldsymbol{z}}} \qquad \text{(E.4)}$$

is at least $d/\Delta^2$, then the optimal value to (E.3) is also at least $d/\Delta^2$, so our assumption that $\sum_{\boldsymbol{z}\in\mathcal{Z}} t_{\boldsymbol{z}} \geq \frac{d}{\Delta^2}$ will be justified.

For $\boldsymbol{z}\neq\boldsymbol{z}^\star$, let $\boldsymbol{\theta}_{\boldsymbol{z}}(\epsilon,t)$ denote the instance

$$\boldsymbol{\theta}_\star - \frac{(\boldsymbol{y}_{\boldsymbol{z}}^\top\boldsymbol{\theta}_\star + \epsilon)\widetilde{\mathbf{A}}(t)^{-1}\boldsymbol{y}_{\boldsymbol{z}}}{\boldsymbol{y}_{\boldsymbol{z}}^\top\widetilde{\mathbf{A}}(t)^{-1}\boldsymbol{y}_{\boldsymbol{z}}}$$

for $\boldsymbol{y}_{\boldsymbol{z}} = \boldsymbol{z}^\star - \boldsymbol{z}$, $\widetilde{\mathbf{A}}(t) = \sum_{\boldsymbol{z}\in\mathcal{Z}}\frac{t_{\boldsymbol{z}}}{\sum_{\boldsymbol{z}'\in\mathcal{Z}} t_{\boldsymbol{z}'}}\boldsymbol{z}\boldsymbol{z}^\top + \mathrm{diag}([\xi^2, \gamma^2/d, \ldots, \gamma^2/d])$, and $\epsilon \leq \min\{\Delta, \xi\}$. Note that $\boldsymbol{y}_{\boldsymbol{z}}^\top\boldsymbol{\theta}_{\boldsymbol{z}}(\epsilon,t) = -\epsilon < 0$ which implies that $\boldsymbol{\theta}_{\boldsymbol{z}}(\epsilon,t)\in\Theta_{\mathrm{alt}}$. Furthermore, we can bound:

**Claim E.2.** *For all $\boldsymbol{z},\boldsymbol{v}\in\mathcal{Z}$,*

$$\mathrm{KL}(\nu_{\boldsymbol{\theta}_\star,\boldsymbol{v}}||\nu_{\boldsymbol{\theta}_{\boldsymbol{z}}(\epsilon,t),\boldsymbol{v}}) \leq 16(\boldsymbol{y}_{\boldsymbol{z}}^\top\boldsymbol{\theta}_\star + \epsilon)^2 \frac{\boldsymbol{y}_{\boldsymbol{z}}^\top\widetilde{\mathbf{A}}(t)^{-1}\boldsymbol{v}\boldsymbol{v}^\top\widetilde{\mathbf{A}}(t)^{-1}\boldsymbol{y}_{\boldsymbol{z}}}{(\boldsymbol{y}_{\boldsymbol{z}}^\top\widetilde{\mathbf{A}}(t)^{-1}\boldsymbol{y}_{\boldsymbol{z}})^2}.$$

This implies that, for any $t$,

$$\sum_{\boldsymbol{v}\in\mathcal{Z}} t_{\boldsymbol{v}}\mathrm{KL}(\nu_{\boldsymbol{\theta}_\star,\boldsymbol{v}}||\nu_{\boldsymbol{\theta}_{\boldsymbol{z}}(\epsilon,t),\boldsymbol{v}}) \leq 16\sum_{\boldsymbol{v}\in\mathcal{Z}} t_{\boldsymbol{v}}(\boldsymbol{y}_{\boldsymbol{z}}^\top\boldsymbol{\theta}_\star + \epsilon)^2 \frac{\boldsymbol{y}_{\boldsymbol{z}}^\top\widetilde{\mathbf{A}}(t)^{-1}\boldsymbol{v}\boldsymbol{v}^\top\widetilde{\mathbf{A}}(t)^{-1}\boldsymbol{y}_{\boldsymbol{z}}}{(\boldsymbol{y}_{\boldsymbol{z}}^\top\widetilde{\mathbf{A}}(t)^{-1}\boldsymbol{y}_{\boldsymbol{z}})^2}$$

$$= 16\sum_{\boldsymbol{v}\in\mathcal{Z}} t_{\boldsymbol{v}}\cdot(\boldsymbol{y}_{\boldsymbol{z}}^\top\boldsymbol{\theta}_\star + \epsilon)^2 \frac{\boldsymbol{y}_{\boldsymbol{z}}^\top\widetilde{\mathbf{A}}(t)^{-1}(\sum_{\boldsymbol{v}\in\mathcal{Z}}\frac{t_{\boldsymbol{v}}}{\sum_{\boldsymbol{v}'\in\mathcal{Z}} t_{\boldsymbol{v}'}}\boldsymbol{v}\boldsymbol{v}^\top)\widetilde{\mathbf{A}}(t)^{-1}\boldsymbol{y}_{\boldsymbol{z}}}{(\boldsymbol{y}_{\boldsymbol{z}}^\top\widetilde{\mathbf{A}}(t)^{-1}\boldsymbol{y}_{\boldsymbol{z}})^2}$$

$$\leq 16\sum_{\boldsymbol{v}\in\mathcal{Z}} t_{\boldsymbol{v}}\cdot(\boldsymbol{y}_{\boldsymbol{z}}^\top\boldsymbol{\theta}_\star + \epsilon)^2 \frac{\boldsymbol{y}_{\boldsymbol{z}}^\top\widetilde{\mathbf{A}}(t)^{-1}\widetilde{\mathbf{A}}(t)\widetilde{\mathbf{A}}(t)^{-1}\boldsymbol{y}_{\boldsymbol{z}}}{(\boldsymbol{y}_{\boldsymbol{z}}^\top\widetilde{\mathbf{A}}(t)^{-1}\boldsymbol{y}_{\boldsymbol{z}})^2}$$

$$= \sum_{\boldsymbol{v}\in\mathcal{Z}} t_{\boldsymbol{v}}\cdot\frac{16(\boldsymbol{y}_{\boldsymbol{z}}^\top\boldsymbol{\theta}_\star + \epsilon)^2}{\|\boldsymbol{y}_{\boldsymbol{z}}\|_{\widetilde{\mathbf{A}}(t)^{-1}}^2}$$

Thus:

$$(\text{E.4}) \geq \min_{\boldsymbol{v}\in\mathcal{Z}} \sum t_{\boldsymbol{v}} \quad \text{s.t.} \quad \min_{\boldsymbol{z}\neq\boldsymbol{z}^\star} \sum_{\boldsymbol{v}\in\mathcal{Z}} t_{\boldsymbol{v}} \text{KL}(\nu_{\boldsymbol{\theta}_\star,\boldsymbol{v}}||\nu_{\boldsymbol{\theta}_{\boldsymbol{z}}(\epsilon,t),\boldsymbol{v}}) \geq \log\frac{1}{2.4\delta}, \zeta \geq \frac{\sum_{i=2}^d t_{\boldsymbol{x}_i}}{\sum_{\boldsymbol{v}\in\mathcal{Z}} t_{\boldsymbol{v}}}$$

$$\geq \min_{\boldsymbol{v}\in\mathcal{Z}} \sum t_{\boldsymbol{v}} \quad \text{s.t.} \quad \sum_{\boldsymbol{v}\in\mathcal{Z}} t_{\boldsymbol{v}} \geq \max_{\boldsymbol{z}\neq\boldsymbol{z}^\star} \frac{\|\boldsymbol{y}_{\boldsymbol{z}}\|_{\widetilde{\mathbf{A}}(t)^{-1}}^2}{16(\boldsymbol{y}_{\boldsymbol{z}}^\top\boldsymbol{\theta}_\star + \epsilon)^2} \cdot \log\frac{1}{2.4\delta}, \zeta \geq \frac{\sum_{i=2}^d t_{\boldsymbol{x}_i}}{\sum_{\boldsymbol{v}\in\mathcal{Z}} t_{\boldsymbol{v}}}$$

$$= \inf_{\lambda\in\widetilde{\triangle}} \max_{\boldsymbol{z}\neq\boldsymbol{z}^\star} \frac{\|\boldsymbol{y}_{\boldsymbol{z}}\|_{\widetilde{\mathbf{A}}(\lambda)^{-1}}^2}{16(\boldsymbol{y}_{\boldsymbol{z}}^\top\boldsymbol{\theta}_\star + \epsilon)^2} \cdot \log\frac{1}{2.4\delta}$$

where $\widetilde{\mathbf{A}}(\lambda) = \sum_{\boldsymbol{z}\in\mathcal{Z}} \lambda_{\boldsymbol{z}}\boldsymbol{z}\boldsymbol{z}^\top$ and $\widetilde{\triangle} = \{\lambda \in \triangle_{\mathcal{Z}} : \zeta \geq \sum_{i=2}^d \lambda_{\boldsymbol{x}_i}\}$. We can further lower bound this by

$$\geq \inf_{\lambda\in\widetilde{\triangle}} \max_{i\geq 2} \frac{\|\boldsymbol{z}^\star - \boldsymbol{x}_i\|_{\widetilde{\mathbf{A}}(\lambda)^{-1}}^2}{16((\boldsymbol{z}^\star - \boldsymbol{x}_i)^\top\boldsymbol{\theta}_\star + \epsilon)^2} \cdot \log\frac{1}{2.4\delta}$$

$$= \inf_{\lambda\in\widetilde{\triangle}} \max_{i\geq 2} \frac{\|\Delta\boldsymbol{e}_1 - \gamma\boldsymbol{e}_i\|_{\widetilde{\mathbf{A}}(\lambda)^{-1}}^2}{16(\Delta + \epsilon)^2} \cdot \log\frac{1}{2.4\delta}.$$

By Lemma E.4, we have

$$\inf_{\lambda\in\widetilde{\triangle}} \max_{i\geq 2} \|\Delta\boldsymbol{e}_1 - \gamma\boldsymbol{e}_i\|_{\widetilde{\mathbf{A}}(\lambda)^{-1}}^2$$

$$\geq \inf_{\lambda\in\triangle_d} \max_{i\geq 2} (\Delta\boldsymbol{e}_1 - \gamma\boldsymbol{e}_i)^\top \Big(2\xi^2\boldsymbol{e}_1\boldsymbol{e}_1^\top + 2\max\{\zeta,\gamma^2\}\lambda_i\boldsymbol{e}_i\boldsymbol{e}_i^\top + \text{diag}([\xi^2, \gamma^2/d, \dots, \gamma^2/d])\Big)^{-1}(\Delta\boldsymbol{e}_1 - \gamma\boldsymbol{e}_i)$$

$$\geq \inf_{\lambda\in\triangle_d} \max_{i\geq 2} (\Delta\boldsymbol{e}_1 - \gamma\boldsymbol{e}_i)^\top \Big(3\xi^2\boldsymbol{e}_1\boldsymbol{e}_1^\top + (2\max\{\zeta,\gamma^2\}\lambda_i + \gamma^2/d)\boldsymbol{e}_i\boldsymbol{e}_i^\top\Big)^{-1}(\Delta\boldsymbol{e}_1 - \gamma\boldsymbol{e}_i)$$

$$= \frac{\Delta^2}{3\xi^2} + \inf_{\lambda\in\triangle_d} \max_{i\geq 2} \frac{1}{2\lambda_i + 1/d}$$

where in the final equality we have used that $\zeta \leq \gamma^2$. However, this is clearly minimized by choosing $\lambda_i = 1/(d-1)$, which gives a lower bound of

$$\frac{1}{2/(d-1) + 1/d} \geq \frac{d-1}{3}.$$

Putting all of this together, we have shown that any feasibly solution $(t_{\boldsymbol{z}})_{\boldsymbol{z}\in\mathcal{Z}}$ to (E.3) must satisfy

$$\sum_{\boldsymbol{z}\in\mathcal{Z}} t_{\boldsymbol{z}} \geq \frac{d-1}{48(\Delta + \epsilon)^2} \cdot \log\frac{1}{2.4\delta}.$$

Using that any feasible solution to (E.3) lower bounds $\sum_{\boldsymbol{z}\in\mathcal{Z}} \mathbb{E}[T(\boldsymbol{z})]$ and taking $\epsilon \to 0$ gives the result.

$\square$

**Lemma E.3.** *Take some $\Delta > 0$ satisfying:*

$$\Delta \leq \min\left\{\frac{1}{2704d^2}, \sqrt{\frac{1}{10816\mathcal{C}_2}}, \left(\frac{1}{10816d^\alpha\mathcal{C}_1}\right)^{\frac{1}{2(1-\alpha)}}\right\}$$

*and set*

$$\xi = \frac{1}{52d}, \qquad \gamma = \max\left\{\frac{2\mathcal{C}_1}{(d/\Delta^2)^{1-\alpha}} + \frac{2\mathcal{C}_2\Delta^2}{d}, d\Delta\right\}.$$

*Then this choice of $\xi, \gamma, \delta$ satisfies (E.1) and, furthermore, $\|\boldsymbol{z}\|_2 \leq 1$ for all $\boldsymbol{z} \in \mathcal{Z}$.*

*Proof.* To satisfy (E.1), we must have $\frac{1}{52\sqrt{d}} \geq \gamma$. Thus, if

$$\Delta \leq \frac{1}{2704d^2}, \quad \Delta \leq \sqrt{\frac{1}{10816\mathcal{C}_2}}, \quad \Delta \leq \left(\frac{1}{10816d^\alpha\mathcal{C}_1}\right)^{\frac{1}{2(1-\alpha)}},$$

some computation shows that choosing $\gamma$ as prescribed will meet the constraint $\frac{1}{52\sqrt{d}} \geq \gamma$ and will also satisfy $\gamma \geq \sqrt{d}\Delta$. The norm bound follows by our choice of $\xi$ and since $\xi \geq \gamma/\sqrt{d} \geq \sqrt{\Delta}$. $\square$

### E.1.1 Additional Proofs

*Proof of Claim E.2.* We first show that $|\langle \boldsymbol{\theta}_{\boldsymbol{z}}(\epsilon, t), \boldsymbol{v}\rangle| \le 13d\xi$ for all $\boldsymbol{z}, \boldsymbol{v} \in \mathcal{Z}$. Note that for all $\boldsymbol{v} \in \mathcal{Z}, |\langle \boldsymbol{v}, \boldsymbol{\theta}_\star\rangle| \le \xi$.

**Case 1: $\boldsymbol{z} = \boldsymbol{z}^\star$.** In this case, $\langle \boldsymbol{y}_{\boldsymbol{z}}, \boldsymbol{\theta}_\star\rangle = 0$ so the result follows from our condition on $\epsilon$.

**Case 2: $\boldsymbol{z} = \boldsymbol{e}_i, i \ge 2$.** Let $\widetilde{\triangle} = \{\lambda \in \triangle_\mathcal{Z} : \zeta \ge \sum_{i=2}^d \lambda_{\boldsymbol{x}_i}\}$. In this case, $\langle \boldsymbol{z}, \boldsymbol{\theta}_\star\rangle = 0$ and $\langle \boldsymbol{y}_{\boldsymbol{z}}, \boldsymbol{\theta}_\star\rangle = \xi$. Furthermore, by Lemma E.4,

$$
\begin{aligned}
\boldsymbol{y}_{\boldsymbol{z}}^\top \widetilde{\mathbf{A}}(t)^{-1} \boldsymbol{y}_{\boldsymbol{z}} &\ge \inf_{\lambda \in \widetilde{\triangle}} \boldsymbol{y}_{\boldsymbol{z}}^\top \left( 2 \sum_{\boldsymbol{z}' \in \mathcal{Z}} \lambda_{\boldsymbol{z}'} \mathrm{diag}([(\boldsymbol{z}')^2]) + \mathrm{diag}([\xi^2, \gamma^2/d, \ldots, \gamma^2/d]) \right)^{-1} \boldsymbol{y}_{\boldsymbol{z}} \\
&\ge \boldsymbol{y}_{\boldsymbol{z}}^\top \left( 2\xi^2 \boldsymbol{e}_1 \boldsymbol{e}_1^\top + 2 \max\{\zeta, \gamma^2\} \boldsymbol{e}_i \boldsymbol{e}_i^\top + \mathrm{diag}([\xi^2, \gamma^2/d, \ldots, \gamma^2/d]) \right)^{-1} \boldsymbol{y}_{\boldsymbol{z}} \\
&\ge \xi^2 \frac{3}{\xi^2} + \frac{1}{2 \max\{\zeta, \gamma^2\} + \gamma^2/d} \\
&= 3 + \frac{1}{2 \max\{\zeta, \gamma^2\} + \gamma^2/d} \\
&\ge \frac{1}{3\gamma^2}
\end{aligned}
$$

where the last inequality follows from our assumption that $\zeta \le \gamma^2$. In the other direction, we can bound

$$
\boldsymbol{v}^\top \widetilde{\mathbf{A}}(t)^{-1} \boldsymbol{y}_{\boldsymbol{z}} \le \boldsymbol{v}^\top \mathrm{diag}([\xi^2, \gamma^2/d, \ldots, \gamma^2/d])^{-1} \boldsymbol{y}_{\boldsymbol{z}} \le \frac{1}{\xi^2} + \frac{d}{\gamma^2} \le \frac{2d}{\gamma^2}
$$

where the last inequality follows by our assumption that $\xi \ge \gamma/\sqrt{d}$. Putting this together, we have

$$
|\langle \boldsymbol{\theta}_{\boldsymbol{z}}(\epsilon, t), \boldsymbol{v}\rangle| \le \xi + (\xi + \epsilon) \frac{2d/\gamma^2}{1/3\gamma^2} \le 13d\xi.
$$

**Case 3: $\boldsymbol{z} = \boldsymbol{x}_i$.** In this case $\langle \boldsymbol{y}_{\boldsymbol{z}}, \boldsymbol{\theta}_\star\rangle = \Delta$. We can apply a calculation analogous to above to lower bound $\boldsymbol{y}_{\boldsymbol{z}}^\top \widetilde{\mathbf{A}}(t)^{-1} \boldsymbol{y}_{\boldsymbol{z}}$, but in this case obtain

$$
\boldsymbol{y}_{\boldsymbol{z}}^\top \widetilde{\mathbf{A}}(t)^{-1} \boldsymbol{y}_{\boldsymbol{z}} \ge \Delta^2 \frac{3}{\xi^2} + \frac{\gamma^2}{2 \max\{\zeta, \gamma^2\} + \gamma^2/d} \ge \frac{1}{3}.
$$

Similarly, we can upper bound

$$
\boldsymbol{v}^\top \widetilde{\mathbf{A}}(t)^{-1} \boldsymbol{y}_{\boldsymbol{z}} \le \boldsymbol{v}^\top \mathrm{diag}([\xi^2, \gamma^2/d, \ldots, \gamma^2/d])^{-1} \boldsymbol{y}_{\boldsymbol{z}} \le \frac{\Delta}{\xi^2} + \frac{d\gamma}{\gamma^2} \le \frac{2d}{\gamma}.
$$

This gives a final upper bound of

$$
|\langle \boldsymbol{\theta}_{\boldsymbol{z}}(\epsilon, t), \boldsymbol{v}\rangle| \le \xi + (\Delta + \epsilon) \frac{6d}{\gamma} \le \xi + \frac{12d\Delta}{\sqrt{\Delta}} \le 13d\xi.
$$

Combining these three cases gives that $|\langle \boldsymbol{\theta}_{\boldsymbol{z}}(\epsilon, t), \boldsymbol{v}\rangle| \le 13d\xi$ for all $\boldsymbol{z}, \boldsymbol{v} \in \mathcal{Z}$. By our assumption that $\xi \le \frac{1}{52d}$, it follows that $|\langle \boldsymbol{\theta}_{\boldsymbol{z}}(\epsilon, t), \boldsymbol{v}\rangle| \le 1/4$ for all $\boldsymbol{z}, \boldsymbol{v} \in \mathcal{Z}$.

By Lemma D.2 of Wagenmaker et al. (2022), as long as $\langle \boldsymbol{\theta}_{\boldsymbol{z}}(\epsilon, t), \boldsymbol{v}\rangle + 1/2 \in (0, 1)$ and $\langle \boldsymbol{\theta}_\star, \boldsymbol{v}\rangle + 1/2 \in (0, 1)$, which will be the case by the definition of $\boldsymbol{\theta}_\star$ and since $|\langle \boldsymbol{\theta}_{\boldsymbol{z}}(\epsilon, t), \boldsymbol{v}\rangle| \le 1/4$ as noted above, we have

$$
\mathrm{KL}(\nu_{\boldsymbol{\theta}_\star, \boldsymbol{v}} || \nu_{\boldsymbol{\theta}_{\boldsymbol{z}}(\epsilon, t), \boldsymbol{v}}) \le \frac{\langle \boldsymbol{\theta}_{\boldsymbol{z}}(\epsilon, t) - \boldsymbol{\theta}_\star, \boldsymbol{v}\rangle^2}{(\langle \boldsymbol{\theta}_{\boldsymbol{z}}(\epsilon, t), \boldsymbol{v}\rangle + 1/2)(1/2 - \langle \boldsymbol{\theta}_{\boldsymbol{z}}(\epsilon, t), \boldsymbol{v}\rangle)}.
$$

Using what we have just shown, we can upper bound this as

$$
\begin{aligned}
\frac{\langle \boldsymbol{\theta}_{\boldsymbol{z}}(\epsilon, t) - \boldsymbol{\theta}_\star, \boldsymbol{v}\rangle^2}{(\langle \boldsymbol{\theta}_{\boldsymbol{z}}(\epsilon, t), \boldsymbol{v}\rangle + 1/2)(1/2 - \langle \boldsymbol{\theta}_{\boldsymbol{z}}(\epsilon, t), \boldsymbol{v}\rangle)} &\le \frac{\langle \boldsymbol{\theta}_{\boldsymbol{z}}(\epsilon, t) - \boldsymbol{\theta}_\star, \boldsymbol{v}\rangle^2}{(-1/4 + 1/2)(1/2 - 1/4)} \\
&= 16 \langle \boldsymbol{\theta}_{\boldsymbol{z}}(\epsilon, t) - \boldsymbol{\theta}_\star, \boldsymbol{v}\rangle^2.
\end{aligned}
$$

By our choice of $\boldsymbol{\theta_z}(\epsilon, t)$, this is equal to:

$$16(\boldsymbol{y_z}^\top \boldsymbol{\theta_\star} + \epsilon)^2 \frac{\boldsymbol{y_z}^\top \widetilde{\mathbf{A}}(t)^{-1} \boldsymbol{vv}^\top \widetilde{\mathbf{A}}(t)^{-1} \boldsymbol{y_z}}{(\boldsymbol{y_z}^\top \widetilde{\mathbf{A}}(t)^{-1} \boldsymbol{y_z})^2}$$

which completes the proof. $\qquad\square$

**Lemma E.4.**

$$\sum_{\boldsymbol{z} \in \mathcal{Z}} \lambda_{\boldsymbol{z}} \boldsymbol{z}\boldsymbol{z}^\top \preceq 2 \sum_{\boldsymbol{z} \in \mathcal{Z}} \lambda_{\boldsymbol{z}} \mathrm{diag}(\boldsymbol{z}^2).$$

*Proof.* This follows since every $\boldsymbol{z} \in \mathcal{Z}$ has at most two non-zero entries, and since $(a\boldsymbol{x} + b\boldsymbol{y})(a\boldsymbol{x} + b\boldsymbol{y})^\top \preceq 2a^2\boldsymbol{x}\boldsymbol{x}^\top + 2b^2\boldsymbol{y}\boldsymbol{y}^\top$. $\qquad\square$

### E.2 Mapping to Linear MDPs

We can map this linear bandit (with parameters chose as in Lemma E.3) to a linear MDP with state space $\mathcal{S} = \{s_0, s_1, \bar{s}_2, \ldots, \bar{s}_{d+1}\}$, action space $\mathcal{A} = \mathcal{Z} \cup \{\boldsymbol{e}_{d+1}/2\}$, parameters

$$\boldsymbol{\theta}_1 = \boldsymbol{0}, \quad \boldsymbol{\theta}_2 = \boldsymbol{e}_1$$

$$\boldsymbol{\mu}_1(s_1) = [2\boldsymbol{\theta}_\star, 1], \quad \boldsymbol{\mu}_1(\bar{s}_i) = \frac{1}{d}[-2\boldsymbol{\theta}_\star, 1],$$

and feature vectors

$$\boldsymbol{\phi}(s_0, \boldsymbol{e}_{d+1}) = \boldsymbol{e}_{d+1}/2, \quad \boldsymbol{\phi}(s_0, \boldsymbol{z}) = [\boldsymbol{z}/2, 1/2], \quad \forall \boldsymbol{z} \in \mathcal{Z}$$
$$\boldsymbol{\phi}(s_1, \boldsymbol{z}) = \boldsymbol{e}_1, \quad \boldsymbol{\phi}(\bar{s}_i, \boldsymbol{z}) = \boldsymbol{e}_i, i \geq 2, \quad \forall \boldsymbol{z} \in \mathcal{A}.$$

Note that, if we take action $\boldsymbol{z}$ in state $s_0$, our expected episode reward is

$$P_1(s_1|s_0, \boldsymbol{z}) \cdot 1 + \sum_{i=2}^{d+1} P_1(\bar{s}_i|s_0, \boldsymbol{z}) \cdot 0 = \langle \boldsymbol{\theta}_\star, \boldsymbol{z} \rangle + 1/2$$

since we always acquire a reward of 1 in any state $s_1$, and a reward of 0 in any state $\bar{s}_i$, and the reward distribution is Bernoulli.

**Lemma E.5.** *The MDP constructed above is a valid linear MDP as defined in Definition 3.1.*

*Proof.* For $\boldsymbol{z} \in \mathcal{Z}$ we have,

$$P_1(s_1|s_0, \boldsymbol{z}) = \langle \boldsymbol{\phi}(s_0, \boldsymbol{z}), \boldsymbol{\mu}_1(s_1) \rangle = \langle \boldsymbol{\theta}_\star, \boldsymbol{z} \rangle + 1/2 \geq 0$$
$$P_1(\bar{s}_i|s_0, \boldsymbol{z}) = \langle \boldsymbol{\phi}(s_0, \boldsymbol{z}), \boldsymbol{\mu}_1(\bar{s}_i) \rangle = \frac{1}{d}(-\langle \boldsymbol{\theta}_\star, \boldsymbol{z} \rangle + 1/2) \geq 0$$

where the inequality follows since $|\langle \boldsymbol{\theta}_\star, \boldsymbol{z} \rangle| \leq \mathcal{O}(1/d)$ for all $\boldsymbol{z} \in \mathcal{Z}$. In addition,

$$P_1(s_1|s_0, \boldsymbol{z}) + \sum_{i=2}^{d+1} P_1(\bar{s}_i|s_0, \boldsymbol{z}) = \langle \boldsymbol{\theta}_\star, \boldsymbol{z} \rangle + 1/2 + d \cdot \frac{1}{d}(-\langle \boldsymbol{\theta}_\star, \boldsymbol{z} \rangle + 1/2) = 1.$$

Thus, $P_1(\cdot|s_0, \boldsymbol{z})$ is a valid probability distribution for $\boldsymbol{z} \in \mathcal{Z}$. A similar calculation shows the same for $\boldsymbol{z} = \boldsymbol{e}_{d+1}/2$.

It remains to check the normalization bounds. Clearly, by our construction of $\mathcal{Z}$, $\|\boldsymbol{\phi}(s, a)\|_2 \leq 1$ for all $s$ and $a$. It is also obvious that $\|\boldsymbol{\theta}_0\|_2 \leq \sqrt{d}$ and $\|\boldsymbol{\theta}_1\|_2 \leq \sqrt{d}$. Finally,

$$\||\boldsymbol{\mu}_1(\mathcal{S})|\|_2 = \left\| \sum_{s \in \mathcal{S} \setminus s_0} |\boldsymbol{\mu}_1(s)| \right\|_2 = \|[2\boldsymbol{\theta}_\star, 1] + d \cdot \frac{1}{d}[2\boldsymbol{\theta}_\star, 1]\|_2 \leq \sqrt{d}.$$

Thus, all normalization bounds are met, so this is a valid linear MDP. $\qquad\square$

*Proof of Proposition* 2. If we assume that the learner has prior access to the feature vectors, and also knows this is a linear MDP, then, even with no knowledge of the dynamics, we can guarantee an optimal policy is contained in the set of policies $\pi^{\boldsymbol{z},\boldsymbol{z}'}$ defined as:

$$\pi_1^{\boldsymbol{z},\boldsymbol{z}'}(s_0) = \boldsymbol{z}, \pi_2^{\boldsymbol{z},\boldsymbol{z}'}(s_0) = \boldsymbol{z}', \pi_h^{\boldsymbol{z},\boldsymbol{z}'}(s_1) = \xi\boldsymbol{e}_1, \pi_h^{\boldsymbol{z},\boldsymbol{z}'}(\bar{s}_i) = \xi\boldsymbol{e}_1$$

This holds because in states $s_1$ and $\bar{s}_i$, the performance of each action is identical since the feature vectors are identical, so it doesn't matter which action we choose in these states. In this case, we can bound $|\Pi| \leq |\mathcal{Z}|^2 \leq 4d^2$.

Now, for $\boldsymbol{z} \in \mathcal{A}$, $\boldsymbol{z} \neq \boldsymbol{e}_{d+1}/2$, we have

$$\boldsymbol{\phi}_{\pi^{\boldsymbol{z},\boldsymbol{z}'},1} = [\boldsymbol{z}/2, 1/2]$$

$$\boldsymbol{\phi}_{\pi^{\boldsymbol{z},\boldsymbol{z}'},2} = (\langle\boldsymbol{\theta}_\star,\boldsymbol{z}\rangle + 1/2)\boldsymbol{e}_1 + \frac{1}{d}(-\langle\boldsymbol{\theta}_\star,\boldsymbol{z}\rangle + 1/2)\sum_{i\geq 2}\boldsymbol{e}_i$$

and if $\boldsymbol{z} = \boldsymbol{e}_{d+1}/2$, $\boldsymbol{\phi}_{\pi^{\boldsymbol{z},\boldsymbol{z}'},1} = \boldsymbol{e}_{d+1}/2$, $\boldsymbol{\phi}_{\pi^{\boldsymbol{z},\boldsymbol{z}'},2} = \boldsymbol{e}_1/2 + \frac{1}{2d}\sum_{i\geq 2}\boldsymbol{e}_i$. Let $\pi_{\exp}$ be the policy that plays action $\boldsymbol{e}_2$ in state $s_0$ at step $h = 1$. Then,

$$\boldsymbol{\Lambda}_{\pi_{\exp},2} = \frac{1}{2}\boldsymbol{e}_1\boldsymbol{e}_1 + \frac{1}{2d}\sum_{i\geq 3}\boldsymbol{e}_i\boldsymbol{e}_i^\top.$$

Since $\langle\boldsymbol{\theta}_\star,\boldsymbol{z}\rangle \leq \mathcal{O}(1/d)$ and $[\boldsymbol{z}]_1 \leq \mathcal{O}(1/d)$ for all $\boldsymbol{z}$ by construction, it follows that we can bound, for all $\boldsymbol{z},\boldsymbol{z}'$,

$$\|\boldsymbol{\phi}_{\pi^{\boldsymbol{z},\boldsymbol{z}'},2}\|_{\boldsymbol{\Lambda}_{\pi_{\exp},2}^{-1}}^2 = \mathcal{O}\left(1 + \sum_{i\geq 2}\frac{1}{d^2}\cdot d\right) = \mathcal{O}(1)$$

so

$$\inf_{\pi_{\exp}}\max_{\pi\in\Pi}\frac{\|\boldsymbol{\phi}_{\pi,2}\|_{\boldsymbol{\Sigma}_{\pi_{\exp},2}^{-1}}^2}{\max\{V_0^\star - V_0^\pi, \Delta_{\min}^\Pi, \epsilon\}^2} \leq \mathcal{O}(1/\epsilon^2).$$

Now let $\pi_{\exp}$ be the policy that, at step $h = 1$, plays $\xi\boldsymbol{e}_1$ with probability $1/4$, $\boldsymbol{e}_{d+1}$ with probability $1/4$, and plays $\boldsymbol{e}_i$ with probability $\frac{1}{4(d-1)}$ for $i \geq \{2,\ldots,d\}$. In this setting, we have

$$\boldsymbol{\Lambda}_{\pi_{\exp},1} = \frac{1}{4}\xi^2\boldsymbol{e}_1\boldsymbol{e}_1^\top + \frac{1}{4}\boldsymbol{e}_{d+1}\boldsymbol{e}_{d+1} + \frac{1}{4(d-1)}\sum_{i\in\{2,\ldots,d\}}\boldsymbol{e}_i\boldsymbol{e}_i^\top.$$

Note that $V_0^\star - V_0^{\pi^{\boldsymbol{z},\boldsymbol{z}'}} = \xi - \langle\boldsymbol{\theta}_\star,\boldsymbol{z}\rangle$, so for $\boldsymbol{z} = \boldsymbol{e}_2,\ldots,\boldsymbol{e}_{d+1}$, we have $V_0^\star - V_0^{\pi^{\boldsymbol{z},\boldsymbol{z}'}} = \xi = \mathcal{O}(1/d)$, while for $\boldsymbol{z} = \xi\boldsymbol{e}_1, \boldsymbol{x}_2,\ldots,\boldsymbol{x}_d$, we have $V_0^\star - V_0^{\pi^{\boldsymbol{z},\boldsymbol{z}'}} = \Delta$.

It's easy to see that for $\boldsymbol{z} = \boldsymbol{e}_2,\ldots,\boldsymbol{e}_{d+1}$, we have $\|\boldsymbol{\phi}_{\pi^{\boldsymbol{z},\boldsymbol{z}'},1}\|_{\boldsymbol{\Lambda}_{\pi_{\exp},1}^{-1}} \leq \mathcal{O}(d)$, and for $\boldsymbol{z} = \xi\boldsymbol{e}_1, \boldsymbol{x}_2,\ldots,\boldsymbol{x}_d$, $\|\boldsymbol{\phi}_{\pi^{\boldsymbol{z},\boldsymbol{z}'},1}\|_{\boldsymbol{\Lambda}_{\pi_{\exp},1}^{-1}} \leq \mathcal{O}(1 + d\gamma^2) = \mathcal{O}(1)$. Combining these bounds with the gap values, we conclude that

$$\inf_{\pi_{\exp}}\max_{\pi\in\Pi}\frac{\|\boldsymbol{\phi}_{\pi,1}\|_{\boldsymbol{\Sigma}_{\pi_{\exp},1}^{-1}}^2}{\max\{V_0^\star - V_0^\pi, \Delta_{\min}^\Pi, \epsilon\}^2} \leq \mathcal{O}(1/\epsilon^2 + \mathrm{poly}(d)).$$

The result then follows by Theorem 6. $\qquad\square$

**Lower bounding the performance of low-regret algorithms.** Assume that we have access to the linear bandit instance constructed in Appendix E.1 with parameters chosen as in Lemma E.3. That is, at every timestep $t$ we can choose an arm $\boldsymbol{z}_t \in \mathcal{Z}$ and obtain and observe reward $y_t \sim$ Bernoulli($\langle\boldsymbol{\theta}_\star,\boldsymbol{z}_t\rangle + 1/2$). Using the mapping up, we can use this bandit to simulate a linear MDP as follows:

1. Start in state $s_0$ and choose any action $\boldsymbol{z}_t \in \mathcal{A}$

2. Play action $z_t$ in our linear bandit. If reward obtained is $y_t = 1$, then in MDP transition to any of the states $s_1$. If reward obtained is $y_t = 0$ transition to any of the states $\bar{s}_2, \ldots, \bar{s}_{d+1}$, each with probability $1/d$. If the chosen action was $z_t = e_{d+1}/2$, then play any action in the linear bandit and transition to state $s_1$ with probability $1/2$ and $\bar{s}_2, \ldots, \bar{s}_{d+1}$ with probability $1/2d$, regardless of $y_t$

3. Take any action in the state in which you end up, and receive reward of 1 if you are in $s_1$, and reward of 0 if you are in $\bar{s}_2, \ldots, \bar{s}_{d+1}$.

Note that this MDP has precisely the transition and reward structure as the MDP constructed above.

**Lemma E.6.** *Assume $\pi$ is $\epsilon < \Delta/2$-optimal in the MDP constructed above. Then, $z^\star = \arg\max_{z \in \mathcal{A}} \pi_1(z|s_0)$.*

*Proof.* Note that the value of $\pi$ in the linear MDP is given by $V_0^\pi = \sum_{z \in \mathcal{Z}} \pi_1(z|s_0)(\langle z, \theta_\star \rangle + 1/2) + \pi_1(e_{d+1}/2|s_0)/2$ and the optimal policy is $\pi_1(z^\star|s_0) = 1$ and has value $V_0^\star = \langle z^\star, \theta_\star \rangle + 1/2$. It follows that if $\pi$ is $\epsilon$-optimal, then

$$\sum_{z \in \mathcal{Z}} \pi_1(z|s_0)(\langle z, \theta_\star \rangle + 1/2) + \pi_1(e_{d+1}/2|s_0)/2 \geq \langle z^\star, \theta_\star \rangle + 1/2 - \epsilon$$

$$\implies \pi_1(z^\star|s_0)(\xi + 1/2) + \sum_{z \in \mathcal{A}, z \neq z^\star} \pi_1(z|s_0)(\xi - \Delta + 1/2) \geq \xi + 1/2 - \epsilon$$

$$\implies -\Delta \sum_{z \in \mathcal{A}, z \neq z^\star} \pi_1(z|s_0) \geq -\epsilon$$

$$\implies \epsilon \geq \Delta \sum_{z \in \mathcal{A}, z \neq z^\star} \pi_1(z|s_0).$$

If $\epsilon < \Delta/2$, this implies that $\sum_{z \in \mathcal{A}, z \neq z^\star} \pi_1(z|s_0) < 1/2$, so it must be the case that $\pi_1(z^\star|s_0) > 1/2$. $\qquad\square$

*Proof of Proposition 3.* Consider running the above procedure for some number of steps. By Lemma E.6, if we can identify an $\epsilon < \Delta/2$-optimal policy in this MDP, we can use it to determine $z^\star$, the optimal arm in the linear bandit. As we have used no extra information other than samples from the linear bandit to construct this, it follows that to find an $\epsilon < \Delta/2$-optimal policy in the MDP, we must take at least the number of samples prescribed by Lemma E.1. $\qquad\square$