# OpenReview forum: "Instance-Dependent Near-Optimal Policy Identification in Linear MDPs via Online Experiment Design"
_NeurIPS.cc/2022/Conference — NeurIPS 2022 Accept_

### Official Review · Reviewer_sWno · 2022-06-29

**Rating:** 8
**Confidence:** 3
**Soundness:** 4 excellent
**Presentation:** 4 excellent
**Contribution:** 4 excellent

**Summary:**

This paper considers instance-dependent complexity of learning under linear MDPs. Based on a novel online experiment design in linear MDPs, the author proposes an algorithm whose sample complexity scaling with the instance-dependent complexity. This algorithm is worst-case optimal and beats any low-regret algorithm on an explicit example constructed by the author.

**Questions:**

- Is the result in Theorem 1 tight with respect to the policy gap and the uncertainty? Does a general lower bound exist? Does the upper and lower bound match?
- Can the algorithm be applied to linear mixture MDPs setting?


**Limitations:**

Lack of tightness w.r.t instance-dependence. Not tight w.r.t $H$ dependence. Computationally inefficient. The sample complexity result scales with the hardest-to-reach direction.

**Strengths And Weaknesses:**

Strengths:
- Interesting question. The question of obtaining worst-case optimal algorithms with linear function approximation has been well-studied recently. When it comes to instance-dependent RL, most of the recent work focuses on the tabular setting. Obtaining computationally efficient algorithms with tight instance-dependent sample complexity under linear MDPs is still an open question in RL.
- Great novelty. The author uses an online experiment design-based procedure to reduce the uncertainty of their algorithm.
- Enough comparison with the previous work. In Section 4.3, the author reduces their result to tabular and deterministic MDPs and carefully compares their results with the existing results.
- Well-written and easy to follow. Clear algorithms, definitions, assumptions and theorems.

Weaknesses:
- Lack of tightness. The paper does not show if the algorithm obtains tight instance-dependence.
- Computationally inefficient algorithm.

---

> ### Author Response · Authors · 2022-08-02
> **Reply to Reviewer sWno**
>
> We would like to thank the reviewer for their feedback and comments. We address the questions that were raised below.
>
> **Tightness:** Currently there does not exist a general instance-dependent lower bound, yet it is likely one could be obtained by appealing to arguments similar to those found in (Kaufmann et al., 2016). Even in the much simpler setting of best-policy identification in tabular MDPs, however, it is known that the instance-dependent lower bound is the solution to a non-convex optimization problem and is relatively uninterpretable (Marjani et al., 2021). Thus, we hold little hope that an instance-dependent lower bound in the linear MDP setting would provide helpful insights. In certain special cases, such as deterministic MDPs, instance-dependent lower bounds can be obtained, and as Corollary 3 shows, our algorithm hits such a lower bound (up to $H$ factors and constants). We would not necessarily expect Theorem 1 to match a general instance-dependent lower bound, however (though it does match the minimax lower bound).
>
> We remark that the majority of existing works in even the simpler bandit setting which do hit an instance-dependent lower bound only hit such a lower bound asymptotically (as $\delta \rightarrow 0$), usually relying on a “tracking” strategy to gradually learn and track the optimal allocation. While such algorithms are able to obtain tight results, over short (non-asymptotic) time horizons they usually perform quite poorly. The focus of our work was to instead obtain an intuitive, finite-time complexity, and a corresponding algorithm for which we would expect reasonable finite-time performance. Obtaining an algorithm which hits an instance-dependent lower bound and has reasonable finite-time complexity is an interesting direction for future work.
>
>
> **Linear Mixture MDP Setting:** Our current algorithm and analysis make use of the linear MDP structure explicitly, and as such don’t directly apply to linear mixture MDPs. However, we believe that a similar result could be obtained in the linear mixture MDP setting using techniques very similar to ours. In particular, our online experiment design technique is quite general and could be instantiated in a variety of different settings, and easily coupled with the policy elimination strategy PEDEL employs to obtain an instance-dependent complexity result.

---

> > ### Comment · Reviewer_sWno · 2022-08-08
> > **Response to the authors**
> >
> > Dear authors/reviewers/AC,
> >
> > I have read the response and other reviews. My major concern about the tightness is addressed by the author. And I would like to keep my positive rating.

---

### Official Review · Reviewer_b4C6 · 2022-07-16

**Rating:** 8
**Confidence:** 4
**Soundness:** 4 excellent
**Presentation:** 4 excellent
**Contribution:** 4 excellent

**Summary:**

This paper studies the problem of sample complexity of RL with linear function approximation. Recent works have extensively studied this problem with a range of results for different versions and structural assumptions. However, this paper presents the first instance dependent sample complexity analysis for the linear MDP setting. The proposed algorithm is based on a policy elimination scheme which iteratively maintains a set of policies which lie within a $2^{-l}$ optimality gap from the optimal policy. The update step for this algorithm is based on a novel online experiment-design procedure which guarantees the collection of a sufficiently 'exploratory' dataset of state-action pairs at each step.

In terms of the formal results, the authors show the first instance-dependent sample complexity analysis for linear MDPs which can be shown to also yield a dimension-optimal rate in the worst case. Further, it is formally shown that there exists instances where the instance-dependent rate improves over the sample complexity of worst case optimal PAC-RL algorithms. Lastly, it is shown that the instance-dependent rate for the linear case recovers the instance-dependent rate for deterministic MDPs whereas it gives another instance-dependent rate for tabular MDP which can be provably better or worse than previous instance-dependent rates for specific instances.

**Questions:**

Please see the comments about computational efficiency and $\lambda_{\min}^*$ above.
In addition, I was wondering if there is a more intuitive interpretation of the complete instance-dependent term in theorem 1 instead of just the individual terms.

**Limitations:**

The authors do discuss the potential improvements in the conclusion and formally show that their algorithm does not yield the best sample complexity for some tabular domains. As such, I do not have any concerns about the discussions of the limitations.

**Strengths And Weaknesses:**

Overall, this paper is very well written and shows important and novel results for linear MDPs. I especially like the thoroughness of this paper where multiple aspects of instance-dependence are explored:
1. Instance-dependent analysis for linear MDPs: The algorithm is well explained and a novel method for the experiment design objective in (4.1) is also proposed and analysed. In the final rate, the authors show that the rate depends on factors which relate to hardness of exploring any direction in the feature space $\phi$, a policy gap term for the input policy class. A final result is obtained by showing that a policy class can be constructed for linear MDPs whose size exponentially scales with $d$ to derive the instance-dependent guarantee with respect to the optimal policy.
2. Demonstrating the utility of the instance-dependent rate: The authors show using Prop. 3 that there exists linear MDPs such that the instance dependent rate improves over the worst case rate by a factor of $d^2$. Further, it is shown that no existing PAC-optimal algorithm will get a polylog(d) rate for this problem.
3. Discussions for tabular and deterministic MDPs: I especially like the result in Prop. 5 which shows that the sample comeplxity for their algorithm is in fact another possible instance-dependent rate for tabular domains and can improve over previously shown instance-adaptive algorithms or in some cases perform worse than them. It brings an interesting question of unifying these different instance-dependent measure of complexity, and highlights possibilities of improvement even for the linear MDP case. For deterministic MDPs, previously known rate can be recovered upto $H$ factors.

**Weakness**:
This is a strong first result in this direction and hence has certain aspects which can possibly be improved:
1. Computational efficiency: Since the algorithm follows a policy elimination template, for establishing a guarantee wrt optimal policy, an exponentially large policy set needs to be constructed. The experiment design objective then iterates over the expected feature vectors for each policy to eliminate sub-optimal candidates. As such, it is not clear if a computationally efficient version of the solution can be derived.
2. Dependence on explorability parameter: The authors need to assume that there exists a policy which generates a full rank covariance matrix for the state-action features. In my understanding, this also affects the algorithm as the covariance matrices constructed in the experiment design steps are guaranteed to be non-singular using this assumption. Can the authors comment about the necessity of this assumption or potential ways to remove this?

---

> ### Author Response · Authors · 2022-08-02
> **Reply to Reviewer b4C6**
>
> We would like to thank the reviewer for their feedback and comments. We address the questions that were raised below.
>
> **On the Necessity of the Explorability Assumption:** We do not believe the explorability assumption is necessary, and it could likely be removed with further work (though this may complicate the analysis significantly). Indeed, while the current algorithm and analysis obtains a $\mathrm{poly}(1/\lambda_{\min}^\star)$ dependence, a slightly different version of the algorithm reduces this to a $O(1/\lambda_{\min}^\star)$ dependence (at the expense of more complicated analysis). The final $O(1/\lambda_{\min}^\star)$ is currently necessary to ensure that our estimates of $\phi_{\pi,h}$ are accurate enough. In particular, if we do not collect covariates that are well-conditioned, it’s possible there are some “hard to reach” directions which we happen to reach, and which causes $\hat{\phi}\_{\pi,h}$ to be skewed in such directions, increasing the estimation error. Collecting covariates that are well-conditioned ensures this will not happen by guaranteeing we can control the error in each direction. However, we believe with a tighter analysis and possibly a slightly different estimator for $\hat{\phi}\_{\pi,h}$ this is removable. We leave fully removing this for future work.
>
> **Intuitive Explanation of the Complexity:** At a high level, the complexity can be thought of as quantifying the difficulty of exploring the MDP so as to eliminate suboptimal policies. The $\| \phi\_{\pi,h} \|\_{\Lambda\_{\mathrm{exp}}^{-1}}^2$ term quantifies the difficulty of collecting data necessary to eliminate policy $\pi$ (in particular, $\phi\_{\pi,h}$ is the direction we need to explore to learn about the performance of policy $\pi$, and the $\Lambda\_{\mathrm{exp}}^{-1}$ quantifies how easily we can reach directions in the MDP to do this), while $V_0^\star(\Pi) - V_0^\pi$ quantifies how suboptimal policy $\pi$ is, and therefore how many samples are needed to distinguish it from the optimal policy in the class (this can be thought of in a similar way to a gap in a bandit problem). Our complexity measure has a close resemblance to the complexity measure for best-arm identification in linear bandits found in (Soare et al., 2014; Fiez et al., 2019), but generalizes that measure to problems with longer horizon where navigation is required. We will offer a more intuitive explanation of our complexity in the final version.

---

> > ### Comment · Reviewer_b4C6 · 2022-08-08
> > **Rebuttal Acknowledgement**
> >
> > Dear authors/reviewers/AC,
> >
> > I've read the response and other reviews and feel confident about my positive rating.
> >
> > I recommend the authors to add the discussion about the explorability assumption and incorporate changes which clarify the other reviewer's ambiguities in the final version as well.

---

### Official Review · Reviewer_zzUK · 2022-07-16

**Rating:** 6
**Confidence:** 4
**Soundness:** 2 fair
**Presentation:** 3 good
**Contribution:** 3 good

**Summary:**

In this paper, the optimization of instance-dependent sample complexity for reinforcement learning (RL) under linear MDP is studied. The PAC policy learning setting is considered, which is to identify the best policy among all. An elimination-style algorithm with a novel exploration strategy is proposed. The key idea lies in using online Frank-Wolfe to explore the directions which reduce the sample variances most. The algorithm achieves the first instance-dependent sample complexity guarantee for linear MDP. The analysis also shows that low-regret algorithms can be sub-optimal in certain instances, showing the advantage of the proposed algorithm.

**Questions:**

Please feel free to point out the errors in the weaknesses that I proposed above.



**Limitations:**

The limitations and potential negative societal impact are properly discussed.

**Strengths And Weaknesses:**

Strengths:

1. The paper proposes an algorithm that is the first to achieve the instance-dependent sample complexity guarantee under linear MDP. The result is original and novel.

2. The exploration strategy based on online Frank-Wolfe is interesting and can be of independent interest for other bandit learning problems.

Weaknesses:

1. According to the algorithm, it is implicitly assumed that the policy set has finite cardinality. This assumption is not reasonable. Since there are infinite states in a linear MDP, the number of all policies should also be infinite. I think this is a major limitation of the proposed algorithm and analysis.

2. Another limitation is that the instance-dependent term in Theorem 1 does not have a clear explanation. It is not easy to think out which kind of linear MDP the proposed algorithm can be advantageous.

3. I also have some doubts about the exploration step. In Line 6 of Algorithm 1, the number of data to collect is set to $K_{h, l}$, which is closely related to the overall sample complexity, but there is no explanation of how this number is set. Furthermore, the sample complexity of Algorithm 2 is proposed in Theorem 6, which is of order $O(1/\epsilon_{exp}$. But there is no definition of $\epsilon_{exp}$ in the paper. Is $\epsilon_{exp}=\epsilon$? If this is true, I can't understand why the instance-dependent bound in Theorem 1 can hold, since as a step of Algorithm 1, Algorithm 2 has already required $O(1/\epsilon_{exp})$ instances.

4. I think some proofreading needs to be done to improve the notational explanations. There are some notations without defining previously or immediately after their first appearance.

Overall, I think the paper proposes interesting and novel results. But the weaknesses discussed above are not ignorable for me currently.

---
after rebuttal:

I increase my score since the responses address most of my doubts. I suggest including the explanations in the paper to improve clarity.

---

> ### Author Response · Authors · 2022-08-02
> **Reply to Reviewer zzUK**
>
> We would like to thank the reviewer for their feedback. We address the weaknesses pointed out below.
>
> 1. While the reviewer is correct that in general one might need a policy set of infinite cardinality to guarantee the set contains an optimal policy, we are only concerned with finding an $\epsilon$-optimal policy (the typical goal of PAC RL), and it is possible to construct a set of policies with finite cardinality guaranteed to contain an $\epsilon$-optimal policy for any linear MDP. We note this in Corollary 1—such a set can be constructed with log-cardinality $O(dH^2 \log 1/\epsilon)$. Technical details of this construction are provided in Lemma A.14. Thus, we believe that is is very reasonable to assume that our policy class has finite cardinality.
>
> 2. We remark that the MDP considered in Proposition 3 is one example where the instance-dependent term offers a significant improvement—rather than a complexity of $O(d^2/\epsilon^2)$, which would be the minimax rate, we obtain a complexity of $O(1/\epsilon^2)$. Thus, the instance-dependence allows us to save a factor of $d^2$ here. Another example is considered in Proposition 5 (the first bullet point), which shows that there are some problems where the instance-dependent complexity ends up scaling independent of $\epsilon$, yielding a significant improvement over the minimax rate.
> One can think of the instance-dependent term as capturing some measure of how easily each direction can be explored, normalized by how “important” that direction is for learning a near-optimal policy. This is similar to complexity measures that appear in the linear bandits literature, see e.g. (Soare et al., 2014; Fiez et al., 2019), but generalized to problems where navigation of the environment is required. We will offer a more intuitive explanation of the complexity in the final version of the paper (see also our answer to Reviewer b4C6 for more explanation).
>
> 3. At stage $\ell$, Algorithm 1 calls Algorithm 2 with $\epsilon_{\mathrm{exp}} \leftarrow \epsilon_\ell^2/\beta_\ell$ (this is made explicit in Algorithm 3 in the appendix). $K_{h,\ell}$ is then the total number of episodes Algorithm 2 runs for, and can be upper bounded by the complexity given in Theorem 6 (note that $K_{h,\ell}$ is random—we don’t know what it will be in advance, which is why it is not explicitly defined, but we can upper bound it by Theorem 6). The instance-dependent guarantee follows since the active policy set, $\Pi_\ell$, is shrinking as $\ell$ increases. Every time Algorithm 2 is called, it aims to explore the directions that the active policies are likely to reach. Therefore, as $\ell$ increases, the tolerance with which we explore in Algorithm 2 decreases, but the number of directions we need to explore decreases. Some calculation shows that this is sufficient to obtain the complexity given in Theorem 1 (see the proofs of Theorem 7 and Corollary 4 in the appendix for this calculation).
>
> 4. We will make sure to double check that all our notation is properly defined.

---

> > ### Comment · Reviewer_zzUK · 2022-08-08
> > **further question about the size of the policy set**
> >
> > Thanks very much for the responses. I have a further question. Corollary 1 states that there exists a finite size policy set to include an $\epsilon$-optimal policy. But can the learner construct this set before learning? I have checked the proof, is this set constructed by the $\epsilon$-net defined in Lemma A.14?

---

> > > ### Author Response · Authors · 2022-08-08
> > > **Size of policy set**
> > >
> > > That's correct. The policy set constructed in Lemma A.14 will contain an $\epsilon$-optimal policy for _any_ linear MDP with linear rewards. As such, a learner can construct this set without any prior knowledge of the MDP, and will be guaranteed to have some policy in the set that is $\epsilon$-optimal on the MDP of interest.
> > >
> > > The intuition behind this is that the optimal policy for any linear MDP will have a linear structure: $\pi^\star_h(s) = \mathrm{argmax}_{a} \langle \phi(s,a), w_h^\star \rangle$. As we don't know the value of $w_h^\star$, we cannot determine $\pi^\star_h(s)$ directly. However, if we cover the space of possible $w_h^\star$ vectors, then we will be guaranteed to have some vector in our set that is close to $w_h^\star$, and so induces a near-optimal policy. For technical reasons we instead use a softmax rather than a max, but the basic intuition of just covering all possible directions so there is one close to $w_h^\star$ still holds. Critically, this can be done without knowledge of $w_h^\star$---it just relies on the fact that we know the optimal policy has the linear form given above.

---

> > > > ### Comment · Reviewer_zzUK · 2022-08-08
> > > > **increasing my score**
> > > >
> > > > Thanks for the explanations. I think the responses address most of my previous doubts. I will increase my score meanwhile suggest making these points clarified in the main paper to help the understanding for future readers.

---

### Meta-Review · Area_Chair_k81h · 2022-08-31

**Recommendation:** Accept
**Confidence:** Certain

**Metareview:**

This paper considers instance-dependent complexity of learning under linear MDPs. Based on a novel online experiment design in linear MDPs, the author proposes an algorithm whose sample complexity scaling with the instance-dependent complexity. This algorithm is worst-case optimal and beats any low-regret algorithm on an explicit example constructed by the author. All reviewers are convinced by the contribution of this paper, and we recommend acceptance.

**Award:**

No

---

### Decision · Program_Chairs · 2022-09-14

Accept